# Mind the spikes: Benign overfitting of kernels and neural networks in fixed dimension

**Moritz Haas**[1]* **David Holzmüller**[2]* **Ulrike von Luxburg**[1] **Ingo Steinwart**[2]

[1]University of Tübingen and Tübingen AI Center, Germany

[2]Institute for Stochastics and Applications, University of Stuttgart, Germany

{mo.haas,ulrike.luxburg}@uni-tuebingen.de
{david.holzmueller,ingo.steinwart}@mathematik.uni-stuttgart.de

## Abstract

The success of over-parameterized neural networks trained to near-zero training error has caused great interest in the phenomenon of benign overfitting, where estimators are statistically consistent even though they interpolate noisy training data. While benign overfitting in fixed dimension has been established for some learning methods, current literature suggests that for regression with typical kernel methods and wide neural networks, benign overfitting requires a high-dimensional setting where the dimension grows with the sample size. In this paper, we show that the smoothness of the estimators, and not the dimension, is the key: benign overfitting is possible if and only if the estimator's derivatives are large enough. We generalize existing inconsistency results to non-interpolating models and more kernels to show that benign overfitting with moderate derivatives is impossible in fixed dimension. Conversely, we show that rate-optimal benign overfitting is possible for regression with a sequence of spiky-smooth kernels with large derivatives. Using neural tangent kernels, we translate our results to wide neural networks. We prove that while infinite-width networks do not overfit benignly with the ReLU activation, this can be fixed by adding small high-frequency fluctuations to the activation function. Our experiments verify that such neural networks, while overfitting, can indeed generalize well even on low-dimensional data sets.

## 1 Introduction

While neural networks have shown great practical success, our theoretical understanding of their generalization properties is still limited. A promising line of work considers the phenomenon of benign overfitting, where researchers try to understand when and how models that interpolate noisy training data can generalize (Zhang et al., 2021, Belkin et al., 2018, 2019). In the high-dimensional regime, where the dimension grows with the number of sample points, consistency of minimum-norm interpolants has been established for linear models and kernel regression (Hastie et al., 2022, Bartlett et al., 2020, Liang and Rakhlin, 2020, Bartlett et al., 2021). In fixed dimension, minimum-norm interpolation with standard kernels is inconsistent (Rakhlin and Zhai, 2019, Buchholz, 2022).

In this paper, we shed a differentiated light on benign overfitting with kernels and neural networks. We argue that the dimension-dependent perspective does not capture the full picture of benign overfitting. In particular, we show that harmless interpolation with kernel methods and neural networks is possible, even in small fixed dimension, with adequately designed kernels and activation functions. The key is to properly design estimators of the form 'signal+spike'. While minimum-norm criteria have widely been considered a useful inductive bias, we demonstrate that designing unusual norms can resolve the shortcomings of standard norms. For wide neural networks, harmless interpolation can be

---

*Equal contribution.

realized by adding tiny fluctuations to the activation function. Such networks do not require explicit regularization and can simply be trained to overfit (Figure 1).

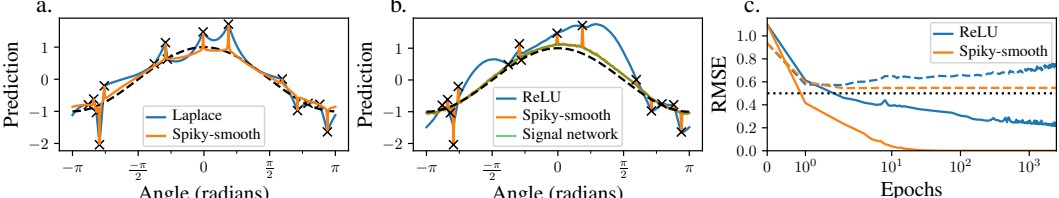

Figure 1: **Spiky-smooth overfitting in 2 dimensions. a.** We plot the predicted function for ridgeless kernel regression with the Laplace kernel (blue) versus our spiky-smooth kernel (4) with Laplace components (orange) on $\mathbb{S}^1$. The dashed black line shows the true regression function, black 'x' denote noisy training points. Further details can be found in Section 6.2. **b.** The predicted function of a trained 2-layer neural network with ReLU activation (blue) versus ReLU plus shifted high-frequency sin-function (8) (orange). Using the weights learned with the spiky-smooth activation function in a ReLU network (green) disentangles the spike component from the signal component. **c.** Training error (solid lines) and test error (dashed lines) over the course of training for b. evaluated on $10^4$ test points. The dotted black line shows the optimal test error. The spiky-smooth activation function does not require regularization and can simply be trained to overfit.

On a technical level, we additionally prove that overfitting in kernel regression can only be consistent if the estimators have large derivatives. Using neural tangent kernels or neural network Gaussian process kernels, we can translate our results from kernel regression to the world of neural networks (Neal, 1996, Jacot et al., 2018). In particular, our results enable the design of activation functions that induce benign overfitting in fixed dimension: the spikes in kernels can be translated into infinitesimal fluctuations that can be added to an activation function to achieve harmless interpolation with neural networks. Such small high frequency oscillations can fit noisy observations without affecting the smooth component too much. Training finite neural networks with gradient descent shows that spiky-smooth activation functions can indeed achieve good generalization even when interpolating small, low-dimensional data sets (Figure 1 b,c).

Thanks to new technical contributions, our inconsistency results significantly extend existing ones. We use a novel noise concentration argument (Lemma D.6) to generalize existing inconsistency results on minimum-norm interpolants to the much more realistic regime of overfitting estimators with comparable Sobolev norm scaling, which includes training via gradient flow and gradient descent with "late stopping" as well as low levels of ridge regularization. Moreover, a novel connection to eigenvalue concentration results for kernel matrices (Proposition 5) allows us to relax the smoothness assumption and to treat heteroscedastic noise in Theorem 6. Lastly, our Lemma E.1 translates inconsistency results from bounded open subsets of $\mathbb{R}^d$ to the sphere $\mathbb{S}^d$, which leads to results for the neural tangent kernel and neural network Gaussian processes.

## 2 Setup and prerequisites

**General approach.** We consider a general regression problem on $\mathbb{R}^d$ with an arbitrary, fixed dimension $d$ and analyze kernel-based approaches to solve this problem: kernel ridge regression, kernel gradient flow and gradient descent, minimum-norm interpolation, and more generally, overfitting norm-bounded estimators. We then translate our results to neural networks via the neural network Gaussian process and the neural tangent kernel. Let us now introduce the formal framework.

**Notation.** We denote scalars by lowercase letters $x$, vectors by bold lowercase letters $\boldsymbol{x}$ and matrices by bold uppercase letters $\boldsymbol{X}$. We denote the eigenvalues of $\boldsymbol{A}$ as $\lambda_1(\boldsymbol{A}) \geq \ldots \geq \lambda_n(\boldsymbol{A})$ and the Moore-Penrose pseudo-inverse by $\boldsymbol{A}^+$. We say that a probability distribution $P$ has lower and upper bounded density if its density $p$ satisfies $0 < c < p(\boldsymbol{x}) < C$ for suitable constants $c, C$ and all $\boldsymbol{x}$ on a given domain.

**Regression setup.** We consider a data set $D = ((\boldsymbol{x}_1, y_1), \ldots, (\boldsymbol{x}_n, y_n)) \in (\mathbb{R}^d \times \mathbb{R})^n$ with i.i.d. samples $(\boldsymbol{x}_i, y_i) \sim P$, written as $D \sim P^n$, where $P$ is a probability distribution on $\mathbb{R}^d \times \mathbb{R}$. We define $\boldsymbol{X} \coloneqq (\boldsymbol{x}_1, \ldots, \boldsymbol{x}_n)$ and $\boldsymbol{y} \coloneqq (y_1, \ldots, y_n)^\top \in \mathbb{R}^n$. Random variables $(\boldsymbol{x}, y) \sim P$ denote

test points independent of $D$, and $P_X$ denotes the probability distribution of $\boldsymbol{x}$. The (least squares) *empirical risk* $R_D$ and *population risk* $R_P$ of a function $f : \mathbb{R}^d \to \mathbb{R}$ are defined as

$$R_D(f) := \frac{1}{n} \sum_{i=1}^{n} (y_i - f(x_i))^2, \qquad R_P(f) := \mathbb{E}_{\boldsymbol{x},y}[(y - f(\boldsymbol{x}))^2] \ .$$

We assume $\mathrm{Var}(y|\boldsymbol{x}) < \infty$ for all $\boldsymbol{x}$. Then, $R_P$ is minimized by the target function $f_P^*(\boldsymbol{x}) = \mathbb{E}[y|\boldsymbol{x}]$, and the *excess risk* of a function $f$ is given by

$$R_P(f) - R_P(f_P^*) = \mathbb{E}_{\boldsymbol{x}}(f_P^*(\boldsymbol{x}) - f(\boldsymbol{x}))^2 \ .$$

We call a data-dependent estimator $f_D$ *consistent for* $P$ if its excess risk converges to 0 in probability, that is, for all $\varepsilon > 0$, $\lim_{n\to\infty} P^n \left( D \in (\mathbb{R}^d \times \mathbb{R})^n \mid R_P(f_D) - R_P(f_P^*) \geq \varepsilon \right) = 0$. We call $f_D$ *consistent in expectation for* $P$ if $\lim_{n\to\infty} \mathbb{E}_D R_P(f_D) - R_P(f_P^*) = 0$. We call $f_D$ *universally consistent* if is it consistent for all Borel probability measures $P$ on $\mathbb{R}^d \times \mathbb{R}$.

**Solutions by kernel regression.** Recall that a kernel $k$ induces a reproducing kernel Hilbert space $\mathcal{H}_k$, abbreviated RKHS (more details in Appendix B). For $f \in \mathcal{H}_k$, we consider the objective

$$\mathcal{L}_\rho(f) := \frac{1}{n} \sum_{i=1}^{n} (y_i - f(\boldsymbol{x}_i))^2 + \rho \|f\|_{\mathcal{H}_k}^2$$

with regularization parameter $\rho \geq 0$. Denote by $f_{t,\rho}$ the solution to this problem that is obtained by optimizing on $\mathcal{L}_\rho$ in $\mathcal{H}_k$ with gradient flow until time $t \in [0, \infty]$, using fixed a regularization constant $\rho > 0$, and initializing at $f = 0 \in \mathcal{H}_k$. We show in Appendix C.1 that it is given as

$$f_{t,\rho}(\boldsymbol{x}) := k(\boldsymbol{x}, \boldsymbol{X}) \left( \boldsymbol{I}_n - e^{-\frac{2}{n} t(k(\boldsymbol{X},\boldsymbol{X}) + \rho n \boldsymbol{I}_n)} \right) (k(\boldsymbol{X}, \boldsymbol{X}) + \rho n \boldsymbol{I}_n)^{-1} \boldsymbol{y} \ , \qquad (1)$$

where $k(\boldsymbol{x}, \boldsymbol{X})$ denotes the row vector $(k(\boldsymbol{x}, \boldsymbol{x}_i))_{i\in[n]}$ and $k(\boldsymbol{X}, \boldsymbol{X}) = (k(\boldsymbol{x}_i, \boldsymbol{x}_j))_{i,j\in[n]}$ the kernel matrix. $f_{t,\rho}$ elegantly subsumes several popular kernel regression estimators as special cases: (i) classical kernel ridge regression for $t \to \infty$, (ii) gradient flow on the unregularized objective for $\rho \searrow 0$, and (iii) kernel "ridgeless" regression $f_{\infty,0}(\boldsymbol{x}) = k(\boldsymbol{x}, \boldsymbol{X})k(\boldsymbol{X}, \boldsymbol{X})^+ \boldsymbol{y}$ in the joint limit of $\rho \to 0$ and $t \to \infty$. If $k(\boldsymbol{X}, \boldsymbol{X})$ is invertible, $f_{\infty,0}$ is the interpolating function $f \in \mathcal{H}_k$ with the smallest $\mathcal{H}_k$-norm.

**From kernels to neural networks: the neural tangent kernel (NTK) and the neural network Gaussian process (NNGP) .** Denote the output of a NN with parameters $\boldsymbol{\theta}$ on input $\boldsymbol{x}$ by $f_{\boldsymbol{\theta}}(\boldsymbol{x})$. It is known that for suitable random initializations $\boldsymbol{\theta}_0$, in the infinite-width limit the random initial function $f_{\boldsymbol{\theta}_0}$ converges in distribution to a Gaussian Process with the so-called Neural Network Gaussian Process (NNGP) kernel (Neal, 1996, Lee et al., 2018, Matthews et al., 2018). In Bayesian inference, the posterior mean function is then of the form $f_{\infty,\rho}$. With minor modifications (Arora et al., 2019, Zhang et al., 2020), training infinitely wide NNs with gradient flow corresponds to learning the function $f_{t,0}$ with the neural tangent kernel (NTK) (Jacot et al., 2018, Lee et al., 2019). If only the last layer is trained, the NNGP kernel should be used instead (Daniely et al., 2016). For ReLU activation functions, the RKHS of the infinite-width NNGP and NTK on the sphere $\mathbb{S}^d$ is typically a Sobolev space (Bietti and Bach, 2021, Chen and Xu, 2021), see Appendix B.4. Using other parametrizations induces feature learning infinite-width limits for neural networks (Yang and Hu, 2021); an analysis of such neural network algorithms is left for future work.

## 3 Related work

We here provide a short summary of related work. A more detailed account is provided in Appendix A.

**Kernel regression.** With appropriate regularization, kernel ridge regularization with typical universal kernels like the Gauss, Matérn, and Laplace kernels is universally consistent (Steinwart and Christmann, 2008, Chapter 9). Optimal rates in Sobolev RKHS can also be achieved using cross-validation of the regularization $\rho$ (Steinwart et al., 2009) or early stopping rules (Yao et al., 2007, Raskutti et al., 2014, Wei et al., 2017). The above kernels as well as NTKs and NNGPs of standard fully-connected neural networks are rotationally invariant. In the high-dimensional regime, the class of functions that is learnable with rotation-invariant kernels is quite limited (Donhauser et al., 2021, Ghorbani et al., 2021, Liang et al., 2020).

**Inconsistency results.** Besides Rakhlin and Zhai (2019) and Buchholz (2022), Beaglehole et al. (2023) derive inconsistency results for ridgeless kernel regression given assumptions on the spectral tail in the Fourier basis, and contemporaneously propose a special case of our spiky-smooth kernel sequence to mimic kernel ridge regression without providing any quantitative statements. Li et al. (2023) show that polynomial convergence is impossible for common kernels including ReLU NTKs. Mallinar et al. (2022) conjecture inconsistency for interpolation with ReLU NTKs based on their semi-rigorous result, which essentially assumes that the eigenfunctions can be replaced by structureless Gaussian random variables. Lai et al. (2023) show an inconsistency-type result for overfitting two-layer ReLU NNs with $d = 1$, but for fixed inputs $\boldsymbol{X}$. They also note that an earlier inconsistency result by Hu et al. (2021) relies on an unproven result. Mücke and Steinwart (2019) show that global minima of NNs can overfit both benignly and harmfully, but their result does not apply to gradient descent training. Overfitting with typical linear models around the interpolation peak is inconsistent (Ghosh and Belkin, 2022, Holzmüller, 2021).

**Classification.** For binary classification, benign overfitting is a more generic phenomenon than for regression (Muthukumar et al., 2021, Shamir, 2022), and consistency has been shown under linear separability assumptions (Montanari et al., 2019, Chatterji and Long, 2021, Frei et al., 2022), through complexity bounds for reference classes (Cao and Gu, 2019, Chen et al., 2021) or as long as the total variation distance of the class conditionals is sufficiently large and $f^*(\boldsymbol{x}) = \mathbb{E}[y|\boldsymbol{x}]$ lies in the RKHS with bounded norm (Liang and Recht, 2023). Chapter 8 of Steinwart and Christmann (2008) discusses how the overlap of the two classes may influence learning rates under positive regularization.

## 4 Inconsistency of overfitting with common kernel estimators

We consider a regression problem on $\mathbb{R}^d$ in arbitrary, fixed dimension $d$ that is solved by kernel regression. In this section, we derive several new results, stating that overfitting estimators with moderate Sobolev norm are inconsistent, in a variety of settings. In the next section, we establish the other direction: overfitting estimators can be consistent when we adapt the norm that is minimized.

### 4.1 Beyond minimum-norm interpolants: general overfitting estimators with bounded norm

Existing generalization bounds often consider the perfect minimum norm interpolant. This is a rather theoretical construction; estimators obtained by training with gradient descent algorithms merely overfit and, in the best case, approximate interpolants with small norm. In this section, we extend existing bounds to arbitrary overfitting estimators whose norm does not grow faster than the minimum norm that would be required to interpolate the training data. Before we can state the theorem, we need to establish some technical assumptions.

**Assumptions on the data generating process.**  The following assumptions (as in Buchholz (2022)) allow for quite general domains and distributions. They are standard in nonparametric statistics.

(D1) Let $P_X$ be a distribution on a bounded open Lipschitz domain $\Omega \subseteq \mathbb{R}^d$ with lower and upper bounded Lebesgue density. Consider data sets $D = \{(\boldsymbol{x}_1, y_1), \ldots, (\boldsymbol{x}_n, y_n)\}$, where $\boldsymbol{x}_i \sim P_X$ i.i.d. and $y_i = f^*(\boldsymbol{x}_i) + \varepsilon_i$, where $\varepsilon_i$ is i.i.d. Gaussian noise with positive variance $\sigma^2 > 0$ and $f^* \in C_c^\infty(\Omega)\backslash\{0\}$ denotes a smooth function with compact support.

**Assumptions on the kernel.**  Our assumption on the kernel is that its RKHS is equivalent to a Sobolev space. For integers $s \in \mathbb{N}$, the norm of a Sobolev space $H^s(\Omega)$ can be defined as

$$\|f\|_{H^s(\Omega)}^2 := \sum_{0 \leq |\alpha| \leq s} \|D^\alpha f\|_{L_2(\Omega)}^2,$$

where $D^\alpha$ denotes partial derivatives in multi-index notation for $\alpha$. It measures the magnitude of derivatives up to some order $s$. For general $s > 0$, $H^s(\Omega)$ is (equivalent to) an RKHS if and only if $s > d/2$. For example, Laplace and Matérn kernels (Kanagawa et al., 2018, Example 2.6) have Sobolev RKHSs. The RKHS of the Gaussian kernel $\mathcal{H}^{\text{Gauss}}$ is contained in every Sobolev space, $\mathcal{H}^{\text{Gauss}} \subsetneq H^s$ for all $s \geq 0$ (Steinwart and Christmann, 2008, Corollary 4.36). Due to its smoothness, the Gaussian kernel is potentially even more prone to harmful overfitting than Sobolev kernels (Mallinar et al., 2022). We make the following assumption on the kernel:

(K) Let $k$ be a positive definite kernel function whose RKHS $\mathcal{H}_k$ is equivalent to the Sobolev space $H^s$ for some $s \in (\frac{d}{2}, \frac{3d}{4}]$.

Now we are ready to state the main result of this section:

**Theorem 1** (**Overfitting estimators with small norms are inconsistent**). *Let assumptions (D1) and (K) hold. Let $c_{\text{fit}} \in (0, 1]$ and $C_{\text{norm}} > 0$. Then, there exist $c > 0$ and $n_0 \in \mathbb{N}$ such that the following holds for all $n \geq n_0$ with probability $1 - O(1/n)$ over the draw of the data set $D$ with $n$ samples: Every function $f \in \mathcal{H}_k$ that satisfies the follwing two conditions*

*(O) $\frac{1}{n} \sum_{i=1}^{n} (f(x_i) - y_i)^2 \leq (1 - c_{\text{fit}}) \cdot \sigma^2$ (training error of $f$ is below Bayes risk)*
*(N) $\|f\|_{\mathcal{H}_k} \leq C_{\text{norm}} \|f_{\infty,0}\|_{\mathcal{H}_k}$ (norm comparable to minimum-norm interpolant (1)),*

*has an excess risk that satisfies*

$$R_P(f) - R_P(f^*) \geq c\sigma^2 > 0 \,. \tag{2}$$

In words: In fixed dimension $d$, every differentiable function $f$ that overfits the training data and is not much "spikier" than the minimum RKHS-norm interpolant is inconsistent!

**Proof idea.** Our proof follows a similar approach as Rakhlin and Zhai (2019), Buchholz (2022), and also holds for kernels with adaptive bandwidths. For small bandwidths, $\|f_{\infty,0}\|_{L_2(P_X)} \ll \|f^*\|_{L_2(P_X)}$ because $f_{\infty,0}$ decays to 0 between the training points, which shows that purely 'spiky' estimators are inconsistent. In this case, the lower bound is independent of $\sigma^2$. For all other bandwidths, interpolating $\Theta(n)$ many noisy labels $y_i$ incurs $\Theta(1)$ error in an area of volume $\Omega(1/n)$ around $\Theta(n)$ data points with high probability, which accumulates to a total error $\Omega(1)$. Our observation is that the same logic holds when overfitting by a constant fraction. Formally, we show that $f^*$ and $f$ must then be separated by a constant on a constant fraction of training points, with high probability, by using the fact that a constant fraction of the total noise cannot concentrate on less than $\Theta(n)$ noise variables, with high probability (Lemma D.6). The full proof can be found in Appendix D. □

Assumption (O) is necessary in Theorem 1, because optimally regularized kernel ridge regression fulfills all other assumptions of Theorem 1 while achieving consistency with minimax optimal convergence rates (see Section 3). The necessity of Assumption (N) is demonstrated by Section 5.

The following proposition establishes that Theorem 1 covers the entire overfitting regime of the popular (regularized) gradient flow estimators $f_{t,\rho}$ for all times $t \in [0, \infty]$ and any regularization $\rho \geq 0$. The proof in Appendix C.2 also covers gradient descent.

**Proposition 2** (**Popular estimators fulfill the norm bound (N)**). *For arbitrary $t \in [0, \infty]$ and $\rho \in [0, \infty)$, $f_{t,\rho}$ as defined in (1) fulfills Assumption (N) with $C_{\text{norm}} = 1$.*

**Remark 3** (**Dimension dependency**). Some works argue that for specific sequences of kernels $(k_d)_{d \in \mathbb{N}}$, the constant $c$ in Theorem 1 decreases with increasing dimension $d$ (Liang et al., 2020, Liang and Rakhlin, 2020, Mallinar et al., 2022). In Theorem 1, if the equivalence constants in Assumption (K) are uniformly bounded in $d$, the behavior in $d$ might still depend on the definition of the Sobolev norms. Overall, similar to Rakhlin and Zhai (2019) and Buchholz (2022), our proof techniques do not allow to easily obtain a dependence on $d$. ◀

## 4.2 Inconsistency of overfitting with neural kernels

We would now like to apply the above results to neural kernels, which would allow us to translate our inconsistency results from the kernel domain to neural networks. However, to achieve this, we need to take one more technical hurdle: the equivalence results for NTKs and NNGPs only hold for probability distributions on the sphere $\mathbb{S}^d$ (detailed summary in Appendix B.4). Lemma E.1 provides the missing technical link: It establishes a smooth correspondence between the respective kernels, Sobolev spaces, and probability distributions. The inconsistency of overfitting with (deep) ReLU NTKs and NNGP kernels then immediately follows from adapting Theorem 1 via Lemma E.1.

**Theorem 4** (**Overfitting with neural network kernels in fixed dimension is inconsistent**). *Let $c \in (0, 1)$, and let $P$ be a probability distribution with lower and upper bounded Lebesgue density on an arbitrary spherical cap $T := \{\boldsymbol{x} \in \mathbb{S}^d \mid x_{d+1} < v\} \subseteq \mathbb{S}^d$, $v \in (-1, 1)$. Let $k$ either be*

*(i) the fully-connected ReLU NTK with 0-initialized biases of any fixed depth $L \geq 2$, and $d \geq 2$, or*
*(ii) the fully-connected ReLU NNGP kernel without biases of any fixed depth $L \geq 3$, and $d \geq 6$.*

*Then, if $f_{t,\rho}$ fulfills Assumption (O) with probability at least c over the draw of the data set D, $f_{t,\rho}$ is inconsistent for P.*

Theorem 4 also holds for more general estimators as in Theorem 1, cf. the proof in Appendix E.

Mallinar et al. (2022) already observed empirically that overfitting common network architectures yields suboptimal generalization performance on large data sets in fixed dimension. Theorem 4 now provides a rigorous proof for this phenomenon since sufficiently wide trained neural networks and the corresponding NTKs have a similar generalization behavior (e.g. (Arora et al., 2019, Theorem 3.2)).

### 4.3 Relaxing smoothness and noise assumptions via spectral concentration bounds

In this section, we consider a different approach to derive lower bounds for the generalization error of overfitting kernel regression: through concentration results for the eigenvalues of kernel matrices. On a high level, we obtain similar results as in the last section. The novelty of this section is on the technical side, and we suggest that non-technical readers skip this section in their first reading.

We define the convolution kernel of a given kernel $k$ as $k_*(\boldsymbol{x}, \boldsymbol{x}') \coloneqq \int k(\boldsymbol{x}, \boldsymbol{x}'') k(\boldsymbol{x}'', \boldsymbol{x}') \, \mathrm{d}P_X(\boldsymbol{x}'')$, which is possible whenever $k(\boldsymbol{x}, \cdot) \in L_2(P_X)$ for all $\boldsymbol{x}$. The latter condition is satisfied for bounded kernels. Our starting point is the following new lower bound:

**Proposition 5** (**Spectral lower bound**). *Assume that the kernel matrix $k(\boldsymbol{X}, \boldsymbol{X})$ is almost surely positive definite, and that $\mathrm{Var}(y|\boldsymbol{x}) \geq \sigma^2$ for $P_X$-almost all $\boldsymbol{x}$. Then, the expected excess risk satisfies*

$$\mathbb{E}_D R_P(f_{t,\rho}) - R_P^* \geq \frac{\sigma^2}{n} \sum_{i=1}^n \mathbb{E}_{\boldsymbol{X}} \frac{\lambda_i(k_*(\boldsymbol{X},\boldsymbol{X})/n) \left(1 - e^{-2t(\lambda_i(k(\boldsymbol{X},\boldsymbol{X})/n)+\rho)}\right)^2}{(\lambda_i(k(\boldsymbol{X},\boldsymbol{X})/n)+\rho)^2} \,. \quad (3)$$

Using concentration inequalities for kernel matrices and the relation between the integral operators of $k$ and $k_*$, it can be seen that for $t = \infty$ and $\rho = 0$, every term in the sum in Eq. (3) should converge to 1 as $n \to \infty$. However, since the number of terms in the sum increases with $n$ and the convergence may not be uniform, this is not sufficient to show inconsistency in expectation. Instead, relative concentration bounds that are even stronger than the ones by Valdivia (2018) would be required to show inconsistency in expectation. However, by combining multiple weaker bounds and further arguments on kernel equivalences, we can still show inconsistency in expectation for a class of dot-product kernels on the sphere, including certain NTK and NNGP kernels (Appendix B.4):

**Theorem 6** (**Inconsistency for Sobolev dot-product kernels on the sphere**). *Let $k$ be a dot-product kernel on $\mathbb{S}^d$, i.e., a kernel of the form $k(\boldsymbol{x}, \boldsymbol{x}') = \kappa(\langle \boldsymbol{x}, \boldsymbol{x}' \rangle)$, such that its RKHS $\mathcal{H}_k$ is equivalent to a Sobolev space $H^s(\mathbb{S}^d)$, $s > d/2$. Moreover, let $P$ be a distribution on $\mathbb{S}^d \times \mathbb{R}$ such that $P_X$ has a lower and upper bounded density w.r.t. the uniform distribution $\mathcal{U}(\mathbb{S}^d)$, and such that $\mathrm{Var}(y|\boldsymbol{x}) \geq \sigma^2 > 0$ for $P_X$-almost all $\boldsymbol{x} \in \mathbb{S}^d$. Then, for every $C > 0$, there exists $c > 0$ independent of $\sigma^2$ such that for all $n \geq 1$, $t \in (C^{-1}n^{2s/d}, \infty]$, and $\rho \in [0, Cn^{-2s/d})$, the expected excess risk satisfies*

$$\mathbb{E}_D R_P(f_{t,\rho}) - R_P^* \geq c\sigma^2 > 0 \,.$$

The assumptions of Theorem 6 and Theorem 4 differ in several ways. Theorem 6 applies to arbitrarily high smoothness $s$ and therefore to ReLU NTKs and NNGPs in arbitrary dimension $d$. Moreover, it applies to distributions on the whole sphere and allows more general noise distributions. On the flip side, it only shows inconsistency in expectation, which we believe could be extended to inconsistency for Gaussian noise. Moreover, it only applies to functions of the form $f_{t,\rho}$ but provides an explicit bound on $t$ and $\rho$ to get inconsistency. For $t = \infty$, the bound $\rho = O(n^{-2s/d})$ appears to be tight, as larger $\rho$ yield consistency for comparable Sobolev kernels on $\mathbb{R}^d$ (Steinwart et al., 2009, Corollary 3).

We only prove Theorem 6 for dot-product kernels on the sphere since we can show for these kernels that $\mathcal{H}_{k_*}$ is equivalent to a Sobolev space (Lemma F.13), while this is not true for open domains $\Omega$ (Schaback, 2018). However, an improved understanding of $\mathcal{H}_{k_*}$ for such $\Omega$ could potentially allow to extend our proof to the non-spherical case.

The spectral lower bounds in Theorem F.2 show that our approach can directly benefit from developing better kernel matrix concentration inequalities. Conversely, the investigation of consistent kernel interpolation might provide information about where such concentration inequalities do not hold.

# 5 Consistency via spiky-smooth estimators – even in fixed dimension

In Section 4, we have seen that when common kernel estimators overfit, they are inconsistent for many kernels and a wide variety of distributions. We now design consistent interpolating kernel estimators. The key is to violate Assumption (N) for every fixed Sobolev RKHS norm $\| \cdot \|_{\mathcal{H}_k}$ and introduce an inductive bias towards learning spiky-smooth functions.

## 5.1 Almost universal consistency of spiky-smooth ridgeless kernel regression

In high dimensional regimes (where the dimension $d$ is supposed to grow with the number of data points), benign overfitting of linear and kernel regression has been understood by an additive decomposition of the minimum-norm interpolant into a smooth regularized component that is responsible for good generalization, and a spiky component that interpolates the noisy data points while not harming generalization (Bartlett et al., 2021). This inspires us to enforce such a decomposition in arbitrary fixed dimension by adding a sharp kernel spike $\rho \check{k}_{\gamma_n}$ to a common kernel $\tilde{k}$. In this way, we can still generate any Sobolev RKHS (see Appendix G.2).

**Definition 7 (Spiky-smooth kernel).** Let $\tilde{k}$ denote any universal kernel function on $\mathbb{R}^d$. We call it the smooth component. Consider a second, translation invariant kernel $\check{k}_\gamma$ of the form $k_\gamma(\boldsymbol{x}, \boldsymbol{y}) = q(\frac{\boldsymbol{x}-\boldsymbol{y}}{\gamma})$, for some function $q : \mathbb{R}^d \to \mathbb{R}$. We call it the spiky component. Then we define the $\rho$-regularized spiky-smooth kernel with spike bandwidth $\gamma$ as

$$k_{\rho,\gamma}(\boldsymbol{x}, \boldsymbol{y}) = \tilde{k}(\boldsymbol{x}, \boldsymbol{y}) + \rho \cdot \check{k}_\gamma(\boldsymbol{x}, \boldsymbol{y}), \qquad \boldsymbol{x}, \boldsymbol{y} \in \mathbb{R}^d. \quad (4)$$

Figure 2: The spiky-smooth kernel with Laplace components (orange) consists of a Laplace kernel (blue) plus a Laplace kernel of height $\rho$ and small bandwidth $\gamma$.

We now show that the minimum-norm interpolant of the spiky-smooth kernel sequence with properly chosen $\rho_n, \gamma_n \to 0$ is consistent for a large class of distributions, on a space with fixed (possibly small) dimension $d$. We establish our result under the following assumption (as in Mücke and Steinwart (2019)), which is weaker than our previous Assumption (D1).

(D2) There exists a constant $\beta_X > 0$ and a continuous function $\phi : [0, \infty) \to [0, 1]$ with $\phi(0) = 0$ such that the data generating probability distribution satisfies $P_X(B_t(\boldsymbol{x})) \leq \phi(t) = O(t^{\beta_X})$ for all $\boldsymbol{x} \in \Omega$ and all $t \geq 0$ (here $B_t(\boldsymbol{x})$ denotes the Euclidean ball of radius $t$ around $\boldsymbol{x}$).

**Theorem 8 (Consistency of spiky-smooth ridgeless kernel regression).** *Assume that the training set D consists of n i.i.d. pairs $(\boldsymbol{x}, y) \sim P$ such that the marginal $P_X$ fulfills (D2) and $\mathbb{E}y^2 < \infty$. Let the kernel components satisfy:*

- *$\tilde{k}$ is a universal kernel, and $\rho_n \to 0$ and $n\rho_n^4 \to \infty$.*
- *$\check{k}_{\gamma_n}$ denotes the Laplace kernel with a sequence of positive bandwidths $(\gamma_n)$ fulfilling $\gamma_n \leq n^{-\frac{2+\alpha}{d}} \left( (\frac{9}{4} + \frac{\alpha}{2}) \ln n \right)^{-1}$, where $\alpha > 0$ arbitrary.*

*Then the minimum-norm interpolant of the $\rho_n$-regularized spiky-smooth kernel sequence $k_n := k_{\rho_n,\gamma_n}$ is consistent for P.*

**Remark 9 (Benign overfitting with optimal convergence rates).** Suppose that we have a Sobolev target function $f^* \in H^{s^*}(\Omega)\backslash\{0\}$, that the noise satisfies a moment condition and that $P_X$ has an upper- and lower-bounded density on a Lipschitz domain or the sphere. Then, we show in Theorem G.5 that, by using smooth components $\tilde{k}$ whose RKHS is equivalent to a Sobolev space $H^s$, $s > \max(s^*, d/2)$, and choosing the spike components $\check{k}_{\gamma_n}$ as in Theorem 8, the minimum-norm interpolant of $k_n := k_{\rho_n,\gamma_n}$ achieves the convergence rate $n^{-\frac{s^*}{(s^*+d/2)}} \log^2(n)$ when choosing the quasi-regularization $\rho_n$ properly. Moreover, for $s^* > d/2$, this rate is known to be optimal up to the factor $\log^2(n)$ (Remark G.6). Since optimal rates can be achieved both with optimal regularization and with interpolation, our results show that in Sobolev RKHSs, overfitting is neither intrinsically helpful nor harmful for generalization with the right choice of kernel function. ◄

**Proof idea.** With sharp spikes $\gamma \to 0$, it holds that $\check{k}_\gamma(\boldsymbol{X}, \boldsymbol{X}) \approx \boldsymbol{I}_n$, with high probability. Hence, ridgeless kernel regression with the spiky-smooth kernel interpolates the training set while approximating kernel ridge regression with the smooth component $\tilde{k}$ and regularization $\rho$. $\qquad\square$

The theorem even holds under much weaker assumptions on the decay behavior of the spike component $\check{k}_{\gamma_n}$, including Gaussian and Matérn kernels. The full version of the theorem and its proof can be found in Appendix G. It also applies to kernels and distributions on the sphere $\mathbb{S}^d$.

**Remark 10** (**Interplay between smoothness and dimensionality**). Irrespective of the dimension $d$, we achieve benign overfitting with estimators in RKHS of arbitrary degrees of smoothness. With increasing $d$, for the Laplace kernel the spike bandwidth is allowed to be chosen as $\gamma_n = \Omega(n^{-(2+\alpha)/d})$, $\alpha > 0$, for covariate distributions with upper bounded Lebesgue density (see Remark G.2). Hence the magnitude of derivatives of the spikes is allowed to scale less aggressively with increasing dimension. $\qquad\blacktriangleleft$

### 5.2 From spiky-smooth kernels to spiky-smooth activation functions

So far, our discussion revolved around the properties of kernels and whether they lead to estimators that are consistent. We now turn our attention to the neural network side. The big question is whether it is possible to specifically design activation functions that enable benign overfitting in fixed, possibly small dimension. We will see that the answer is yes: similarly to adding sharp spikes to a kernel, we add tiny fluctuations to the activation function. Concretely, we exploit (Simon et al., 2022, Theorem 3.1). It states that any dot-product kernel on the sphere that is a dot-product kernel in every dimension $d$ can be written as an NNGP kernel or an NTK of two-layer fully-connected networks with a specifically chosen activation function. Further details can be found in Appendix H.

**Theorem 11** (**Connecting kernels and activation functions** (Simon et al., 2022)). *Let $\kappa : [-1, 1] \to \mathbb{R}$ be a function such that $k_d : \mathbb{S}^d \times \mathbb{S}^d \to \mathbb{R}$, $k_d(\boldsymbol{x}, \boldsymbol{x}') = \kappa(\langle \boldsymbol{x}, \boldsymbol{x}' \rangle)$ is a kernel for every $d \geq 1$. Then, there exist $b_i \geq 0$ with $\sum_{i=0}^{\infty} b_i < \infty$ such that $\kappa(t) = \sum_{i=0}^{\infty} b_i t^i$, and for any choice of signs $(s_i)_{i \in \mathbb{N}_0} \subseteq \{-1, +1\}$, the kernel $k_d$ can be realized as the NNGP kernel or NTK of a two-layer fully-connected network without biases and with activation function*

$$\phi_{NNGP}^{k_d}(x) = \sum_{i=0}^{\infty} s_i (b_i)^{1/2} h_i(x), \qquad \phi_{NTK}^{k_d}(x) = \sum_{i=0}^{\infty} s_i \left( \frac{b_i}{i+1} \right)^{1/2} h_i(x). \qquad (5)$$

*Here, $h_i$ denotes the $i$-th Probabilist's Hermite polynomial normalized such that $\|h_i\|_{L_2(\mathcal{N}(0,1))} = 1$.*

The following proposition justifies the approach of adding spikes $\rho^{1/2} \phi^{\check{k}_\gamma}$ to an activation function to enable harmless interpolation with wide neural networks. Here we state the result for the case of the NTK; an analogous result holds for induced NNGP activation functions.

**Proposition 12** (**Additive decomposition of spiky-smooth activation functions**). *Fix $\tilde{\gamma}, \rho > 0$ arbitrary. Let $k = \tilde{k} + \rho \check{k}_\gamma$ denote the spiky-smooth kernel where $\tilde{k}$ and $\check{k}_\gamma$ are Gaussian kernels of bandwidth $\tilde{\gamma}$ and $\gamma$, respectively. Assume that we choose signs $\{s_i\}_{i \in \mathbb{N}}$ and then the activation functions $\phi_{NTK}^k$, $\phi_{NTK}^{\tilde{k}}$ and $\phi_{NTK}^{\check{k}_\gamma}$ as in Theorem 11. Then, for $\gamma > 0$ small enough, it holds that*

$$\|\phi_{NTK}^k - (\phi_{NTK}^{\tilde{k}} + \sqrt{\rho} \cdot \phi_{NTK}^{\check{k}_\gamma})\|_{L_2(\mathcal{N}(0,1))}^2 \leq 2^{1/2} \rho \gamma^{3/2} \exp\left( -\frac{1}{\gamma} \right) + \frac{4\pi(1+\tilde{\gamma})\gamma}{\tilde{\gamma}}.$$

**Proof idea.** When the spikes are sharp enough ($\gamma$ small enough), the smooth and the spiky component of the activation function are approximately orthogonal in $L_2(\mathcal{N}(0,1))$ (Figure 3c), so that the spiky-smooth activation function can be approximately additively decomposed into the smooth activation component $\phi^{\tilde{k}}$ and the spike component $\phi^{\check{k}}$ responsible for interpolation. $\qquad\square$

To motivate why the added spike functions $\rho^{1/2} \phi^{\check{k}_\gamma}$ should have small amplitudes, observe that Gaussian activation components $\phi^{\check{k}_\gamma}$ satisfy

$$\|\phi_{NNGP}^{\check{k}_\gamma}\|_{L_2(\mathcal{N}(0,1))}^2 = 1, \qquad \|\phi_{NTK}^{\check{k}_\gamma}\|_{L_2(\mathcal{N}(0,1))}^2 = \frac{\gamma}{2}\left( 1 - \exp\left( -\frac{2}{\gamma} \right) \right). \qquad (6)$$

Hence, the average amplitude of NNGP spike activation components $\rho^{1/2} \phi^{\check{k}_\gamma}$ does not depend on $\gamma$, while the average amplitude of NTK spike components decays to 0 with $\gamma \to 0$. Since consistency requires the quasi-regularization $\rho \to 0$, the spiky component of induced NTK as well as NNGP activation functions should vanish for large data sets $n \to \infty$ to achieve consistency.

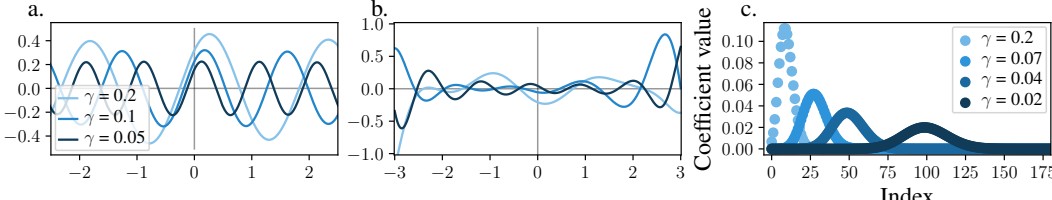

Figure 3: **a., b.** Gaussian NTK activation components $\phi_{NTK}^{\check{k}_\gamma}$ defined via (5) induced by the Gaussian kernel with varying bandwidth $\gamma \in [0.2, 0.1, 0.05]$ (the darker, the smaller $\gamma$) for **a.** bi-alternating signs $s_i = +1$ iff $\lfloor i/2 \rfloor$ even, and **b.** randomly iid chosen signs $s_i \sim \mathcal{U}(\{-1, +1\})$. **c.** Coefficients of the Hermite series of a Gaussian NTK activation component with varying bandwidth $\gamma$. Observe peaks at $2/\gamma$. For reliable approximations of activation functions use a truncation $\geq 4/\gamma$. The sum of squares of the coefficients follows Eq. (6). Figure I.8 visualizes NNGP activation components.

## 6 Experiments

Now we explore how appropriate spiky-smooth activation functions might look like and whether they indeed enable harmless interpolation for trained networks of finite width on finite data sets. Further experimental results are reported in Appendix I.

### 6.1 What do common activation functions lack in order to achieve harmless interpolation?

To understand which properties we have to introduce into activation functions to enable harmless interpolation, we plot NTK spike components $\phi^{\check{k}_\gamma}$ induced by the Gaussian kernel (Figure 3a,b) as well as their Hermite series coefficients (Figure 3c). Remarkably, the spike components $\phi^{\check{k}_\gamma}$ approximately correspond to a shifted, high-frequency sin-curve, when choosing the signs $s_i$ in (5) to alternate every second $i$, that is $s_i = +1$ iff $\lfloor i/2 \rfloor$ even (Figure 3a). Proposition H.1 shows that the NNGP activation functions correspond to the fluctuation function

$$\omega_{\mathrm{NNGP}}(x; \gamma) := \sqrt{2} \cdot \sin\left(\sqrt{2/\gamma} \cdot x + \pi/4\right) = \sin\left(\sqrt{2/\gamma} \cdot x\right) + \cos\left(\sqrt{2/\gamma} \cdot x\right), \quad (7)$$

where the last equation follows from the trigonometric addition theorem. For small bandwidths $\gamma$, the NTK activation functions are increasingly well approximated (Appendix I.6) by

$$\omega_{\mathrm{NTK}}(x; \gamma) := \sqrt{\gamma} \cdot \sin\left(\sqrt{2/\gamma} \cdot x + \pi/4\right) = \sqrt{\gamma/2}\left(\sin\left(\sqrt{2/\gamma} \cdot x\right) + \cos\left(\sqrt{2/\gamma} \cdot x\right)\right). \quad (8)$$

With decreasing bandwidth $\gamma \to 0$ the frequency increases, while the amplitude decreases for the NTK and remains constant for the NNGP (see Eq. (6)). Plotting equivalent spike components $\phi^{\check{k}_\gamma}$ with different choices of the signs $s_i$ (Figure 3b and Appendix I.5) suggests that harmless interpolation requires activation functions that contain **small high-frequency oscillations** or that **explode at large** $|x|$, which only affects few neurons. The Hermite series expansion of suitable activation functions should contain **non-negligible weight spread across high-order coefficients** (Figure 3c). While Simon et al. (2022) already truncate the Hermite series of induced activation functions at order 5, Figure 3c shows that an accurate approximation of spiky-smooth activation functions requires the truncation index to be larger than $2/\gamma$. Only a careful implementation allows us to capture the high-order fluctuations in the Hermite series of the spiky activation functions. Our implementation can be found at `https://github.com/moritzhaas/mind-the-spikes`.

### 6.2 Training neural networks to achieve harmless interpolation in low dimension

In Figure 1, we plot the results of (a) ridgeless kernel regression and (b) trained 2-layer neural networks with standard choices of kernels and activation functions (blue) as well as our spiky-smooth alternatives (orange). We trained on 15 points sampled i.i.d. from $x = (x_1, x_2) \sim \mathcal{U}(\mathbb{S}^1)$ and $y = x_1 + \varepsilon$ with $\varepsilon \sim \mathcal{N}(0, 0.25)$. The figure shows that both the Laplace kernel and standard ReLU networks interpolate the training data too smoothly in low dimension, and do not generalize well. However, our spiky-smooth kernel and neural networks with spiky-smooth activation functions achieve close to optimal generalization while interpolating the training data with sharp spikes.

We achieve this by using the adjusted activation function with high-frequency oscillations $x \mapsto \mathrm{ReLU}(x) + \omega_{\mathrm{NTK}}(x; \frac{1}{5000})$ as defined in Eq. (8). With this choice, we avoid activation functions with exploding behavior, which would induce exploding gradients. Other choices of amplitude and frequency in Eq. (8) perform worse. To bring our neural networks close to the kernel regime, we use the neural tangent parameterization (Jacot et al., 2018) and make the networks very wide (20000 hidden neurons). To ensure that the initial function is identically zero, we use the antisymmetric initialization trick (Zhang et al., 2020). Over the course of training (Figure 1c), the standard ReLU network exhibits harmful overfitting, whereas the NN with a spiky-smooth activation function quickly interpolates the training set with nearly optimal generalization. Training details and hyperparameter choices can be found in Appendix I.1. Although the high-frequency oscillations perturb the gradients, the NN with spiky smooth activation has a stable training trajectory using gradient descent with a large learning rate of $0.4$ or stochastic gradient descent with a learning rate of $0.04$. Since our activation function is the sum of two terms, we can additively decompose the network into its ReLU-component and its $\omega_{\mathrm{NTK}}$-component. Figure 1b and Appendix I.2 demonstrate that our interpretation of the $\omega_{\mathrm{NTK}}$-component as 'spiky' is accurate: The oscillations in the hidden neurons induced by $\omega_{\mathrm{NTK}}$ interfere constructively to interpolate the noise in the training points and regress to $0$ between training points. This entails immediate access to the signal component of the trained neural network in form of its ReLU-component.

## 7 Conclusion

Conceptually, our work shows that inconsistency of overfitting is quite a generic phenomenon for regression in fixed dimension. However, particular spiky-smooth estimators enable benign overfitting, even in fixed dimension. We translate the spikes that lead to benign overfitting in kernel regression into infinitesimal fluctuations that can be added to activation functions to consistently interpolate with wide neural networks. Our experiments verify that neural networks with spiky-smooth activation functions can exhibit benign overfitting even on small, low-dimensional data sets.

Technically, our inconsistency results cover many distributions, Sobolev spaces of arbitrary order, and arbitrary RKHS-norm-bounded overfitting estimators. Lemma E.1 serves as a generic tool to extend generalization bounds to the sphere $\mathbb{S}^d$, allowing us to cover (deep) ReLU NTKs and ReLU NNGPs.

**Future work.** While our experiments serve as a promising proof of concept, it remains unclear how to design activation functions that enable harmless interpolation of more complex neural network architectures and data sets. As another interesting insight, our consistent kernel sequence shows that although kernels may have equivalent RKHS (see Appendix G.2), their generalization error can differ arbitrarily much; the constants of the equivalence matter and the narrative that depth does not matter in the NTK regime as in Bietti and Bach (2021) is too simplified. More promisingly, analyses that extend our analysis in the infinite-width limit to a joint scaling of width and depth could help us to understand the influence of depth (Fort et al., 2020, Li et al., 2021, Seleznova and Kutyniok, 2022). Finite-sample analyses of moderate-width neural networks with feature learning parametrizations (Yang and Hu, 2021) and other initializations could enable to understand how to induce a spiky-smooth inductive bias in feature learning neural architectures.

## Acknowledgments and Disclosure of Funding

Funded by Deutsche Forschungsgemeinschaft (DFG, German Research Foundation) under Germany's Excellence Strategy - EXC 2075 - 390740016 and EXC 2064/1 - Project 390727645, as well as the DFG Priority Program 2298/1, project STE 1074/5-1. The authors thank the International Max Planck Research School for Intelligent Systems (IMPRS-IS) for supporting Moritz Haas and David Holzmüller. We want to thank Tizian Wenzel and Václav Voráček for interesting discussions. We also thank Nadine Große, Jens Wirth, and Daniel Winkle for helpful comments on Sobolev spaces, and Nilotpal Sinha for pointing us to Laurent series.

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

# Appendices

**Appendix Contents.**

# A Detailed related work

Motivated by Zhang et al. (2021) and Belkin et al. (2018), an abundance of papers have tried to grasp when and how benign overfitting occurs in different settings. Rigorous understanding is mainly restricted to linear (Bartlett et al., 2020), feature (Hastie et al., 2022) and kernel regression (Liang and Rakhlin, 2020) under restrictive distributional assumptions. In the well-specified linear setting under additional assumptions, the minimum-norm interpolant is consistent if and only if $k \ll n \ll d$, the top-$k$ eigendirections of the covariate covariance matrix align with the signal, followed by sufficiently many 'quasi-isotropic' directions with eigenvalues of similar magnitude (Bartlett et al., 2020).

**Kernel methods.** The analysis of kernel methods is more nuanced and depends on the interplay between the chosen kernel, the choice of regularization and the data distribution. $L_2$-generalization error bounds can be derived in the eigenbasis of the kernel's integral operator (Mcrae et al., 2022), where upper bounds of the form $\sqrt{\boldsymbol{y}^\top k(\boldsymbol{X}, \boldsymbol{X})^{-1} \boldsymbol{y}/n}$ promise good generalization when the regression function $f^*$ is aligned with the dominant eigendirections of the kernel, or in other words, when $\|f^*\|_{\mathcal{H}}$ is small. Most recent work focuses on high-dimensional limits, where the data dimensionality $d \to \infty$. For $d \to \infty$, the Hilbert space and its norm change, so that consistency results that demand bounded Hilbert norm of rotation-invariant kernels do not even include simple functions like sparse products (Donhauser et al., 2021, Lemma 2.1). In the regime $d^{l+\delta} \le n \le d^{l+1-\delta}$, rotation-invariant (neural) kernel methods (Ghorbani et al., 2021, Donhauser et al., 2021) can in fact only learn the polynomial parts up to order $l$ of the regression function $f^*$, and fully-connected NTKs do so. Liang et al. (2020) uncover a related multiple descent phenomenon in kernel regression, where the risk vanishes for most $n \to \infty$, but peaks at $n = d^i$ for all $i \in \mathbb{N}$. The slower $d$ grows, the slower the optimal rate $n^{-\frac{1}{2i+1}}$ between the peaks. Note, however, that these bounds are only upper bounds, and whether they are optimal remains an open question to the best of our knowledge. Another recent line of work analyzes how different inductive biases, measured in $\|\cdot\|_p$-norm minimization, $p \in [1, 2]$, (Donhauser et al., 2022) or in the filter size of convolutional kernels (Aerni et al., 2023), affects the generalization properties of minimum-norm interpolants. While the risk on noiseless training samples (bias) decreases with decreasing $p$ or small filter size, the sensitivity to noise in the training data (variance) increases. Hence only 'weak inductive biases', that is large $p$ or large filter sizes, enable harmless interpolation. Our results suggest that to achieve harmless interpolation in fixed dimension one has to construct and minimize more unusual norms than $\|\cdot\|_p$-norms.

**Regularised kernel regression achieves optimal rates.** With appropriate regularization, kernel ridge regularization with typical universal kernels like the Gauss, Matérn, and Laplace kernels is universally consistent (Steinwart and Christmann, 2008, Chapter 9). Steinwart et al. (2009, Corollary 6) even implies minimax optimal nonparametric rates for clipped kernel ridge regression with Sobolev kernels and $f^* \in H^\beta$ where $d/2 < \beta \le s$ for the choice $\rho_n = n^{-2s/(2\beta+d)}$. Although $f^*$ is not necessarily in the RKHS, KRR is adaptive and can still achieve optimal learning rates. Lower smoothness $\beta$ of $f^*$ as well as higher smoothness of the kernel should be met with faster decay of $\rho_n$. Optimal rates in Sobolev RKHS can also be achieved using cross-validation of the regularization $\rho$ (Steinwart et al., 2009), early stopping rules based on empirical localized Rademacher (Raskutti et al., 2014) or Gaussian complexity (Wei et al., 2017) or smoothing of the empirical risk via kernel matrix eigenvalues (Averyanov and Celisse, 2020).

**Lower bounds for kernel regression.** Besides Rakhlin and Zhai (2019) and Buchholz (2022), Beaglehole et al. (2023) derive inconsistency results for kernel ridgeless regression given assumptions on the spectral tail in the Fourier basis. Mallinar et al. (2022) provide a characterization of kernel ridge regression into benign, tempered and catastrophic overfitting using a *heuristic* approximation of the risk via the kernel's eigenspectrum, essentially assuming that the eigenfunctions can be replaced by structureless Gaussian random variables. A general lower bound for ridgeless linear regression Holzmüller (2021) predicts bad generalization near the "interpolation threshold", where the dimension of the feature space is close to $n$, also known as the *double descent* phenomenon. In this regime, Ghosh and Belkin (2022) also consider overfitting by a fraction beyond the noise level and derive a lower bound for linear models.

**Benign overfitting in fixed dimension.** Only few works have established consistency results for interpolating models in fixed dimension. The first statistical guarantees for Nadaraya-Watson kernel

smoothing with singular kernels were given by Devroye et al. (1998). Optimal non-asymptotic results have only been established more recently. Belkin et al. (2019) show that Nadaraya-Watson kernel smoothing achieves minimax optimal convergence rates for $a \in (0, d/2)$ under smoothness assumptions on $f^*$, when using singular kernels such as truncated Hilbert kernels $K(u) = \|u\|_2^a \mathbb{1}_{\|u\| \leq 1}$, which do not induce RKHS that only contain weakly differentiable functions (as our results do). By thresholding the kernel they can adjust the amount of overfitting without affecting the generalization bound. To the best of our knowledge, rigorously proving or disproving analogous bounds for kernel ridge regression remains an open question. Arnould et al. (2023) show that median random forests are able to interpolate consistently in fixed dimension because of an averaging effect introduced through feature randomization. They conjecture consistent interpolation for Breiman random forests based on numerical experiments.

**Classification.** For binary classification tasks, benign overfitting is a more generic phenomenon than for regression tasks (Muthukumar et al., 2021, Shamir, 2022). Consistency has been shown under linear separability assumptions (Montanari et al., 2019, Chatterji and Long, 2021, Frei et al., 2022) and through complexity bounds with respect to reference classes like the 'Neural Tangent Random Feature' model (Cao and Gu, 2019, Chen et al., 2021). Most recently, Liang and Recht (2023) have shown that the 0-1-generalization error of minimum RKHS-norm interpolants $\hat{f}_0$ is upper bounded by $\frac{\|\hat{f}_0\|_{\mathcal{H}}^2}{n}$ and analogously that kernel ridge regression $\hat{f}_\rho$ generalizes as $\frac{\boldsymbol{y}^\top (k(\boldsymbol{X}, \boldsymbol{X}) + \rho \boldsymbol{I})^{-1} \boldsymbol{y}}{n}$, where the numerator upper bounds $\|\hat{f}_\rho\|_{\mathcal{H}}^2$. Their bounds imply consistency as long as the total variation distance between the class conditionals is sufficiently large and the regression function has bounded RKHS-norm, and their Lemma 7 shows that the upper bound is rate optimal. Under a noise condition on the regression function $f^*(\boldsymbol{x}) = \mathbb{E}[y|\boldsymbol{x}]$ for binary classification and bounded $\|f^*\|_{\mathcal{H}}$, our results together with Liang and Recht (2023) reiterate the distinction between benign overfitting for binary classification and inconsistent overfitting for least squares regression for a large class of distributions in kernel regression over Sobolev RKHS. Chapter 8 of Steinwart and Christmann (2008) discusses how the overlap of the two classes may influence learning rates under positive regularization. Using Nadaraya-Watson kernel smoothing, Wang and Scott (2022) offer the first consistency result for a simple interpolating ensemble method with data-independent base classifiers.

**Connection to neural networks.** It is known that neural networks can behave like kernel methods in certain infinite-width limits. For example, the function represented by a randomly initialized NN behaves like a Gaussian process with the NN Gaussian process (NNGP) kernel, which depends on details such as the activation function and depth of the NN (Neal, 1996, Lee et al., 2018, Matthews et al., 2018). Hence, Bayesian inference in infinitely wide NNs is GP regression, whose posterior predictive mean function is of the form $f_{\infty,\rho}$, where $\rho$ depends on the assumed noise variance. Moreover, gradient flow training of certain infinitely wide NNs is similar to gradient flow training with the so-called *neural tangent kernel* (NTK) (Jacot et al., 2018, Lee et al., 2019, Arora et al., 2019), and the correspondence can be made exact using small modifications to the NN to remove the stochastic effect of the random initial function (Arora et al., 2019, Zhang et al., 2020). In other words, certain infinitely wide NNs trained with gradient flow learn functions of the form $f_{t,0}$.

When considering the sphere $\Omega = \mathbb{S}^d$, the NTK and NNGP kernels of fully-connected NNs are dot-product kernels, i.e., $k(\boldsymbol{x}, \boldsymbol{x}') = \kappa(\langle \boldsymbol{x}, \boldsymbol{x}' \rangle)$ for some function $\kappa : [-1, 1] \to \mathbb{R}$. Moreover, from Bietti and Bach (2021) and Chen and Xu (2021) it follows that the RKHSs of typical NTK and NNGP kernels for the ReLU activation function are equivalent to the Sobolev spaces $H^{(d+1)/2}(\mathbb{S}^d)$ and $H^{(d+3)/2}(\mathbb{S}^d)$, respectively, cf Appendix B.4.

Regarding consistency, Ji et al. (2021) use the NTK correspondence to show that early-stopped wide NNs for classification are universally consistent under some assumptions. On the other hand, Holzmüller and Steinwart (2022) show that zero-initialized biases can prevent certain two-layer ReLU NNs from being universally consistent. Lai et al. (2023) show an inconsistency-type result for overfitting two-layer ReLU NNs with $d = 1$, but for fixed inputs $\boldsymbol{X}$. They also note that an earlier inconsistency result by Hu et al. (2021) relies on an unproven result. Li et al. (2023) show that consistency with polynomial convergence rates is impossible for minimum-norm interpolants of common kernels including ReLU NTKs. Mallinar et al. (2022) conjecture tempered overfitting and therefore inconsistency for interpolation with ReLU NTKs based on their semi-rigorous result and the results of Bietti and Bach (2021) and Chen and Xu (2021). Xu and Gu (2023) establish consistency of overfitting wide 2-layer neural networks beyond the NTK regime for binary classification in very

high dimension $d = \Omega(n^2)$ and for a quite restricted class of distributions (the mean difference $\mu$ of the class conditionals needs to fulfill $\mu = \Omega((d/n)^{1/4} \log^{1/4}(md/n))$ and $\mu = O((d/n)^{1/2})$).

## B  Kernels and Sobolev spaces on the sphere

### B.1  Background on Sobolev spaces

We say that two Hilbert spaces $\mathcal{H}_1, \mathcal{H}_2$ are equivalent, written as $\mathcal{H}_1 \cong \mathcal{H}_2$, if they are equal as sets and the corresponding norms $\| \cdot \|_{\mathcal{H}_1}$ and $\| \cdot \|_{\mathcal{H}_2}$ are equivalent.

Let $\Omega$ be an open set with $C^\infty$ boundary. In this paper, we will mainly consider $\ell_2$-balls for $\Omega$. There are multiple equivalent ways to define a (fractional) Sobolev space $H^s(\Omega)$, $s \in \mathbb{R}_{\geq 0}$, these are equivalent in the sense that the resulting Hilbert spaces will be equivalent. For example, $H^s(\Omega)$ can be defined through restrictions of functions from $H^s(\mathbb{R}^d)$, through interpolation spaces, or through Sobolev-Slobodetski norms (see e.g. Chapter 5 and 14 in Agranovich, 2015 and Chapters 7–10 in Lions and Magenes, 2012). Some requirements on $\Omega$ can be relaxed, for example to Lipschitz domains, by using more general extension operators (e.g. DeVore and Sharpley, 1993). Since our results are based on equivalent norms and not specific norms, we do not care which of these definitions is used. Further background on Sobolev spaces can be found in Adams and Fournier (2003), Wendland (2005) and Di Nezza et al. (2012).

### B.2  General kernel theory and notation

There is a one-to-one correspondence between kernel functions $k$ and the corresponding reproducing kernel Hilbert spaces (RKHS) $\mathcal{H}_k$. Mercer's theorem (Steinwart and Christmann, 2008, Theorem 4.49) states that for compact $\Omega$, continuous $k$ and a Borel probability measure $P_X$ on $\Omega$ whose support is $\Omega$, the integral operator $T_{k,P_X} : L_2(P_X) \to L_2(P_X)$ given by

$$T_{k,P_X} f(\boldsymbol{x}) = \int_\Omega f(\boldsymbol{x}') k(\boldsymbol{x}, \boldsymbol{x}') dP_X(\boldsymbol{x}'),$$

can be decomposed into an orthonormal basis $(e_i)_{i \in I}$ of $L_2(P_X)$ and corresponding eigenvalues $(\lambda_i)_{i \in I} \geq 0$, $\lambda_l \searrow 0$, such that

$$T_{k,P_X} f = \sum_{i \in I} \lambda_i \langle f, e_i \rangle e_i, \qquad f \in L_2(P_X).$$

We write $\lambda_i(T_{k,P_X}) := \lambda_i$. Moreover, $k(\boldsymbol{x}, \boldsymbol{x}') = \sum_{i \in I} \lambda_i e_i(\boldsymbol{x}) e_i(\boldsymbol{x}')$ converges absolutely and uniformly, and the RKHS is given by

$$\mathcal{H}_k = \left\{ \sum_{i \in I} a_i \sqrt{\lambda_i} e_i \;\middle|\; \sum_{i \in I} a_i^2 < \infty \right\}. \tag{B.1}$$

The corresponding inner product between $f = \sum_{i \in I} a_i \sqrt{\lambda_i} e_i \in \mathcal{H}$ and $g = \sum_{i \in I} b_i \sqrt{\lambda_i} e_i \in \mathcal{H}$ can then be written as

$$\langle f, g \rangle_\mathcal{H} = \sum_{i \in I} a_i b_i. \tag{B.2}$$

We use asymptotic notation $O, \Omega, \Theta$ for integers $n$ in the following way: We write

$$f(n) = O(g(n)) \Leftrightarrow \exists C > 0 \forall n : f(n) \leq Cg(n)$$
$$f(n) = \Omega(g(n)) \Leftrightarrow g(n) = O(f(n))$$
$$f(n) = \Theta(g(n)) \Leftrightarrow f(n) = O(g(n)) \text{ and } g(n) = O(f(n)) .$$

Above, we require that the inequality $f(n) \leq Cg(n)$ holds for all $n$ and not only for $n \geq n_0$. This implies that if $f(n) = \Omega(g(n))$, then $f$ must be nonzero whenever $g$ is nonzero. This is an important detail when arguing about equivalence of RKHSs, since it allows the following statement: If we have two kernels $k, \tilde{k}$ with Mercer representations

$$k(\boldsymbol{x}, \boldsymbol{x}') = \sum_{i \in I} \lambda_i e_i(\boldsymbol{x}) e_i(\boldsymbol{x}')$$

$$\tilde{k}(\boldsymbol{x}, \boldsymbol{x}') = \sum_{i \in I} \tilde{\lambda}_i e_i(\boldsymbol{x}) e_i(\boldsymbol{x}')$$

with identical eigenfunctions $e_i$ and eigenvalues satisfying $\lambda_i = \Theta(\tilde{\lambda}_i)$, then the associated RKHSs are equivalent by (B.1) and (B.2).

### B.3 Dot-product kernels on the sphere

A kernel of the form $k(\boldsymbol{x}, \boldsymbol{x}') = \kappa(\langle \boldsymbol{x}, \boldsymbol{x}' \rangle)$ for some function $\kappa$ is called *dot-product kernel*. Dot-product kernels are rotationally invariant. Especially, NTKs and NNGPs of fully-connected NNs restricted to the sphere $\mathbb{S}^d$ are dot-product kernels. Moreover, kernels like the Laplace, Matérn, and Gaussian kernels that only depend on the distance between their inputs are also dot-product kernels when restricted to the sphere $\mathbb{S}^d$. Therefore, in this section, we will assume that $k : \mathbb{S}^d \times \mathbb{S}^d \to \mathbb{R}$ is a dot-product kernel.

We can then leverage some convenient results from the theory of dot-product kernels on the sphere, which are summarized in more detail by Hubbert et al. (2023). Let $\{Y_{l,1}, \ldots, Y_{l,N_{l,d}}\}$ be a real orthonormal basis for the space of spherical harmonics of degree $l$ within $L_2(\mathbb{S}^d)$. Moreover, let $\omega_d$ be the surface area of $\mathbb{S}^d$, then the $\tilde{Y}_{l,i} := \sqrt{\omega_d} Y_{l,i}$ form a corresponding orthonormal basis w.r.t. the uniform distribution $\mathcal{U}(\mathbb{S}^d)$. Then, a Mercer representation of $k$ is given by

$$k(\boldsymbol{x}, \boldsymbol{x}') = \sum_{l=0}^{\infty} \mu_l \sum_{i=1}^{N_{l,d}} Y_{l,i}(\boldsymbol{x}) Y_{l,i}(\boldsymbol{x}') = \sum_{l=0}^{\infty} \tilde{\mu}_l \sum_{i=1}^{N_{l,d}} \tilde{Y}_{l,i}(\boldsymbol{x}) \tilde{Y}_{l,i}(\boldsymbol{x}') \,,$$

with $\tilde{\mu}_l = \mu_l / \omega_d$. Especially, the integral operator $T_{k, \mathcal{U}(\mathbb{S}^d)}$ for the uniform distribution $\mathcal{U}(\mathbb{S}^d)$ has eigenvalues $\tilde{\mu}_l$ with multiplicity $N_{l,d}$ and eigenfunctions $\tilde{Y}_{l,i}$. The RKHS of $k$ is then given by

$$\mathcal{H}_k = \left\{ \sum_{l=0}^{\infty} \sqrt{\mu_l} \sum_{i=1}^{N_{l,d}} a_{l,i} Y_{l,i} \;\middle|\; \sum_{l=0}^{\infty} \sum_{i=1}^{N_{l,d}} a_{l,i}^2 < \infty \right\} \,.$$

Since the index $l$ can be zero, we will denote decay asymptotics for $l$ in the form $\Theta((l+1)^{-q})$ and not $\Theta(l^{-q})$, cf. our definition of $\Theta$ notation in Appendix B.2.

**Lemma B.1** (Sobolev dot-product kernels on the sphere). *For a dot-product kernel $k$ on $\mathbb{S}^d$ as above, the RKHS $\mathcal{H}_k$ is equivalent to the Sobolev space $H^s(\mathbb{S}^d), s > d/2$, if and only if $\mu_l = \Theta((l+1)^{-2s})$. In this case, we have*

$$\lambda_i(T_{k, \mathcal{U}(\mathbb{S}^d)}) = \Theta(i^{-2s/d}) \,.$$

*Proof.* **Step 0: Equivalence.** If $\mu_l = \Theta((l+1)^{-2s})$, it is stated in Section 3 in Hubbert et al. (2023) that $\mathcal{H}_k \cong H^s(\mathbb{S}^d)$. On the other hand, if $\mu_l \neq \Theta((l+1)^{-2s})$, it is easy to see that $\mathcal{H}_k$ is not equivalent to the RKHS of a kernel with $\mu_l = \Theta((l+1)^{-2s})$. It remains to show $\lambda_i(T_{k, \mathcal{U}(\mathbb{S}^d)}) = \Theta(i^{-2s/d})$.

**Step 1: Ordering the eigenvalues.** Consider a permutation $\pi : \mathbb{N}_0 \to \mathbb{N}_0$ such that

$$\tilde{\mu}_{\pi(0)} \geq \tilde{\mu}_{\pi(1)} \geq \ldots$$

We can then define the partial sums

$$S_l := \sum_{i=0}^{l} N_{\pi(i),d} \,.$$

For $S_{l-1} < i \leq S_l$, we then have $\lambda_i(T_{k, \mathcal{U}(\mathbb{S}^d)}) = \tilde{\mu}_{\pi(l)}$.

**Step 2: Show $\pi(i) = \Theta(i)$.** Let $c, C > 0$ such that $c(l+1)^{-2s} \leq \tilde{\mu}_l \leq C(l+1)^{-2s}$ for all $l \in \mathbb{N}_0$. For indices $i, j \in \mathbb{N}_0$, we have the implications

$$i > j \Rightarrow c(\pi(i) + 1)^{-2s} \leq \tilde{\mu}_{\pi(i)} \leq \tilde{\mu}_{\pi(j)} \leq C(\pi(j) + 1)^{-2s}$$

$$\Rightarrow \pi(i) + 1 \geq \left(\frac{c}{C}\right)^{1/(2s)} (\pi(j) + 1) \,.$$

Therefore, for $i \geq 1$ and $j \geq 0$,

$$\pi(i) + 1 \geq \left(\frac{c}{C}\right)^{1/(2s)} \max_{i' < i}(\pi(i') + 1) \geq \left(\frac{c}{C}\right)^{1/(2s)} ((i-1) + 1) \geq \Omega(i+1) \,,$$

$$\pi(j) + 1 \leq \left(\frac{C}{c}\right)^{1/(2s)} \min_{j' > j}(\pi(j') + 1) \leq \left(\frac{C}{c}\right)^{1/(2s)} ((j+1) + 1) \leq O(j+1) \,.$$

We can thus conclude that $\pi(i) + 1 = \Theta(i+1)$.

**Step 3: Individual Eigenvalue decay.** As explained in Section 2.1 in Hubbert et al. (2023), we have $N_{l,d} = \Theta((l+1)^{d-1})$. Therefore,

$$S_l = \sum_{i=0}^{l} \Theta((\pi(i) + 1)^{d-1}) = \sum_{i=0}^{l} \Theta((i+1)^{d-1}) = \Theta((l+1)^d) \,.$$

Now, let $i \geq 1$ and let $l \in \mathbb{N}_0$ such that $S_{l-1} < i \leq S_l$. We have $i \geq \Omega(l^d)$, and $i \leq O((l+1)^d)$, which implies $i = \Theta((l+1)^d)$ since $i \geq 1$. Therefore,

$$\lambda_i = \tilde{\mu}_{\pi(l)} = \Theta((\pi(l) + 1)^{-2s}) = \Theta((l+1)^{-2s}) = \Theta\left(i^{-2s/d}\right) \,. \qquad \square$$

## B.4 Neural kernels

Several NTK and NNGP kernels have RKHSs that are equivalent to Sobolev spaces on $\mathbb{S}^d$. In the following cases, we can deduct this from known results:

- Consider fully-connected NNs with $L \geq 3$ layers without biases and the activation function $\varphi(x) = \max\{0, x\}^m$, $m \in \mathbb{N}_0$. Especially, the case $m = 1$ corresponds to the ReLU activation. Vakili et al. (2021) generalize the result by Bietti and Bach (2021) from $m = 1$ to $m \geq 1$, showing that the NTK-RKHS is equivalent to $H^s(\mathbb{S}^d)$ for $s = (d + 2m - 1)/2$ and the NNGP-RKHS is equivalent to $H^s(\mathbb{S}^d)$ for $s = (d + 2m + 1)/2$. For $m = 0$, Bietti and Bach (2021) essentially show that the NNGP-RKHS is equivalent to $H^s(\mathbb{S}^d)$ for $s = (d + 2^{2-L})/2$. However, all of the aforementioned result have the problem that the main theorem by Bietti and Bach (2021) allows for the possibility that finitely many $\mu_l$ are zero, which can change the RKHS. Using our Lemma B.2 below, it follows that all $\mu_l$ are in fact nonzero for NNGPs and NTKs since they are kernels in every dimension $d$ using the same function $\kappa$ independent of the dimension. Hence, the equivalences to Sobolev spaces stated before are correct.
- Chen and Xu (2021) prove that the RKHS of the NTK corresponding to fully-connected ReLU NNs with zero-initialized biases and $L \geq 2$ (as opposed to no biases and $L \geq 3$ above) layers is equivalent to the RKHS of the Laplace kernel on the sphere. Since the Laplace kernel is a Matérn kernel of order $\nu = 1/2$ (see e.g. Section 4.2 in Rasmussen and Williams (2005)), we can use Proposition 5.2 of Hubbert et al. (2023) to obtain equivalence to $H^s(\mathbb{S}^d)$ with $s = (d+1)/2$. Alternatively, we can obtain the RKHS of the Laplace kernel from Bietti and Bach (2021) combined with Lemma B.2.

Bietti and Bach (2021) also show that under an integrability condition on the derivatives, $C^\infty$ activations induce NTK and NNGP kernels whose RKHSs are smaller than every Sobolev space.

**Lemma B.2** (Guaranteeing non-zero eigenvalues)**.** *Let* $\kappa : [-1, 1] \to \mathbb{R}$, *let* $d \geq 1$, *and let*

$$k_d : \mathbb{S}^d \times \mathbb{S}^d, k_d(\boldsymbol{x}, \boldsymbol{x}') := \kappa(\langle \boldsymbol{x}, \boldsymbol{x}' \rangle)$$
$$k_{d+2} : \mathbb{S}^{d+2} \times \mathbb{S}^{d+2}, k_{d+2}(\boldsymbol{x}, \boldsymbol{x}') := \kappa(\langle \boldsymbol{x}, \boldsymbol{x}' \rangle) \,.$$

*Suppose that* $k_{d+2}$ *is a kernel. Then,* $k_d$ *is a kernel. Moreover, if the corresponding eigenvalues* $\mu_l$ *of* $k_d$ *satisfy* $\mu_l > 0$ *for infinitely many* $l$, *then they satisfy* $\mu_l > 0$ *for all* $l \in \mathbb{N}_0$.

*Proof.* The fact that $k_d$ is a kernel follows directly from the inclusion $\Phi_{d+2} \subseteq \Phi_d$ mentioned in Gneiting (2013). For $D \in \{d, d+2\}$, let $\mu_{l,d}$ be the sequence of eigenvalues $\mu_l$ associated with $k_D$. Then, as mentioned for example by Hubbert et al. (2023), the Schoenberg coefficients $b_{l,d}$ satisfy

$$b_{l,d} = \frac{\Gamma\left(\frac{d+1}{2}\right) N_{m,d} \mu_{l,d}}{2\pi^{(d+1)/2}} \,.$$

Especially, the Schoenberg coefficients $b_{l,d}$ have the same sign as the eigenvalues $\mu_{l,d}$. We use

$$0 \leq b_{l,d+2} = \begin{cases} b_{l,d} - \frac{1}{2}b_{l+2,d} & , l = 0 \text{ and } d = 1 \\ \frac{1}{2}(l+1)(b_{l,d} - b_{l+2,d}) & , l \geq 1 \text{ and } d = 1 \\ \frac{(l+d-1)(l+d)}{d(2l+d-1)}b_{l,d} - \frac{(l+1)(l+2)}{d(2l+d+3)}b_{l+2,d} & , d \geq 2 \,, \end{cases}$$

where the inequality follows from the fact that $k_{d+2}$ is a kernel and the equality is the statement of Corollary 3 by Gneiting (2013). In any of the three cases, $b_{l+2,d} > 0$ implies $b_{l,d} > 0$. Hence, if $b_{l,d} > 0$ for infinitely many $l$, then $b_{l,d} > 0$ for all $l$, which implies $\mu_{l,d} > 0$ for all $l$. $\qquad\square$

# C   Gradient flow and gradient descent with kernels

## C.1   Derivation of gradient flow and gradient descent

Here, we derive expressions for gradient flow and gradient descent in the RKHS for the regularized loss

$$L(f) := \frac{1}{n}\sum_{i=1}^{n}(y_i - f(\boldsymbol{x}_i))^2 + \rho\|f\|_{\mathcal{H}_k}^2 = \frac{1}{n}\sum_{i=1}^{n}(y_i - \langle k(\boldsymbol{x}_i, \cdot), f\rangle_{\mathcal{H}_k})^2 + \rho\langle f, f\rangle_{\mathcal{H}_k}^2 \,.$$

Note that we will take derivatives in the RKHS with respect to $f$, which is different from taking derivatives w.r.t. the coefficients $\boldsymbol{c}$ in a model $f(\boldsymbol{x}) = \boldsymbol{c}^\top k(\boldsymbol{X}, \boldsymbol{x})$.

In the RKHS-Norm, the Fréchet derivative of $L$ is

$$\frac{\partial L(f)}{f} = \frac{2}{n}\sum_{i=1}^{n}(f(\boldsymbol{x}_i) - y_i)\langle k(\boldsymbol{x}_i, \cdot), \cdot\rangle_{\mathcal{H}_k} + 2\rho\langle f, \cdot\rangle_{\mathcal{H}_k} \,,$$

which is represented in $\mathcal{H}_k$ by

$$L'(f) = \frac{2}{n}\sum_{i=1}^{n}(f(\boldsymbol{x}_i) - y_i)k(\boldsymbol{x}_i, \cdot) + 2\rho f \,.$$

Now assume that $f = \sum_{i=1}^{n}a_i k(\boldsymbol{x}_i, \cdot) = \boldsymbol{a}^\top k(\boldsymbol{X}, \cdot)$. Then,

$$\begin{aligned} L'(f) &= \frac{2}{n}\sum_{i=1}^{n}(\boldsymbol{a}^\top k(\boldsymbol{X}, \boldsymbol{x}_i) - y_i)k(\boldsymbol{x}_i, \cdot) + 2\rho\boldsymbol{a}^\top k(\boldsymbol{X}, \cdot) \\ &= \frac{2}{n}\left(\boldsymbol{a}^\top k(\boldsymbol{X}, \boldsymbol{X})k(\boldsymbol{X}, \cdot) - \boldsymbol{y}^\top k(\boldsymbol{X}, \cdot) + \rho n\boldsymbol{a}^\top k(\boldsymbol{X}, \cdot)\right) \\ &= \frac{2}{n}\left((k(\boldsymbol{X}, \boldsymbol{X}) + \rho n\boldsymbol{I}_n)\boldsymbol{a} - \boldsymbol{y}\right)^\top k(\boldsymbol{X}, \cdot) \,. \end{aligned}$$

Especially, under gradient flow of $f$, the coefficients $\boldsymbol{a}$ follow the dynamics

$$\dot{\boldsymbol{a}}(t) = \frac{2}{n}\left(\boldsymbol{y} - (k(\boldsymbol{X}, \boldsymbol{X}) + \rho n\boldsymbol{I}_n)\boldsymbol{a}(t)\right) \,,$$

which is solved for $\boldsymbol{a}(0) = 0$ by

$$\boldsymbol{a}(t) = \left(\boldsymbol{I}_n - e^{-\frac{2}{n}t(k(\boldsymbol{X}, \boldsymbol{X}) + \rho n\boldsymbol{I}_n)}\right)(k(\boldsymbol{X}, \boldsymbol{X}) + \rho n\boldsymbol{I}_n)^{-1}\boldsymbol{y} \,,$$

which is the closed form expression (1) of $f_{t,\rho}$.

For gradient descent, assuming that $f_{t,\rho}^{\mathrm{GD}} = \boldsymbol{c}_{t,\rho}^\top k(\boldsymbol{X}, \cdot)$, we have

$$\begin{aligned} f_{t+1,\rho}^{\mathrm{GD}} &= f_{t,\rho}^{\mathrm{GD}} - \eta_t L'(f_{t,\rho}^{\mathrm{GD}}) = \boldsymbol{c}_{t,\rho}^\top k(\boldsymbol{X}, \cdot) - \eta_t\frac{2}{n}\left((k(\boldsymbol{X}, \boldsymbol{X}) + \rho n\boldsymbol{I}_n)\boldsymbol{c}_{t,\rho} - \boldsymbol{y}\right)^\top k(\boldsymbol{X}, \cdot) \\ &= \left(\boldsymbol{c}_{t,\rho} + \eta_t\frac{2}{n}\left(\boldsymbol{y} - (k(\boldsymbol{X}, \boldsymbol{X}) + \rho n\boldsymbol{I}_n)\boldsymbol{c}_{t,\rho}\right)\right)^\top k(\boldsymbol{X}, \cdot) \end{aligned}$$

If $f_{0,\rho}^{\mathrm{GD}} \equiv 0$, the coefficients evolve as $\boldsymbol{c}_0 = \boldsymbol{0}$ and

$$\boldsymbol{c}_{t+1,\rho} = \boldsymbol{c}_{t,\rho} + \eta_t\frac{2}{n}\left(\boldsymbol{y} - (k(\boldsymbol{X}, \boldsymbol{X}) + \rho n\boldsymbol{I}_n)\boldsymbol{c}_{t,\rho}\right) \,.$$

For an analysis of gradient descent for kernel regression with $\rho = 0$, we refer to, e.g., Yao et al. (2007).

## C.2  Gradient flow and gradient descent initialized at $0$ have monotonically growing $\mathcal{H}$-norm

In the following proposition we show that under gradient flow and gradient descent with sufficiently small learning rates initialized at 0, the RKHS norm grows monotonically with time $t$. This immediately implies that Assumption (N) with $C_{\mathrm{norm}} = 1$ holds for all estimators $f_{t,\rho}$ from (1).

**Proposition C.1.**

  (i) *For any $t \in [0, \infty]$ and any $\rho \geq 0$, $f_{t,\rho}$ from (1) fulfills Assumption (N) with $C_{\mathrm{norm}} = 1$.*
  (ii) *For any $t \in \mathbb{N}_0 \cup \{\infty\}$ and any $\rho \geq 0$, with sufficiently small fixed learning rate $0 \leq \eta \leq \frac{1}{2(\rho + \lambda_{\max}(k(\boldsymbol{X}, \boldsymbol{X}))/n)}$, $f_{t,\rho}^{\mathrm{GD}}$ fulfills Assumption (N) with $C_{\mathrm{norm}} = 1$.*

*Proof.*  **Proof of (i):**

We write $f_{t,\rho}(\boldsymbol{x}) = k(\boldsymbol{x}, \boldsymbol{X})\boldsymbol{c}_{t,\rho}$, where $\boldsymbol{c}_{t,\rho} := A_{t,\rho}(\boldsymbol{X})\boldsymbol{y}$. We now show that the RKHS-norm of $f_{t,\rho}$ grows monotonically in $t$, by using the eigendecomposition $k(\boldsymbol{X}, \boldsymbol{X}) = \boldsymbol{U}\Lambda\boldsymbol{U}^{\top}$, where $\Lambda = \mathrm{diag}(\lambda_1, \ldots, \lambda_n) \in \mathbb{R}^{n \times n}$ is diagonal and $\boldsymbol{U} \in \mathbb{R}^{n \times n}$ is orthonormal, and writing $\tilde{\boldsymbol{y}} := \boldsymbol{U}^{\top}\boldsymbol{y}$. Then it holds that

$$\|f_{t,\rho}\|_{\mathcal{H}}^2 = (\boldsymbol{c}_{t,\rho})^{\top} k(\boldsymbol{X}, \boldsymbol{X})\boldsymbol{c}_{t,\rho} = \tilde{\boldsymbol{y}}^{\top}(\Lambda + \rho n \boldsymbol{I}_n)^{-1}\left(\boldsymbol{I}_n - \exp\left(-\frac{2t}{n}(\Lambda + \rho n \boldsymbol{I}_n)\right)\right)\Lambda \cdot$$

$$\left(\boldsymbol{I}_n - \exp\left(-\frac{2t}{n}(\Lambda + \rho n \boldsymbol{I}_n)\right)\right)(\Lambda + \rho n \boldsymbol{I}_n)^{-1}\tilde{\boldsymbol{y}}$$

$$= \sum_{k=1,\lambda_k+\rho n>0}^{n} \tilde{y}_k^2 \underbrace{\frac{\lambda_k}{(\lambda_k + \rho n)^2}}_{\leq 1/\lambda_k} \underbrace{\left(1 - \exp\left(-\frac{2t}{n}(\lambda_k + \rho n)\right)\right)}_{\leq 1}$$

$$\leq \sum_{k=1,\lambda_k>0}^{n} \tilde{y}_k^2 \frac{1}{\lambda_k} = \|f_{\infty,0}\|_{\mathcal{H}}^2.$$

**Proof of (ii):**

Expanding the iteration in the definition of $\boldsymbol{c}_{t,\rho}$ yields

$$\boldsymbol{c}_{t+1,\rho} = \sum_{i=0}^{t} \prod_{j=0}^{t-i-1}\left(\boldsymbol{I} - \frac{2\eta_{t-j}}{n}(k(\boldsymbol{X}, \boldsymbol{X}) + \rho n \boldsymbol{I})\right)\frac{2\eta_i}{n}\boldsymbol{y}.$$

We again use the eigendecomposition $k(\boldsymbol{X}, \boldsymbol{X}) = \boldsymbol{U}\Lambda\boldsymbol{U}^{\top}$, where $\Lambda = \mathrm{diag}(\lambda_1, \ldots, \lambda_n) \in \mathbb{R}^{n \times n}$ is diagonal and $\boldsymbol{U} \in \mathbb{R}^{n \times n}$ is orthonormal, and write $\tilde{\boldsymbol{y}} := \boldsymbol{U}^{\top}\boldsymbol{y}$. Then, using sufficiently small learning rates $0 \leq \eta_t \leq \frac{1}{2(\rho + \lambda_{\max}(k(\boldsymbol{X}, \boldsymbol{X}))/n)}$ in all time steps $t \in \mathbb{N}$, it holds that

$$\|f_{t,\rho}^{\mathrm{GD}}\|_{\mathcal{H}}^2$$

$$= (\boldsymbol{c}_{t,\rho})^{\top} k(\boldsymbol{X}, \boldsymbol{X})\boldsymbol{c}_{t,\rho}$$

$$= \tilde{\boldsymbol{y}}^{\top}\left(\sum_{i=0}^{t}\frac{2\eta_i}{n}\prod_{j=0}^{t-i-1}((1 - 2\eta_{t-j}\rho)\boldsymbol{I} - \frac{2\eta_{t-j}}{n}\Lambda)\right)\Lambda\left(\sum_{i=0}^{t}\frac{2\eta_i}{n}\prod_{j=0}^{t-i-1}((1 - 2\eta_{t-j}\rho)\boldsymbol{I} - \frac{2\eta_{t-j}}{n}\Lambda)\right)\tilde{\boldsymbol{y}}$$

$$= \sum_{k=1}^{n} \underbrace{\tilde{y}_k^2 \lambda_k}_{\geq 0}\left(\sum_{i=0}^{t}\frac{2\eta_i}{n}\prod_{j=0}^{t-i-1}\underbrace{(1 - 2\eta_{t-j}(\rho + \lambda_k/n))}_{\in[0,1]}\right)^2. \tag{C.1}$$

The last display shows that $\|f_{t,\rho}^{\mathrm{GD}}\|_{\mathcal{H}}^2$ grows monotonically in $t$, strictly monotonically if $\eta_t \in (0, \frac{1}{2(\rho + \lambda_{\max}(k(\boldsymbol{X}, \boldsymbol{X}))/n)})$ holds for all $t$. It also shows that if $\rho' \geq \rho$ then $\|f_{t,\rho'}^{\mathrm{GD}}\|_{\mathcal{H}} \leq \|f_{t,\rho}^{\mathrm{GD}}\|_{\mathcal{H}}$ for any $t \in \mathbb{N} \cup \{\infty\}$. To see that $\|f_{t,\rho}^{\mathrm{GD}}\|_{\mathcal{H}}^2 \leq \|f_{\infty,0}\|_{\mathcal{H}}^2$ for all $t \in \mathbb{N} \cup \{\infty\}$ and all $\rho \geq 0$, observe that with fixed learning rates $\eta_t = \eta \in (0, \frac{1}{2(\rho + \lambda_{\max}(k(\boldsymbol{X}, \boldsymbol{X}))/n)}) \subseteq (0, \frac{1}{2\lambda_{\max}(k(\boldsymbol{X}, \boldsymbol{X}))/n})$, for all $t \in \mathbb{N} \cup \{\infty\}$ it holds that

$$\sum_{i=0}^{t}\frac{2\eta_i}{n}\prod_{j=0}^{t-i-1}(1 - 2\eta_{t-j}\lambda_k/n) = \frac{2\eta}{n}\sum_{i=0}^{t}(1 - 2\eta\lambda_k/n)^{t-i}$$

$$= \frac{2\eta}{n} \sum_{i=0}^{t} (1 - 2\eta\lambda_k/n)^i = \frac{2\eta}{n} \frac{1 - (1 - 2\eta\lambda_k/n)^{t+1}}{2\eta\lambda_k/n} \leq \frac{1}{\lambda_k}.$$

Since it suffices to consider the case $\rho \to 0$, using the above derivation in (C.1) yields $\|f_{t,\rho}^{\mathrm{GD}}\|_{\mathcal{H}}^2 \leq \|f_{\infty,0}\|_{\mathcal{H}}^2$ for all $t \in \mathbb{N}$, which concludes the proof. $\qquad\square$

# D  Proof of Theorem 1

Our goal in this section is to prove Theorem D.1, which can be seen as a generalization of Theorem 1 to varying bandwidths. To be able to speak of bandwidths, we need to consider translation-invariant kernels. Although Theorem 1 is formulated for general kernels with Sobolev RKHS, it follows from Theorem D.1 since we can always find, for a fixed bandwidth, a translation-invariant kernel with equivalent RKHS, such that only the constant $C_{\mathrm{norm}}$ changes in the theorem statement.

To generate the RKHS $H^s$, Buchholz (2022) uses the translation-invariant kernel $k^B(\boldsymbol{x}, \boldsymbol{y}) = u^B(\boldsymbol{x} - \boldsymbol{y})$ defined via its Fourier transform $\hat{u}^B(\boldsymbol{\xi}) = (1 + |\boldsymbol{\xi}|^2)^{-s}$. Adapting the bandwidth, the kernel is then normalized in the usual $L_1$-sense,

$$k_\gamma^B(\boldsymbol{x}, \boldsymbol{y}) = \gamma^{-d} u^B((\boldsymbol{x} - \boldsymbol{y})/\gamma). \tag{D.1}$$

**Theorem D.1 (Inconsistency of overfitting estimators).** *Let assumptions (D1) and (K) hold. Let $c_{\mathrm{fit}} \in (0, 1]$ and $C_{\mathrm{norm}} > 0$. Then, there exist $c > 0$ and $n_0 \in \mathbb{N}$ such that the following holds for all $n \geq n_0$ with probability $1 - O(1/n)$ over the draw of the data set $D$ with $n$ samples: For every function $f \in \mathcal{H}_k$ with*

*(O)* $\frac{1}{n} \sum_{i=1}^{n} (f(x_i) - y_i)^2 \leq (1 - c_{\mathrm{fit}}) \cdot \sigma^2$ *(training error below Bayes risk) and*
*(N)* $\|f\|_{\mathcal{H}_k} \leq C_{\mathrm{norm}} \|f_{\infty,0}\|_{\mathcal{H}_k}$ *(norm comparable to minimum-norm interpolant (1)),*

*the excess risk satisfies*

$$R_P(f) - R_P(f^*) \geq c > 0. \tag{D.2}$$

*If $k_\gamma$ denotes a $L_1$-normalized translation-invariant kernel with bandwidth $\gamma > 0$, i.e. there exists a $q : \mathbb{R}^d \to \mathbb{R}$ such that $k_\gamma(x, y) = \gamma^{-d} q(\frac{x-y}{\gamma})$, then inequality (D.2) holds with $c$ independent of the sequence of bandwidths $(\gamma_n)_{n \in \mathbb{N}} \subseteq (0, 1)$, as long as $f_D$ fulfills (N) for the sequence $(\mathcal{H}_{\gamma_n})_{n \in \mathbb{N}}$ with constant $C_{\mathrm{norm}} > 0$.*

*Proof.* By assumption, the RKHS norm $\|\cdot\|_{\mathcal{H}_k}$ induced by the kernel $k$ (or $k_\gamma$ if we allow bandwidth adaptation) is equivalent to the RKHS norm $\|\cdot\|_{\mathcal{H}_\gamma}$ induced by a kernel of the form (D.1) with an arbitrary but fixed choice of bandwidth $\gamma \in (0, 1)$, which means that there exists a constant $C_\gamma > 0$ such that $\frac{1}{C_\gamma} \|f\|_{\mathcal{H}_\gamma} \leq \|f\|_{\mathcal{H}_k} \leq C_\gamma \|f\|_{\mathcal{H}_\gamma}$ for all $f \in \mathcal{H}_k$. Hence the minimum-norm interpolant $g_{D,\gamma}$ in $\mathcal{H}_\gamma$ satisfies

$$\|f_D\|_{\mathcal{H}_\gamma} \leq C_\gamma \|f_D\|_{\mathcal{H}_k} \leq C_\gamma C_{\mathrm{norm}} \|g_D\|_{\mathcal{H}_k} \leq C_\gamma C_{\mathrm{norm}} \|g_{D,\gamma}\|_{\mathcal{H}_k} \leq C_\gamma^2 C_{\mathrm{norm}} \|g_{D,\gamma}\|_{\mathcal{H}_\gamma},$$

where $\|g_D\|_{\mathcal{H}_k} \leq \|g_{D,\gamma}\|_{\mathcal{H}_k}$ because, $g_D$ is the minimum-norm interpolant in $\mathcal{H}_k$.

Now consider the RKHS norm $\|\cdot\|_{\tilde{\mathcal{H}}_\gamma}$ of a translation-invariant kernel $k_\gamma$. Then the functions $\{h_p(x) = e^{ip \cdot x}\}_{p \in \mathbb{R}^d}$ are eigenfunctions of the kernel's integral operator, so that the RKHS norm can be written as (Rakhlin and Zhai, 2019)

$$\|f\|_{\tilde{\mathcal{H}}_\gamma}^2 = \frac{1}{(2\pi)^d} \int_{\mathbb{R}^d} \frac{|\hat{f}(\omega)|^2}{\hat{q}(\omega)} d\omega,$$

where $\hat{f}$ denotes the Fourier transform of $f$.

By assumption we know that there exists a $C_{\gamma_0} > 0$ such that $\frac{1}{C_{\gamma_0}} \|f\|_{\mathcal{H}_{\gamma_0}} \leq \|f\|_{\tilde{\mathcal{H}}_{\gamma_0}} \leq C_{\gamma_0} \|f\|_{\mathcal{H}_{\gamma_0}}$ holds for some fixed bandwidth $\gamma_0 > 0$, then substituting by $\tilde{\omega} = \frac{\gamma}{\gamma_0} \omega$ yields

$$\|f\|_{\tilde{\mathcal{H}}_\gamma} = \frac{1}{(2\pi)^d} \int_{R^d} \frac{|\hat{f}(\omega)|^2}{\hat{q}_1(\gamma\omega)} d\omega = \frac{1}{(2\pi)^d} \int_{R^d} \frac{|\hat{f}(\frac{\gamma_0}{\gamma}\tilde{\omega})|^2}{\hat{q}_1(\gamma_0\tilde{\omega})} \left(\frac{\gamma_0}{\gamma}\right)^d d\tilde{\omega} = \|\tilde{f}\|_{\tilde{\mathcal{H}}_{\gamma_0}}$$

$$\leq C_{\gamma_0}\|\tilde{f}\|_{\mathcal{H}_{\gamma_0}} = \frac{C_{\gamma_0}}{(2\pi)^d}\int_{R^d}\frac{|\hat{f}(\frac{\gamma_0}{\gamma}\tilde{\omega})|^2}{\hat{q}_2(\gamma_0\tilde{\omega})}\left(\frac{\gamma_0}{\gamma}\right)^d d\tilde{\omega} = \frac{C_{\gamma_0}}{(2\pi)^d}\int_{R^d}\frac{|\hat{f}(\omega)|^2}{\hat{q}_2(\gamma\omega)}d\omega = C_{\gamma_0}\|f\|_{\mathcal{H}_\gamma}.$$

In the same way we get $\frac{1}{C_{\gamma_0}}\|f\|_{\mathcal{H}_\gamma} \leq \|f\|_{\tilde{\mathcal{H}}_\gamma}$ for arbitrary $\gamma \in (0,1)$. This shows that the constant $C_{\gamma_0}$, that quantifies the equivalence between $\|\cdot\|_{\mathcal{H}_\gamma}$ and $\|\cdot\|_{\tilde{\mathcal{H}}_\gamma}$ does not depend on the bandwidth $\gamma$. Finally Proposition D.4, Proposition D.2 and Remark D.3 together yield the result. $\qquad\square$

The following proposition generalizes the inconsistency result for large bandwidths, Proposition 4 in Buchholz (2022), beyond interpolating estimators to estimators that overfit at least an arbitrary constant fraction beyond the Bayes risk and whose RKHS norm is at most a constant factor larger than the RKHS norm of the minimum-norm interpolant. Compared to Rakhlin and Zhai (2019), Buchholz gets a statement in probability over the draw of a training set $D$ and less restrictive assumptions on the domain $\Omega$ and dimension $d$.

**Proposition D.2 (Inconsistency for large bandwidths).** *Let $c_{\text{fit}} \in (0,1]$ and $C_{\text{norm}} > 0$. Let the data set $D = \{(\boldsymbol{x}_1, y_1), \ldots, (\boldsymbol{x}_n, y_n)\}$ be drawn i.i.d. from a distribution $P$ that fulfills Assumption (D1), let $g_{D,\gamma}$ be the minimum-norm interpolant in $\mathcal{H} := \mathcal{H}_\gamma$ with respect to the kernel (D.1) for a bandwidth $\gamma > 0$. Then, for every $A > 0$, there exist $c > 0$ and $n_0 \in \mathbb{N}$ such that the following holds for all $n \geq n_0$ with probability $1 - O(1/n)$ over the draw of the data set $D$ with $n$ samples:*

*For every function $f \in \mathcal{H}$ that fulfills Assumption (O) with $c_{\text{fit}}$ and Assumption (N) with $C_{\text{norm}}$ the excess risk satisfies*

$$\mathbb{E}_{\boldsymbol{x}}(f(\boldsymbol{x}) - f^*(\boldsymbol{x}))^2 \geq c > 0,$$

*where $c$ depends neither on $n$ nor on $1 > \gamma > An^{-1/d} > 0$.*

**Remark D.3.** Proposition D.2 holds for any kernel that fulfills Assumption (K). The reason is that any kernel $k$ that fulfills assumption (K) and the kernel defined in (D.1) have the same RKHS and equivalent norms. Therefore every function $f \in \mathcal{H}_k = \mathcal{H}_\gamma$ (equality as sets) that fulfills Assumptions (O) and (N) for the kernel $k$ also fulfills Assumptions (O) and (N) with an adapted constant $C_{\text{norm}}$ for the kernel (D.1). $\qquad\blacktriangleleft$

*Proof.* **Step 1: Generalizing the procedure in Buchholz (2022).**

We write $[n] = \{1, \ldots, n\}$ and follow the proof of Proposition 4 in Buchholz (2022). Define $u(\boldsymbol{x}) = f(\boldsymbol{x}) - f^*(\boldsymbol{x})$. We need to show that with probability at least $1 - O(n^{-1})$ over the draw of $D$ it holds that $\|u\|_{L^2(P_X)} \geq c > 0$, where $c$ depends neither on $n$ nor on $\gamma$.

For this purpose we show that with probability at least $1 - 3n^{-1}$ over the draw of $D$ there exist a constants $c'', \kappa'' > 0$ depending only on $c_{\text{fit}}$ and a subset $\mathcal{P}'' \subseteq [n]$ with $|\mathcal{P}''| \geq \lfloor \kappa'' \cdot n \rfloor$ such that

$$|f(\boldsymbol{x}_i) - f^*(\boldsymbol{x}_i)| \geq c'' > 0 \text{ holds for all } i \in \mathcal{P}''. \tag{D.3}$$

Then via Lemma 7 in Buchholz (2022) as well as Lemma D.7 we can choose a large subset $\mathcal{P}''' \subseteq [n]$ of the training point indices with $|\mathcal{P}'''| \geq n - |\mathcal{P}''|/2$, such that the $\boldsymbol{x}_i$ for $i \in \mathcal{P}'''$ are well-separated in the sense that $\min_{\{i,j \in \mathcal{P}''', i \neq j\}}\|\boldsymbol{x}_i - \boldsymbol{x}_j\| \geq d_{min}$ with $d_{min} := c'''n^{-1/d}$, where $c'''$ depends on $c_{\text{fit}}, d$, the upper bound on the Lebesgue density $C_u$ and on the smoothness of the RKHS $s$. Then the intersection $\mathcal{P}'' \cap \mathcal{P}'''$ contains at least $\frac{|\mathcal{P}''|}{2}$ points. Now we can replace $\mathcal{P}'$ in the proof of Proposition 4 for $s \in \mathbb{N}$ in Buchholz (2022) by the intersection $\mathcal{P}'' \cap \mathcal{P}'''$. The rest of the proof applies without modification, where (42) holds by our assumption $\|f\|_{\mathcal{H}} \leq C_{\mathcal{H}}\|g_D\|_{\mathcal{H}}$. Our modifications do not affect Buchholz' arguments for the extension to $s \notin \mathbb{N}$.

**Step 2: The existence of $\mathcal{P}''$.**

Given a choice of $\kappa'', c'' > 0$, consider the event (over the draw of $D$)

$$\begin{aligned}
E &:= \{\nexists \, \mathcal{P}'' \subseteq [n] \text{ with } |\mathcal{P}''| \geq \lfloor \kappa'' \cdot n \rfloor \text{ that fulfills (D.3)}\} \\
&= \{\exists \, \tilde{\mathcal{P}} \subseteq [n] \text{ with } |\tilde{\mathcal{P}}| \geq \lceil (1 - \kappa'')n \rceil \text{ such that } |f^*(\boldsymbol{x}_i) - f(\boldsymbol{x}_i)| < c'' \quad \forall i \in \tilde{\mathcal{P}}\}.
\end{aligned}$$

With the proper choices of $c''$ and $\kappa''$ independent of $n$ and $f$, we will show $P(E) \leq 3n^{-1}$. We will find a small $c'' > 0$ such that if $f^*$ and $f$ are closer than $c''$ on too many training points $\tilde{\mathcal{P}}$

and $f$ overfits by at least the fraction $c_{\text{fit}}$, the noise variables $\varepsilon_i$ on the complement $\tilde{\mathcal{P}}^c$ would have to be unreasonably large, contradicting the event $E_{6i}$ defined below, and implying (D.3) with high probability. We will use the notation $\|\boldsymbol{f}\|_{\mathcal{P}}^2 := \sum_{i \in \mathcal{P}} f(\boldsymbol{x}_i)^2$ and $\|\boldsymbol{y}\|_{\mathcal{P}}^2 := \sum_{i \in \mathcal{P}} y_i^2$.

**Step 2b: Noise bounds.**

Lemma D.6 $(i)$ states that there exists a $\kappa'' > 0$ small enough such that the event (over the draw of $D$)

$$E_{6i} := \{\forall \mathcal{P}_1 \subseteq [n] \text{ with } |\mathcal{P}_1| \leq \lfloor \kappa'' \cdot n \rfloor \text{ it holds that } \frac{1}{n}\|\boldsymbol{f}^* - \boldsymbol{y}\|_{\mathcal{P}_1}^2 = \frac{1}{n}\sum_{i \in \mathcal{P}_1} \varepsilon_i^2 < \frac{c_{\text{fit}}}{4}\sigma^2\},$$

fulfills, for $n$ large enough, $P(E_{6i}) \geq 1 - n^{-1}$.

Lemma D.6 $(ii)$ implies that there exists a $c_{\text{lower}} > 0$ such that the event (over the draw of $D$)

$$E_{6ii} := \{\forall \mathcal{P}_2 \text{ with } |\mathcal{P}_2| \geq \lfloor (1 - \kappa'')n \rfloor \text{ it holds that } \frac{1}{n}\|\boldsymbol{f}^* - \boldsymbol{y}\|_{\mathcal{P}_2}^2 \geq c_{\text{lower}} \cdot \sigma^2\},$$

fulfills, for $n$ large enough, $P(E_{6ii}) \geq 1 - n^{-1}$.

Lemma D.5 states that the total amount of noise $\|\boldsymbol{\varepsilon}\|_{[n]}^2$ concentrates around its mean $n\sigma^2$. More precisely, we will use that for any $c_\varepsilon \in (0, 1)$ the event (over the draw of $D$)

$$E_5 := \left\{ \frac{1}{n}\|\boldsymbol{f}^* - \boldsymbol{y}\|_{[n]}^2 \geq c_\varepsilon \cdot \sigma^2 \right\},$$

fulfills $P(E_5) \geq 1 - \exp\left(-n \cdot \left(\frac{1-c_\varepsilon}{2}\right)^2\right)$.

**Step 2c: Lower bounding $\|\varepsilon\|_{\tilde{\mathcal{P}}^c}^2$.**

Given some function $f \in \mathcal{H}$, assume in steps 2c and 2d that event $E$ holds and that $\tilde{\mathcal{P}} \subseteq [n]$ denotes a subset of the training set that fulfills $|\tilde{\mathcal{P}}| \geq \lceil (1 - \kappa'')n \rceil$ and $|f^*(\boldsymbol{x}_i) - f(\boldsymbol{x}_i)| < c''$ $\forall i \in \tilde{\mathcal{P}}$.

In step 2c, assume we choose $\tilde{c}_{\text{fit}} > 0$ such that $\tilde{c}_{\text{fit}}\|\boldsymbol{f}^* - \boldsymbol{y}\|_{\tilde{\mathcal{P}}}^2 \leq \|\boldsymbol{f} - \boldsymbol{y}\|_{\tilde{\mathcal{P}}}^2$. Then by the overfitting Assumption (O) it holds that

$$\frac{1}{n}\left(\tilde{c}_{\text{fit}}\|\boldsymbol{f}^* - \boldsymbol{y}\|_{\tilde{\mathcal{P}}}^2 + \|\boldsymbol{f} - \boldsymbol{y}\|_{\tilde{\mathcal{P}}^c}^2\right) \leq \frac{1}{n}\left(\|\boldsymbol{f} - \boldsymbol{y}\|_{\tilde{\mathcal{P}}}^2 + \|\boldsymbol{f} - \boldsymbol{y}\|_{\tilde{\mathcal{P}}^c}^2\right) \leq (1 - c_{\text{fit}})\sigma^2. \qquad \text{(D.4)}$$

If we restrict ourselves to event $E_5$, dropping the term $\|\boldsymbol{f} - \boldsymbol{y}\|_{\tilde{\mathcal{P}}^c}^2$ in (D.4), then dividing by $\tilde{c}_{\text{fit}}$ and subtracting the result from the inequality in the definition of event $E_5$ yields

$$\frac{1}{n}\|\boldsymbol{\varepsilon}\|_{\tilde{\mathcal{P}}^c}^2 = \frac{1}{n}\|\boldsymbol{f}^* - \boldsymbol{y}\|_{\tilde{\mathcal{P}}^c}^2 \geq c_\varepsilon \sigma^2 - \frac{1 - c_{\text{fit}}}{\tilde{c}_{\text{fit}}}\sigma^2. \qquad \text{(D.5)}$$

**Step 2d: Choosing the constants.**

If we choose $c_\varepsilon := 1 - \frac{c_{\text{fit}}}{4}$ and $\tilde{c}_{\text{fit}} := \frac{2 - 2c_{\text{fit}}}{2 - c_{\text{fit}}} \in (0, 1)$, then (D.5) becomes

$$\frac{1}{n}\|\boldsymbol{\varepsilon}\|_{\tilde{\mathcal{P}}^c}^2 \geq \frac{c_{\text{fit}}}{4}\sigma^2.$$

Now it is left to show that the condition $\tilde{c}_{\text{fit}}\|\boldsymbol{f}^* - \boldsymbol{y}\|_{\tilde{\mathcal{P}}}^2 \leq \|\boldsymbol{f} - \boldsymbol{y}\|_{\tilde{\mathcal{P}}}^2$, that is required for Step 2c, holds with high probability with our choice of $\tilde{c}_{\text{fit}}$.

With some arbitrary but fixed $\varepsilon_{\text{lower}} \in (0, \sqrt{c_{\text{lower}}})$, choose $c'' := (1 - \sqrt{\tilde{c}_{\text{fit}}})\left(\sqrt{\frac{c_{\text{lower}}}{1 - \kappa''}} - \frac{\varepsilon_{\text{lower}}}{\sqrt{1 - \kappa''}}\right)\sigma$. Then on event $E_{6ii}$, for $n$ large enough, it holds that

$$(1 - \sqrt{\tilde{c}_{\text{fit}}})\frac{1}{\sqrt{n}}\|\boldsymbol{f}^* - \boldsymbol{y}\|_{\tilde{\mathcal{P}}} \geq (1 - \sqrt{\tilde{c}_{\text{fit}}})\sqrt{c_{\text{lower}}}\sigma \geq \sqrt{1 - \kappa''} \cdot c'' + \frac{c''}{\sqrt{n}}. \qquad \text{(D.6)}$$

By definition of $\tilde{\mathcal{P}}$, it holds that

$$\|\boldsymbol{f} - \boldsymbol{f}^*\|_{\tilde{\mathcal{P}}}^2 = \sum_{i \in \tilde{\mathcal{P}}} (f(\boldsymbol{x}_i) - f^*(\boldsymbol{x}_i))^2 < \lceil (1 - \kappa'')n \rceil (c'')^2,$$

so that

$$\frac{1}{\sqrt{n}}\|\boldsymbol{f} - \boldsymbol{f}^*\|_{\tilde{\mathcal{P}}} < \sqrt{1 - \kappa''} \cdot c'' + \frac{c''}{\sqrt{n}}. \tag{D.7}$$

Now, using the triangle inequality, (D.7) and (D.6) yields the condition required for Step 2c,

$$\frac{1}{\sqrt{n}}\|\boldsymbol{f} - \boldsymbol{y}\|_{\tilde{\mathcal{P}}}$$
$$\geq \frac{1}{\sqrt{n}}\|\boldsymbol{f}^* - \boldsymbol{y}\|_{\tilde{\mathcal{P}}} - \frac{1}{\sqrt{n}}\|\boldsymbol{f} - \boldsymbol{f}^*\|_{\tilde{\mathcal{P}}}$$
$$\geq \frac{1}{\sqrt{n}}\|\boldsymbol{f}^* - \boldsymbol{y}\|_{\tilde{\mathcal{P}}} - \sqrt{1 - \kappa''} \cdot c'' - \frac{c''}{\sqrt{n}}$$
$$\geq \sqrt{\tilde{c}_{\text{fit}}} \frac{1}{\sqrt{n}}\|\boldsymbol{f}^* - \boldsymbol{y}\|_{\tilde{\mathcal{P}}}.$$

**Step 2e: Upper bounding the probability of $E$.**

To conclude, we have seen in steps 2c and 2d that on $E \cap E_{6ii} \cap E_5$, it holds that

$$\frac{1}{n}\|\boldsymbol{\varepsilon}\|_{\tilde{\mathcal{P}}^c}^2 \geq \frac{c_{\text{fit}}}{4}\sigma^2.$$

On $E_{6i}$, it holds that

$$\frac{1}{n}\|\boldsymbol{\varepsilon}\|_{\tilde{\mathcal{P}}^c}^2 < \frac{c_{\text{fit}}}{4}\sigma^2.$$

Hence $E_{6i} \cap E \cap E_{6ii} \cap E_5 = \emptyset$. This implies $E \subseteq (E_5 \cap E_{6i} \cap E_{6ii})^c$, where the right hand side is independent of $f \in \mathcal{H}$ and just depends on the training data $D$. Since $P(E_{6i}) \geq 1 - n^{-1}$ and $P(E_{6ii} \cap E_5) \geq 1 - n^{-1} - \exp\left(-n \cdot \left(\frac{1-c_\varepsilon}{2}\right)^2\right)$, it must hold that, for $n$ large enough,

$$P(E) \leq P((E_5 \cap E_{6i} \cap E_{6ii})^c) \leq 2n^{-1} + \exp\left(-n \cdot \left(\frac{1-c_\varepsilon}{2}\right)^2\right) \leq 3n^{-1}. \qquad \square$$

The following proposition generalizes the inconsistency result for small bandwidths, Proposition 5 in Buchholz (2022), beyond interpolating estimators to estimators whose RKHS norm is at most a constant factor larger than the RKHS norm of the minimum-norm interpolant. The intuition is that if the bandwidth is too small, then the minimum-norm interpolant $g_{D,\gamma}$ returns to $0$ between the training points. Then $\|g_{D,\gamma}\|_{L_2(\rho)}$ is smaller and bounded away from $\|f^*\|_{L_2(\rho)}$. We can replace $g_{D,\gamma}$ by any other function $f \in \mathcal{H}$ that fulfills Assumption (N).

**Proposition D.4** (Inconsistency for small bandwidths). *Under the assumptions of Proposition D.2, there exist constants $B, c > 0$ such that, with probability $1 - O(n^{-1})$ over the draw of $D$: For any function $f \in \mathcal{H}$ that fulfills Assumption (N) but not necessarily Assumption (O), the excess risk satisfies*

$$\mathbb{E}_{\boldsymbol{x}}(f(\boldsymbol{x}) - f^*(\boldsymbol{x}))^2 \geq c > 0,$$

*where $c$ depends neither on $n$ nor on $\gamma < Bn^{-1/d}$.*

*Proof.* Denote the upper bound on the Lebesgue density of $P_X$ by $C_u$. The triangle inequality implies

$$\|f^* - f\|_{L_2(P_X)} \geq \|f^*\|_{L_2(P_X)} - \|f\|_{L_2(P_X)} \geq \|f^*\|_{L_2(P_X)} - \sqrt{C_u}\|f\|_2$$
$$\geq \|f^*\|_{L_2(P_X)} - \sqrt{C_u}\|f\|_{\mathcal{H}} \geq \|f^*\|_{L_2(P_X)} - C_{\mathcal{H}}\sqrt{C_u}\|g_{D,\gamma}\|_{\mathcal{H}},$$

where $\|f\|_2 \leq \|f\|_{\mathcal{H}}$ follows from the fact that the Fourier transform $\hat{k}$ of the kernel satisfies $\hat{k}(\xi) \leq 1$. Now in the proof of Lemma 17 in Buchholz (2022) $a > 0$ can be chosen smaller to generalize the statement to

$$\|g_{D,\gamma}\|_{\mathcal{H}}^2 \leq \frac{1}{6C_{\mathcal{H}}^2 C_u}\|f^*\|_{L_2(P_X)}^2 + c_9(\gamma^2 n^{2/d} + \gamma^{2s} n^{2s/d}),$$

where $c_9$ depends on $c_u, f^*, d, s$ and $C_{\text{norm}}$. Finally we can choose $B$ small enough such that Eq. (32) in Buchholz (2022) can be replaced by $C_{\mathcal{H}}\sqrt{C_u}\|g_{D,\gamma}\|_{\mathcal{H}} \leq \frac{2}{3}\|f^*\|_{L_2(P_X)}$ so that we get

$$\|f^* - f\|_{L_2(P_X)} \geq \frac{1}{3}\|f^*\|_{L_2(P_X)} > 0. \qquad \square$$

## D.1 Auxiliary results for the proof of Theorem 1

**Lemma D.5** (Concentration of $\chi_n^2$ variables)**.** *Let $U$ be a chi-squared distributed random variable with $n$ degrees of freedom. Then, for any $c \in (0,1)$ it holds that*

$$P\left(\frac{U}{n} \leq c\right) \leq \exp\left(-n \cdot \left(\frac{1-c}{2}\right)^2\right).$$

*Proof.* Lemma 1 in Laurent and Massart (2000) implies for any $x > 0$,

$$P\left(\frac{U}{n} \leq 1 - 2\sqrt{\frac{x}{n}}\right) \leq \exp\left(-x\right).$$

Solving $c = 1 - 2\sqrt{\frac{x}{n}}$ for $x$ yields $x = n \cdot \left(\frac{1-c}{2}\right)^2$. $\qquad\square$

**Lemma D.6.** *Let $\varepsilon_1, \ldots, \varepsilon_n$ be i.i.d. $\mathcal{N}(0, \sigma^2)$ random variables, $\sigma^2 > 0$. Let $(\varepsilon^2)^{(i)}$ denote the $i$-th largest of $\varepsilon_1^2, \ldots, \varepsilon_n^2$.*

    *(i)* ***A constant fraction of noise cannot concentrate on less than $\Theta(n)$ points:*** *For all constants $\alpha, c > 0$ there exists a constant $C \in (0,1)$ such that with probability at least $1 - n^{-\alpha}$, for $n$ large enough,*

$$\frac{1}{n}\sum_{i=1}^{\lfloor Cn \rfloor} (\varepsilon^2)^{(i)} < c\sigma^2 .$$

    *(ii)* ***$\Theta(n)$ points amount to a constant fraction of noise:*** *For all constants $\alpha > 0$ and $\kappa \in (0,1)$ there exists a constant $c > 0$ such that with probability at least $1 - n^{-\alpha}$, for $n$ large enough,*

$$\frac{1}{n}\sum_{i=1}^{\lfloor (1-\kappa)n \rfloor} (\varepsilon^2)^{(n-i+1)} \geq c\sigma^2 .$$

*Proof.* Without loss of generality, we can assume $\sigma^2 = 1$.

    (i) For a constant $C \in (0,1)$ yet to be chosen, consider the sum

$$S_{C,n} := \frac{1}{n}\sum_{i=1}^{\lfloor Cn \rfloor} (\varepsilon^2)^{(i)} .$$

For $T > 0$ yet to be chosen, we consider the random set $\mathcal{I}_T := \{i \in [n] \mid \varepsilon_i^2 > T\}$ and denote its size by $K := |\mathcal{I}_T|$. To bound $K$, we note that $K = \xi_1 + \ldots + \xi_n$, where $\xi_i = \mathbb{1}_{\varepsilon_i^2 > T}$. We first want to bound $p_T := \mathbb{E}\xi_i = P(\varepsilon_i^2 > T)$.

The random variables $\varepsilon_i^2$ follow a $\chi_1^2$-distribution, whose CDF we denote by $F(t)$ and whose PDF is

$$f(t) = \mathbb{1}_{(0,\infty)}(t)C_1 t^{-1/2}\exp(-t/2) \tag{D.8}$$

for some absolute constant $C_1$. Moreover, we use $\Phi$ and $\phi$ to denote the CDF and PDF of $\mathcal{N}(0,1)$, respectively.

**Step 1: Tail bounds.** Following Duembgen (2010), we have for $x > 0$:

$$1 - \Phi(x) > \frac{2\phi(x)}{\sqrt{4 + x^2} + x} \geq \frac{2\phi(x)}{2 + x + x} = \frac{\phi(x)}{1 + x}$$

$$1 - \Phi(x) < \frac{2\phi(x)}{\sqrt{2 + x^2} + x} \leq \frac{2\phi(x)}{1 + x} .$$

Hence, for $t > 0$, we have

$$1 - F(t) = 2(1 - \Phi(\sqrt{t})) > \frac{2\phi(\sqrt{t})}{1 + \sqrt{t}} = \sqrt{\frac{2}{\pi}}\frac{\exp(-t/2)}{1 + \sqrt{t}}$$

$$1 - F(t) = 2(1 - \Phi(\sqrt{t})) < \frac{4\phi(\sqrt{t})}{1 + \sqrt{t}} = \sqrt{\frac{8}{\pi}}\frac{\exp(-t/2)}{1 + \sqrt{t}} .$$

By choosing $T := -2\log(C\sqrt{\pi/32}) > 0$, we obtain

$$p_T = 1 - F(T) < \sqrt{\frac{8}{\pi}}\exp(-T/2) = C/2 \,.$$

**Step 2: Bounding $K$.** The random variables $\xi_i$ from above satisfy $\xi_i \in [0,1]$. By Hoeffding's inequality (Steinwart and Christmann, 2008, Theorem 6.10), we have for $\tau > 0$

$$P\left(\frac{1}{n}\sum_{i=1}^{n}(\xi_i - \mathbb{E}\xi_i) \geq (1-0)\sqrt{\frac{\tau}{2n}}\right) \leq \exp(-\tau) \,.$$

We choose $\tau := C^2 n/2$, such that with probability $\geq 1 - \exp(-C^2 n/2)$, we have

$$K/n - p_T = \frac{1}{n}\sum_{i=1}^{n}(\xi_i - \mathbb{E}\xi_i) \leq \sqrt{\frac{C^2 n/2}{2n}} = C/2 \,.$$

Suppose that this holds. Then, $K \leq np_T + Cn/2 < Cn$ and, since $K$ is an integer, $K \leq \lfloor Cn \rfloor$. This implies

$$S_{C,n} \leq \frac{1}{n}\left(\sum_{i=1}^{K}(\varepsilon^2)^{(i)} + (\lfloor Cn \rfloor - K)T\right) \leq CT + \frac{1}{n}\sum_{i=1}^{K}(\varepsilon^2)^{(i)} \,. \tag{D.9}$$

We now want to bound $\sum_{i=1}^{K}(\varepsilon^2)^{(i)}$. To this end, we note that conditioned on $K = k$ for some $k \in [n]$, the $k$ random variables $(\varepsilon_i)_{i \in \mathcal{I}_T}$ are i.i.d. drawn from the distribution of $\varepsilon^2$ given $\varepsilon^2 > T$, for $\varepsilon \sim \mathcal{N}(0,1)$. By $X, X_1, X_2, \ldots$, we denote i.i.d. random variables drawn from the distribution of $\varepsilon^2 - T \mid \varepsilon^2 > T$. This means that conditioned on $K = k$,

$$\sum_{i=1}^{k}(\varepsilon^2)^{(i)} = \sum_{i \in \mathcal{I}_T}\varepsilon_i^2 \quad \text{is distributed as} \quad kT + \sum_{i=1}^{k}X_i \,. \tag{D.10}$$

**Step 3: Conditional expectation.** The density of $X$ is given by

$$p_X(t) = \mathbb{1}_{t>0}\frac{f(T+t)}{1 - F(T)} \stackrel{(D.8)}{\leq} \mathbb{1}_{t>0}\frac{C_1(T+t)^{-1/2}\exp(-(t+T)/2)}{\sqrt{2/\pi}\exp(-T/2)/(1+\sqrt{T})}$$
$$\leq \mathbb{1}_{t>0}C_2\exp(-t/2) \,,$$

where we have used that for $t > 0$,

$$\frac{1 + \sqrt{T}}{\sqrt{T+t}} \leq \frac{1 + \sqrt{T}}{\sqrt{T}} = 1 + \frac{1}{\sqrt{T}} \leq 2$$

since $T = -2\log(C\sqrt{\pi/32}) \geq -2\log(\sqrt{\pi/32}) \approx 1.008$. We can now bound

$$\mathbb{E}[X] = \int_0^{\infty}tp_X(t)\,\mathrm{d}t$$
$$\leq \int_0^{\infty}C_2 t\exp(-t/2)\,\mathrm{d}t = 4C_2 \,. \tag{D.11}$$

**Step 4: Conditional subgaussian norm.** For $t \geq 0$,

$$P(|X| > t) = P(X > t) = \frac{1 - F(T+t)}{1 - F(T)} \leq 2\frac{1 + \sqrt{T}}{1 + \sqrt{T+t}}\frac{\exp(-(T+t)/2)}{\exp(-T/2)}$$
$$\leq 2\exp(-t/2) \,.$$

Since the denominator 2 in $2\exp(-t/2)$ is constant, by Proposition 2.7.1 and Definition 2.7.5 in Vershynin (2018), the subexponential norm $\|X\|_{\psi_1}$ is therefore bounded by an absolute constant $C_3$. Moreover, by Excercise 2.7.10 in Vershynin (2018), we have $\|X - \mathbb{E}X\|_{\psi_1} \leq C_4\|X\|_{\psi_1} \leq C_5$ for absolute constants $C_4, C_5$.

**Step 5: Conditional Concentration.** Now, Bernstein's inequality for subexponential random variables (Vershynin, 2018, Corollary 2.8.1) yields for $t \geq 0$ and some absolute constant $C_6 > 0$:

$$P\left(\left|\sum_{i=1}^{k}X_i - \mathbb{E}X_i\right| \geq t\right) \leq 2\exp\left(-C_6\min\left(\frac{t^2}{kC_5^2}, \frac{t}{C_5}\right)\right) \,. \tag{D.12}$$

We choose $t = C_5 C n$ and obtain for $k \leq C n$

$$P \left( \sum_{i=1}^{k} (\varepsilon^2)^{(i)} \geq kT + 4C_2 k + C_5 C n \,\middle|\, K = k \right)$$

$$\overset{\text{(D.10)}}{=} P \left( \sum_{i=1}^{k} X_i \geq 4C_2 k + C_5 C n \,\middle|\, K = k \right)$$

$$\overset{\text{(D.11)}}{\leq} P \left( \left| \sum_{i=1}^{k} X_i - \mathbb{E} X_i \right| \geq t \right)$$

$$\overset{\text{(D.12)}}{\leq} 2 \exp\left( -C_6 C n \right) .$$

**Step 6: Final bound.** From Step 2, we know that $K \leq \lfloor C n \rfloor$ with probability $\geq 1 - \exp(-C^2 n / 2)$. Moreover, in this case, Step 5 yields

$$\sum_{i=1}^{K} (\varepsilon^2)^{(i)} < KT + 4C_2 K + C_5 C n \leq C n (T + 4C_2 + C_5)$$

with probability $\geq 1 - \exp(-C_6 C n)$. By Eq. (D.9), we therefore have

$$S_{C,n} < CT + C(T + 4C_2 + C_5) = -4C \log(C \sqrt{\pi/32}) + C_7 C .$$

Since $\lim_{C \searrow 0} -C \log(C) = 0$, we can choose $C \in (0, 1)$ such that $-4C \log(C \sqrt{\pi/32}) + C_7 C < c$ for the given constant $c > 0$ from the theorem statement, and obtain the desired bound with high probability in $n$.

(ii) Since the $\varepsilon_i^2$ are non-negative and their distribution has a density, there must exist $T > 0$ with $P(\varepsilon_i^2 < T) \leq (1 - \kappa)/4$. Similar to the proof of (i), we then want to bound $K := |\{i \in [n] \mid \varepsilon_i^2 < T\}| = \xi_1 + \ldots + \xi_i$ with $\xi_i = \mathbb{1}_{\varepsilon_i^2 < T}$. The $\xi_i \in [0, 1]$ are independent with $\mathbb{E} \xi_i = P(\varepsilon_i^2 < T) \leq (1 - \kappa)/4$. As in Step 2 of (i), Hoeffding's inequality then yields for $\tau > 0$:

$$P \left( \frac{1}{n} \sum_{i=1}^{n} (\xi_i - \mathbb{E} \xi_i) \geq (1 - 0) \sqrt{\frac{\tau}{2n}} \right) \leq \exp(-\tau) .$$

We set $\tau := (1 - \kappa)^2 n / 2$, such that with probability $\geq 1 - \exp((1 - \kappa)^2 n / 2)$, we have

$$K/n - (1 - \kappa)/4 \leq K/n - P(\varepsilon_i^2 < T) = \frac{1}{n} \sum_{i=1}^{n} (\xi_i - \mathbb{E} \xi_i) < \sqrt{\frac{(1 - \kappa)^2 n / 2}{2n}}$$

$$= \frac{1 - \kappa}{2} .$$

In this case, we have

$$\frac{1}{n} \sum_{i=1}^{\lfloor (1-\kappa)n \rfloor} (\varepsilon^2)^{(n-i+1)} \geq \frac{1}{n} (\lfloor (1-\kappa)n \rfloor - K) T \geq \frac{1}{n} (((1-\kappa)n - 1) - K) T$$

$$\geq \left( \frac{1 - \kappa}{4} - \frac{1}{n} \right) T ,$$

where the right-hand side is lower bounded by $c := (1 - \kappa) T / 8$ for $n$ large enough. $\qquad \square$

The next lemma is a generalization of Lemma 9 in Buchholz (2022) to arbitrary fractions $\kappa$ of the training points. Therefore, for any $\kappa \in (0, 1)$ define

$$\delta_{\min}(\kappa) = n^{-1/d} \left( \frac{\kappa}{C_\rho \omega_d} \right)^{1/d} ,$$

**Lemma D.7** (Generalization of Lemma 9 in Buchholz (2022))**.** *Let $\kappa, \nu \in (0, 1)$, and let $c_\Omega > 0$ be a constant that satisfies $P_X(\mathrm{dist}(\boldsymbol{x}, \partial\Omega) < c_\Omega) \leq \kappa$. Let $\mathcal{P} = \{\boldsymbol{x}_1, \ldots, \boldsymbol{x}_n\}$ be i.i.d. points distributed according to the measure $P_X$, which has lower and upper bounded density on its entire*

*bounded open Lipschitz domain* $\Omega \subseteq \mathbb{R}^d$, $C_l \leq p_X(\boldsymbol{x}) \leq C_u$. *Then there exists a constant* $\Theta > 0$ *depending on* $d, C_u, \nu$ *such that with probability at least* $1 - \exp\left(-\frac{3\kappa n}{7}\right)$ *there exists a good subset* $\mathcal{P}' \subseteq \mathcal{P}$, $|\mathcal{P}'| \geq (1 - 7\kappa)n$, *with the following properties: For* $\boldsymbol{x} \in \mathcal{P}'$ *we have* $\mathrm{dist}(\boldsymbol{x}, \partial\Omega) \geq c_\Omega$, $|\boldsymbol{x} - \boldsymbol{y}| > \delta_{\min}(\kappa)$ *for* $\boldsymbol{x} \neq \boldsymbol{y} \in \mathcal{P}'$, *and for all* $\boldsymbol{x} \in \mathcal{P}'$ *we have*

$$\sum_{\boldsymbol{y} \in \mathcal{P}' \setminus \{\boldsymbol{x}\}} |\boldsymbol{x} - \boldsymbol{y}|^{-d-2\nu} \leq \frac{2\Theta\delta_{\min}(\kappa)^{-2\nu} n}{\kappa^2}.$$

*Proof.* First by the definition of $\delta_{\min}$, it holds that

$$P\left(\boldsymbol{x}_j \in \bigcup_{i<j} B\left(\boldsymbol{x}_i, \delta_{\min}\right)\right) \leq C_u \omega_d \delta_{\min}^d n \leq \kappa$$

Also for all $\boldsymbol{y} \in \Omega$

$$\mathbb{E}_{\boldsymbol{x}}\left((\boldsymbol{x} - \boldsymbol{y})^{-d-2s} \mathbf{1}\left(|\boldsymbol{x} - \boldsymbol{y}| \geq \delta_{\min}\right)\right) = \int_{B(\boldsymbol{y},\delta_{\min})^c} |\boldsymbol{x} - \boldsymbol{y}|^{-d-2\nu} p_X(\boldsymbol{x}) \mathrm{d}\boldsymbol{x}$$

$$\leq C_u \int_{B(\boldsymbol{x},\delta_{\min})^c} |\boldsymbol{x} - \boldsymbol{y}|^{-d-2\nu} \mathrm{d}\boldsymbol{y} \leq \Theta\delta_{\min}^{-2\nu}$$

for some $\Theta > 0$ depending only on $C_u, d$ and $\nu$. We conclude that for each $j$

$$P\left(\sum_{i<j} |\boldsymbol{x}_i - \boldsymbol{x}_j|^{-d-2\nu} \mathbf{1}\left(|\boldsymbol{x}_i - \boldsymbol{x}_j| > \delta_{\min}\right) > \frac{\Theta\delta_{\min}^{-2\nu} n}{\kappa}\right) \leq \kappa.$$

Also $P\left(\mathrm{dist}\left(\boldsymbol{x}_j, \partial\Omega\right) < c_\Omega\right) < \kappa$. The union bound implies that

$$P\left(\boldsymbol{x}_j \notin \bigcup_{i<j} B\left(\boldsymbol{x}_i, \delta_{\min}\right), \sum_{i<j} |\boldsymbol{x}_i - \boldsymbol{x}_j|^{-d-2\nu} \mathbf{1}_{|\boldsymbol{x}_i - \boldsymbol{x}_j| > \delta_{\min}} < \frac{\Theta\delta_{\min}^{-2\nu} n}{\kappa}, \mathrm{dist}\left(\boldsymbol{x}_j, \partial\Omega\right) > c_\Omega\right)$$

$$= P\left(\boldsymbol{x}_j \notin \bigcup_{i<j} B\left(\boldsymbol{x}_i, \delta_{\min}\right), \sum_{i<j} |\boldsymbol{x}_i - \boldsymbol{x}_j|^{-d-2\nu} < \frac{\Theta\delta_{\min}^{-2\nu} n}{\kappa}, \mathrm{dist}\left(\boldsymbol{x}_j, \partial\Omega\right) > c_\Omega\right) \geq 1 - 3\kappa.$$

We use a martingale construction similar to the one in Lemma 7 of Buchholz (2022) by defining

$$E_j := \left\{\boldsymbol{x}_j \in \bigcup_{i<j} B\left(\boldsymbol{x}_i, \delta_{\min}\right), \text{ or } \sum_{i<j} |\boldsymbol{x}_i - \boldsymbol{x}_j|^{-d-2\nu} \geq \frac{\Theta\delta_{\min}^{-2\nu} n}{\kappa}, \text{ or } \mathrm{dist}(\boldsymbol{x}_j, \partial\Omega) \leq c_\Omega\right\}.$$

Now define $S_n := \sum_{i=1}^n \mathbf{1}_{E_i}$. Using the filtration $\mathcal{F}_i = \sigma(\boldsymbol{x}_1, \ldots, \boldsymbol{x}_i)$, $S_n$ can be decomposed into $S_n = M_n + A_n$, where $M_n$ is a martingale and $A_n$ is predictable with respect to $\mathcal{F}_n$. We then get $A_n \leq \sum_{i=1}^n P(E_i | \mathcal{F}_{i-1}) \leq 3\kappa n$ as well as $\mathrm{Var}(M_i | \mathcal{F}_{i-1}) \leq 3\kappa$. Hence Freedman's inequality Theorem D.8 yields

$$P(S_n \geq 6\kappa n) \leq P(A_n \geq 3\kappa n) + P(M_n \geq 3\kappa n) \leq \exp\left(-\frac{3\kappa n}{7}\right).$$

This implies that with probability at least $1 - \exp\left(-\frac{3\kappa n}{7}\right)$ we can find a subset $\mathcal{P}_s = \{\boldsymbol{z}_1, \ldots, \boldsymbol{z}_m\}$ with $|\mathcal{P}_s| \geq (1 - 6\kappa)n$ on which it holds that $\min_{i \neq j} |\boldsymbol{z}_i - \boldsymbol{z}_j| \geq \delta_{\min}$, $\mathrm{dist}\left(\boldsymbol{z}_j, \partial\Omega\right) \geq c_\Omega$ and

$$\sum_{i \neq j} |\boldsymbol{z}_i - \boldsymbol{z}_j|^{-d-2\nu} \leq \frac{2\Theta\delta_{\min}^{-2\nu} n^2}{\kappa}.$$

Using Markov's inequality we see that there are at most $\kappa n$ points in $\mathcal{P}_s$ such that

$$\sum_{\boldsymbol{z}' \in \mathcal{P}_s, \boldsymbol{z} \neq \boldsymbol{z}'} |\boldsymbol{z} - \boldsymbol{z}'|^{-d-2\nu} \geq \frac{2\Theta\delta_{\min}^{-2\nu} n}{\kappa^2}.$$

Removing those points we find a subset $\mathcal{P}' \subset \mathcal{P}_s$ such that $|\mathcal{P}'| \geq (1 - 7\kappa)n$ and for each $\boldsymbol{z} \in \mathcal{P}'$

$$\sum_{\boldsymbol{z}' \in \mathcal{P}_s, \boldsymbol{z} \neq \boldsymbol{z}'} |\boldsymbol{z} - \boldsymbol{z}'|^{-d-2\nu} \leq \frac{2\Theta \delta_{\min}^{-2\nu} n}{\kappa^2}. \qquad \qquad \square$$

**Theorem D.8** (Freedman's inequality, Theorem 6.1 in Chung and Lu (2006))**.** *Let $M_i$ be a discrete martingale adapted to the filtration $\mathcal{F}_i$ with $M_0 = 0$ that satisfies for all $i \geq 0$*

$$|M_{i+1} - M_i| \leq K$$
$$\mathrm{Var}\,(M_i \mid \mathcal{F}_{i-1}) \leq \sigma_i^2.$$

*Then*

$$P\,(M_n - \mathbb{E}\,(M_n) \geq \lambda) \leq e^{-\frac{\lambda^2}{2\sum_{i=1}^n \sigma_i^2 + K\lambda/3}}.$$

# E   Translating between $\mathbb{R}^d$ and $\mathbb{S}^d$

Since the RKHS of the ReLU NTK and NNGP kernels mentioned in Theorem 4 are equivalent to the Sobolev spaces $H^{(d+1)/2}(\mathbb{S}^d)$ and $H^{(d+3)/2}(\mathbb{S}^d)$, respectively (Chen and Xu, 2021, Bietti and Bach, 2021) (detailed summary in Appendix B.4). Inconsistency of functions in these RKHS that fulfill Assumptions (O) and (N), as in Theorem 1, follows immediately by adapting Theorem 1 via Lemma E.1. In particular, inconsistency holds for the gradient flow and gradient descent estimators $f_{t,\rho}$ and $f_{t,\rho}^{\mathrm{GD}}$ as soon as they overfit with lower bounded probability.

For arbitrary open sphere caps $T := \{\boldsymbol{x} \in \mathbb{S}^d \mid x_{d+1} < v\}$, $v \in (-1, 1)$, and the open unit ball $B_1(0) := \{\boldsymbol{y} \in \mathbb{R}^d \mid \|\boldsymbol{y}\|_2 < 1\}$, define the scaled stereographic projection $\phi : T \to B_1(0) \subseteq \mathbb{R}^d$ as

$$\phi(x_1, \ldots, x_{d+1}) = \left( \frac{c_v x_1}{1 - x_{d+1}}, \ldots, \frac{c_v x_d}{1 - x_{d+1}} \right),$$

where the normalization constant $c_v = \sqrt{\frac{1-v}{1+v}}$ ensures surjectivity.

Straightforward calculations show that $\phi$ defines a diffeomorphism. Its inverse $\phi^{-1} : B_1(0) \to T$ is given by

$$\phi^{-1}(y_1, \ldots, y_d) = \left( \frac{2c_v^{-1} y_1}{c_v^{-2}\|\boldsymbol{y}\|_2^2 + 1}, \ldots, \frac{2c_v^{-1} y_d}{c_v^{-2}\|\boldsymbol{y}\|_2^2 + 1}, \frac{c_v^{-2}\|\boldsymbol{y}\|_2^2 - 1}{c_v^{-2}\|\boldsymbol{y}\|_2^2 + 1} \right).$$

We can translate kernel learning with the kernel $k$ on $\mathbb{S}^d$ and the probability distribution $P$, where $P_X$ is supported on $T$, to kernel learning with a transformed kernel $\tilde{k}$ and $\tilde{P}$ using a sufficiently smooth diffeomorphism like $\phi : T \to B_1(0) \subseteq \mathbb{R}^d$. If the RKHS of $k$ is equivalent to $H^s(\mathbb{S}^d)$ then the RKHS of $\tilde{k}$ is equivalent to $H^s(B_1(0))$. We formalize this argument in the following lemma. As a consequence it suffices to prove all inconsistency results for Sobolev kernels on $B_1(0)$.

**Lemma E.1** (**Transfer to sphere caps**)**.** *Let $k$ be a kernel on $\mathbb{S}^d$ whose RKHS is equivalent to a Sobolev space $H^s(\mathbb{S}^d)$. For fixed $v \in (-1, 1)$, consider an "open sphere cap" $T := \{\boldsymbol{x} \in \mathbb{S}^d \mid x_{d+1} < v\}$. Furthermore, consider a distribution $P$ such that $P_X$ is supported on $T$ and has lower and upper bounded density $p_X$ on $T$, i.e. $0 < C_l \leq p_X(\boldsymbol{x}) \leq C_u < \infty$ for all $\boldsymbol{x} \in T$. Then*

- *$\tilde{k}(\boldsymbol{x}, \boldsymbol{x}') := k(\phi^{-1}(\boldsymbol{x}), \phi^{-1}(\boldsymbol{x}'))$ defines a positive definite kernel on $B_1(0) \subseteq \mathbb{R}^d$ whose RKHS is equivalent to the Sobolev space $H^s(B_1(0))$,*
- *$\tilde{P} := P \circ \psi^{-1}$ with $\psi(\boldsymbol{x}, y) := (\phi(\boldsymbol{x}), y)$ defines a probability distribution such that $\tilde{P}_{\tilde{X}}$ has lower and upper bounded density on $B_1(0) \subseteq \mathbb{R}^d$,*

*and kernel learning with $(k, P)$ or with $(\tilde{k}, \tilde{P})$ is equivalent in the following sense:*

*For every function $f \in \mathcal{H}(k|_T)$ the transformed function $\tilde{f} = f \circ \phi^{-1} \in \mathcal{H}(\tilde{k})$ has the same RKHS norm, i.e. $\|f\|_{\mathcal{H}(k|_T)} = \|\tilde{f}\|_{\mathcal{H}(\tilde{k})}$. Furthermore, the excess risks of $f$ over $P$ and of $\tilde{f}$ over $\tilde{P}$ coincide, i.e.*

$$\mathbb{E}_{\boldsymbol{x} \sim P_X}(f(\boldsymbol{x}) - f_P^*(\boldsymbol{x}))^2 = \mathbb{E}_{\tilde{\boldsymbol{x}} \sim \tilde{P}_X}(\tilde{f}(\tilde{\boldsymbol{x}}) - \tilde{f}_{\tilde{P}}^*(\tilde{\boldsymbol{x}}))^2,$$

*where $\tilde{f}_{\tilde{P}}^*(\tilde{\boldsymbol{x}}) = \mathbb{E}_{(\tilde{X}, \tilde{Y}) \sim \tilde{P}}(\tilde{Y} | \tilde{X} = \tilde{\boldsymbol{x}})$ denotes the Bayes optimal predictor under $\tilde{P}$.*

**Remark E.2.** Many kernel regression estimators can be explicitly written as $f_D^k(\boldsymbol{x}) = \hat{f}_n(k(\boldsymbol{x}, \boldsymbol{X}), k(\boldsymbol{X}, \boldsymbol{X}), \boldsymbol{y})$ where $\hat{f}_n : \mathbb{R}^n \times \mathbb{R}^{n \times n} \times \mathbb{R}^n \to \mathbb{R}$ denotes a measurable function for all $n \in \mathbb{N}$. Then the explicit form is preserved under the transformation, i.e. $f \circ \phi^{-1} = f_{\tilde{D}}^{\tilde{k}}$ with the transformed data set $\tilde{D} = \{(\phi(\boldsymbol{x}_i), y_i)\}_{i \in [n]}$. ◀

*Proof of Lemma E.1.* **Step 1: Bounded density.** For $i \in [d], j \in [d+1]$, the partial derivatives of $\phi$ are given by

$$\partial_{x_j} \phi_i(\boldsymbol{x}) = \begin{cases} \frac{c_v}{1 - x_{d+1}}, & \text{for } i = j, \\ \frac{c_v x_i}{(1 - x_{d+1})^2}, & \text{for } i \in [d], \ j = d+1, \\ 0, & \text{otherwise.} \end{cases}$$

Given an arbitrary multi-index $\alpha$, the partial derivatives $\partial_\alpha \phi_i \in L^2(T)$, $\partial_\alpha \phi_j^{-1} \in L^2(B_1(0))$ are bounded for all $i \in [d], j \in [d+1]$, using $x_{d+1} \leq v < 1$ and the inverse function theorem.

Now define $\tilde{k}(\boldsymbol{x}, \boldsymbol{x}') := k(\phi^{-1}(\boldsymbol{x}), \phi^{-1}(\boldsymbol{x}'))$, $\psi(\boldsymbol{x}, y) := (\phi(\boldsymbol{x}), y)$ and $\tilde{P} := P \circ \psi^{-1}$. Then using integration by substitution (Stroock et al., 2011, Theorem 5.2.16), the Lebesgue density of $\tilde{P}_X$ is given by

$$p_{\tilde{X}}(\tilde{\boldsymbol{x}}) = p_X(\phi^{-1}(\tilde{\boldsymbol{x}})) J\phi^{-1}(\tilde{\boldsymbol{x}}),$$

where

$$J\phi^{-1}(\tilde{\boldsymbol{x}}) := \left[ \det \left( \left( \langle \partial_i \phi^{-1}(\tilde{\boldsymbol{x}}), \partial_j \phi^{-1}(\tilde{\boldsymbol{x}}) \rangle_{\mathbb{R}^{d+1}} \right)_{i,j \in \{1,\ldots,d\}} \right) \right]^{1/2}.$$

$J\phi$ and $J\phi^{-1}$ can be continuously extended to $\bar{T}$ and $\bar{B}_1(0)$, respectively. Then, since $J\phi^{-1}$ is continuous on a compact set and because $\phi$ with the extended domain remains a diffeomorphism so that $J\phi^{-1}$ cannot attain the value 0, there exists a constant $C_\phi > 0$ such that $\frac{1}{C_\phi} \leq J\phi^{-1}(\tilde{\boldsymbol{x}}) \leq C_\phi$ for all $\tilde{\boldsymbol{x}} \in B_1(0)$. Hence, $p_{\tilde{X}}$ is lower and upper bounded.

**Step 2: Excess risks coincide.** If $(\tilde{X}, \tilde{Y}) \sim \tilde{P}$, the Bayes predictor of $\tilde{Y}$ given $\tilde{X}$ is given by $\tilde{f}^*(\tilde{\boldsymbol{x}}) = \mathbb{E}(\tilde{Y}|\tilde{X} = \tilde{\boldsymbol{x}}) = f^*(\phi^{-1}(\tilde{\boldsymbol{x}}))$.

Let $\pi_1(\boldsymbol{x}, y) = \boldsymbol{x}$ be the projection onto the first component. Then, $\phi(\pi_1(\boldsymbol{x}, y)) = \phi(\boldsymbol{x}) = \pi_1(\phi(\boldsymbol{x}), y) = \pi_1(\psi(\boldsymbol{x}, y))$ and hence

$$\begin{aligned} \mathbb{E}_{\boldsymbol{x} \sim P_X}(f(\boldsymbol{x}) - f^*(\boldsymbol{x}))^2 &= \mathbb{E}_{(\boldsymbol{x}, y) \sim P}(f(\pi_1(\boldsymbol{x}, y)) - f^*(\pi_1(\boldsymbol{x}, y)))^2 \\ &= \mathbb{E}_{(\boldsymbol{x}, y) \sim P}(f(\phi^{-1}(\phi(\pi_1(\boldsymbol{x}, y)))) - f^*(\phi^{-1}(\phi(\pi_1(\boldsymbol{x}, y)))))^2 \\ &= \mathbb{E}_{(\boldsymbol{x}, y) \sim P}(\tilde{f}(\pi_1(\psi(\boldsymbol{x}, y))) - \tilde{f}^*(\pi_1(\psi(\boldsymbol{x}, y))))^2 \\ &= \mathbb{E}_{(\boldsymbol{x}, y) \sim \tilde{P}}(\tilde{f}(\pi_1(\boldsymbol{x}, y)) - \tilde{f}^*(\pi_1(\boldsymbol{x}, y)))^2 \\ &= \mathbb{E}_{\boldsymbol{x} \sim \tilde{P}_{\tilde{X}}}(\tilde{f}(\boldsymbol{x}) - \tilde{f}^*(\boldsymbol{x}))^2 . \end{aligned}$$

**Step 3: Transformed RKHS.** We want to show that $\mathcal{H}(k|_T) \to \mathcal{H}(\tilde{k}), f \mapsto f \circ \phi^{-1}$ defines an isometric isomorphism, which especially shows the statement $\|f\|_{\mathcal{H}(k|_T)} = \|\tilde{f}\|_{\mathcal{H}(\tilde{k})}$ from the proposition. For this, we use the following theorem characterizing RKHSs:

**Theorem E.3** (Theorem 4.21 in Steinwart and Christmann (2008)). *Let $k : X \times X \to \mathbb{R}$ be a positive definite kernel function with feature space $H_0$ and feature map $\Phi_0 : X \to H_0$. Then*

$$H = \{f : X \to \mathbb{R} \mid \exists w \in H_0 : \ f = \langle w, \Phi_0(\cdot) \rangle_{H_0}\} \ \text{with}$$
$$\|f\|_H := \inf\{\|w\|_{H_0} : \ f = \langle w, \Phi_0(\cdot) \rangle_{H_0}\},$$

*is the only RKHS for which $k$ is a reproducing kernel.*

A feature map for $k|_T$ is given by $\Phi : T \to \mathcal{H}(k|_T)$, $\Phi(\boldsymbol{x}) = k(\boldsymbol{x}, \cdot)$. Hence a feature map for $\tilde{k}$ is given by $\Phi \circ \phi^{-1} : B_1(0) \to \mathcal{H}(k|_T)$. Theorem E.3 states that

$$\mathcal{H}(k|_T) = \{f : T \to \mathbb{R} \mid \exists w \in \mathcal{H}(k|_T) : \ f = \langle w, \Phi(\cdot) \rangle_{\mathcal{H}(k|_T)}\} \ \text{with} \tag{E.1}$$
$$\|f\|_{\mathcal{H}(k|_T)} := \inf\{\|w\|_{\mathcal{H}(k|_T)} : \ f = \langle w, \Phi(\cdot) \rangle_{\mathcal{H}(k|_T)}\},$$

as well as

$$\mathcal{H}(\tilde{k}) = \left\{ \tilde{f} : B_1(0) \to \mathbb{R} \mid \exists w \in \mathcal{H}(k|_T) : \ \tilde{f} = \langle w, \Phi \circ \phi^{-1}(\cdot) \rangle_{\mathcal{H}(k|_T)} \right\} \text{ with} \qquad \text{(E.2)}$$

$$\|\tilde{f}\|_{\mathcal{H}(\tilde{k})} := \inf\{\|w\|_{\mathcal{H}(k|_T)} : \ \tilde{f} = \langle w, \Phi \circ \phi^{-1}(\cdot) \rangle_{\mathcal{H}(k|_T)}\}.$$

As $\phi^{-1}$ is bijective, this characterization induces an isometric isomorphism between $\mathcal{H}(k|_T)$ and $\mathcal{H}(\tilde{k})$ by mapping $f = \langle w, \Phi(\cdot) \rangle_{\mathcal{H}(k|_T)} \in \mathcal{H}(k|_T)$ to $\tilde{f} = f \circ \phi^{-1} = \langle w, \Phi \circ \phi^{-1}(\cdot) \rangle_{\mathcal{H}(k|_T)} \in \mathcal{H}(\tilde{k})$. This shows $\|f\|_{\mathcal{H}(k|_T)} = \|\tilde{f}\|_{\mathcal{H}(\tilde{k})}$.

**Step 4: RKHS of $\tilde{k}$.** We now show that the RKHS of $\tilde{k}$, denoted as $\mathcal{H}(\tilde{k})$, is equivalent to $H^s(B_1(0))$. To this end, denoting $\mathcal{A} \circ \phi := \{f \circ \phi \mid f \in \mathcal{A}\}$ and $\mathcal{A}|_T := \{f|_T \mid f \in \mathcal{A}\}$, we show the following equality of sets (ignoring the norms):

$$\mathcal{H}(\tilde{k}) \circ \phi \overset{\text{(I)}}{=} \mathcal{H}(k|_T) \overset{\text{(II)}}{=} \mathcal{H}(k)|_T \overset{\text{(III)}}{=} H^s(\mathbb{S}^d)|_T \overset{\text{(IV)}}{=} H^s(B_1(0)) \circ \phi \ .$$

Since $\phi$ is bijective, this implies $\mathcal{H}(\tilde{k}) = H^s(B_1(0))$ as sets, and the norm equivalence then follows from Lemma F.7.

Equality (I) follows from Step 3. Equality (II) follows from Theorem E.3 by observing that if $\Phi$ is a feature map for $k$, then $\Phi|_T$ is a feature map for $k|_T$. Equality (III) holds by assumption. To show (IV), we need a characterization of $H^s(\mathbb{S}^d)$ that allows to work with charts like $\phi$.

**Step 4.1: Chart-based characterization of $H^s(\mathbb{S}^d)$.** A trivialization of a Riemannian manifold $(M, g)$ with bounded geometry of dimension $d$ consists of a locally finite open covering $\{U_\alpha\}_{\alpha \in I}$ of $M$, smooth diffeomorphisms $\kappa_\alpha : V_\alpha \subset \mathbb{R}^d \to U_\alpha$, also called charts, and a partition of unity $\{h_\alpha\}_{\alpha \in I}$ of $M$ that fulfills $\text{supp}(h_\alpha) \subseteq U_\alpha$, $0 \le h_\alpha \le 1$ and $\sum_{\alpha \in I} h_\alpha = 1$. An admissible trivialization of $(M, g)$ is a uniformly locally finite trivialization of $M$ that is compatible with geodesic coordinates, for details see (Schneider and Große, 2013, Definition 12).

In our case, define an open neighborhood of $T$ by $U_1 := \{\boldsymbol{x} \in \mathbb{S}^d \mid x_{d+1} < v + \varepsilon\}$ with some $\varepsilon \in (0, 1 - v)$ arbitrary but fixed, and $U_2 := \{\boldsymbol{x} \in \mathbb{S}^d \mid x_{d+1} > v + \varepsilon/2\}$. It holds that $U_1 \cup U_2 = \mathbb{S}^d$. Moreover, there exists an appropriate partition of unity consisting of $C^\infty$ functions $h_1, h_2 : \mathbb{S}^d \to [0, 1]$. Especially, we have $h_1(T) \subseteq h_1(U_2^c) = \{1\}$. Let $\phi_1 : U_1 \to B_{r_1}(0)$ denote the stereographic projection with respect to $\boldsymbol{x}_0 = (0, \ldots, 0, 1)$ as above, scaled such that $\phi_1|_T = \phi$ and hence $\phi_1(T) = B_1(0)$. Similarly, let $\phi_2 : U_2 \to B_{r_2}(0)$ denote an arbitrarily scaled stereographic projection with respect to $\boldsymbol{x}_0 = (0, \ldots, 0, -1)$. Then $(\{U_1, U_2\}, \{\phi_1^{-1}, \phi_1^{-1}\}, \{h_1, h_2\})$ yields an admissible trivialization of $\mathbb{S}^d$ consisting of only two charts. A detailed derivation can be found in (Hubbert et al., 2015, Section 1.7). Therefore (Schneider and Große, 2013, Theorem 14) lets us define the Sobolev norm on $\mathbb{S}^d$ (up to equivalence) as[2]

$$\|g\|_{H^s(\mathbb{S}^d)} := \left( \sum_{\alpha \in I} \|(h_\alpha g) \circ \kappa_\alpha\|_{H^s(\mathbb{R}^d)}^2 \right)^{1/2}$$

$$= \left( \|(h_1 g) \circ \phi_1^{-1}\|_{H^s(\mathbb{R}^d)}^2 + \|(h_2 g) \circ \phi_2^{-1}\|_{H^s(\mathbb{R}^d)}^2 \right)^{1/2},$$

for any distribution $g \in \mathcal{D}'(\mathbb{S}^d)$ (i.e. any continuous linear functional on $C_c^\infty(\mathbb{S}^d)$). Then $g \in H^s(\mathbb{S}^d)$ if and only if $\|g\|_{H^s(\mathbb{S}^d)} < \infty$.

**Step 4.2: Showing (IV).** First, let $g \in H^s(\mathbb{S}^d)$. Then, as we saw in Step 4.1, we must have $\|(h_1 g) \circ \phi_1^{-1}\|_{H^s(\mathbb{R}^d)} < \infty$ and thus $(h_1 g) \circ \phi_1^{-1} \in H^s(\mathbb{R}^d)$. By our discussion in Appendix B.1, we then have

$$(g|_T) \circ \phi^{-1} = ((h_1 g) \circ \phi_1^{-1})|_{B_1(0)} \in H^s(B_1(0)) \ ,$$

which shows $g|_T \in H^s(B_1(0)) \circ \phi$.

Now, let $f \in H^s(B_1(0))$. Then, again following our discussion in Appendix B.1, there exists an extension $\bar{f} \in H^s(\mathbb{R}^d)$ with $\bar{f}|_{B_1(0)} = f$. The set $\mathcal{B} := \phi_1(U_1 \setminus U_2)$ is a closed ball $\overline{B_r(0)}$ of radius

---

[2]Here, the norms are taken on $H^s(\mathbb{R}^d)$ since the respective functions can be extended to $\mathbb{R}^d$ by zero outside of their domain of definition, thanks to the properties of the partition of unity.

$1 < r < r_1$. Hence, we can find $\varphi \in C^\infty(\mathbb{R}^d)$ with $\varphi(B_1(0)) = \{1\}$ and $\varphi((\overline{B_r(0)})^c) = \{0\}$. Since $\varphi$ is smooth with compact support, we have $\varphi \cdot \bar{f} \in H^s(\mathbb{R}^d)$. Define

$$f_{\mathbb{S}^d} : \mathbb{S}^d \to \mathbb{R}, \boldsymbol{x} \mapsto \begin{cases} (\varphi \cdot \bar{f})(\phi(\boldsymbol{x})) & , \boldsymbol{x} \in U_1 \\ 0 & , \boldsymbol{x} \notin U_1 . \end{cases}$$

By construction, we have $f_{\mathbb{S}^d}(\boldsymbol{x}) = 0$ for all $\boldsymbol{x} \in U_2$. Hence, the equivalent Sobolev norm from Step 4.1 is

$$\begin{aligned}
\|f_{\mathbb{S}^d}\|_{H^s(\mathbb{S}^d)} &= \left( \|(h_1 f_{\mathbb{S}^d}) \circ \phi_1^{-1}\|_{H^s(\mathbb{R}^d)}^2 + \|(h_2 f_{\mathbb{S}^d}) \circ \phi_2^{-1}\|_{H^s(\mathbb{R}^d)}^2 \right)^{1/2} \\
&= \|(h_1 \circ \phi_1^{-1}) \cdot \varphi \cdot \bar{f}\|_{H^s(\mathbb{R}^d)} < \infty ,
\end{aligned}$$

which shows $f_{\mathbb{S}^d} \in H^s(\mathbb{S}^d)$. But then, $f \circ \phi = f_{\mathbb{S}^d}|_T \in H^s(\mathbb{S}^d)|_T$.

In total, we obtain $H^s(\mathbb{S}^d)|_T = H^s(B_1(0)) \circ \phi$, which shows (IV). $\qquad\square$

# F  Spectral lower bound

## F.1  General lower bounds

A common first step to analyze the expected excess risk caused by label noise is to perform a bias-variance decomposition and integrate over $\boldsymbol{y}$ first (see e.g. Liang and Rakhlin, 2020, Hastie et al., 2022, Holzmüller, 2021), which is also used in the following lemma.

**Lemma F.1.** *Consider an estimator of the form $f_{\boldsymbol{X},\boldsymbol{y}}(\boldsymbol{x}) = (\boldsymbol{v}_{\boldsymbol{X},\boldsymbol{x}})^\top \boldsymbol{y}$. If $\mathrm{Var}_P(y|\boldsymbol{x}) \geq \sigma^2$ for $P_X$-almost all $\boldsymbol{x}$, then the expected excess risk satisfies*

$$\mathbb{E}_D R_P(f_{\boldsymbol{X},\boldsymbol{y}}) - R_P^* \geq \sigma^2 \mathbb{E}_{\boldsymbol{X},\boldsymbol{x}} \mathrm{tr}(\boldsymbol{v}_{\boldsymbol{X},\boldsymbol{x}}(\boldsymbol{v}_{\boldsymbol{X},\boldsymbol{x}})^\top) .$$

*Proof.* A standard bias-variance decomposition lets us lower-bound the expected excess risk by the estimator variance due to the label noise, which can then be further simplified:

$$\begin{aligned}
\mathbb{E}_D R_P(f_{\boldsymbol{X},\boldsymbol{y}}) - R_P^* &\geq \mathbb{E}_{\boldsymbol{X},\boldsymbol{x}} \left( \mathbb{E}_{\boldsymbol{y}|\boldsymbol{X}} \left[ f_{\boldsymbol{X},\boldsymbol{y}}(\boldsymbol{x})^2 \right] - \left( \mathbb{E}_{\boldsymbol{y}|\boldsymbol{X}}[f_{\boldsymbol{X},\boldsymbol{y}}(\boldsymbol{x})] \right)^2 \right) . \\
&= \mathbb{E}_{\boldsymbol{X},\boldsymbol{x}} \mathbb{E}_{\boldsymbol{y}|\boldsymbol{X}} \left( f_{\boldsymbol{X},\boldsymbol{y}}(\boldsymbol{x}) - \mathbb{E}_{\boldsymbol{y}|\boldsymbol{X}} f_{\boldsymbol{X},\boldsymbol{y}}(\boldsymbol{x}) \right)^2 \\
&= \mathbb{E}_{\boldsymbol{X},\boldsymbol{x}} \mathbb{E}_{\boldsymbol{y}|\boldsymbol{X}} (\boldsymbol{v}_{\boldsymbol{X},\boldsymbol{x}})^\top (\boldsymbol{y} - \mathbb{E}_{\boldsymbol{y}|\boldsymbol{X}} \boldsymbol{y})(\boldsymbol{y} - \mathbb{E}_{\boldsymbol{y}|\boldsymbol{X}} \boldsymbol{y})^\top \boldsymbol{v}_{\boldsymbol{X},\boldsymbol{x}} \\
&= \mathbb{E}_{\boldsymbol{X},\boldsymbol{x}} (\boldsymbol{v}_{\boldsymbol{X},\boldsymbol{x}})^\top \left[ \mathbb{E}_{\boldsymbol{y}|\boldsymbol{X}} (\boldsymbol{y} - \mathbb{E}_{\boldsymbol{y}|\boldsymbol{X}} \boldsymbol{y})(\boldsymbol{y} - \mathbb{E}_{\boldsymbol{y}|\boldsymbol{X}} \boldsymbol{y})^\top \right] \boldsymbol{v}_{\boldsymbol{X},\boldsymbol{x}} \\
&= \mathbb{E}_{\boldsymbol{X},\boldsymbol{x}} (\boldsymbol{v}_{\boldsymbol{X},\boldsymbol{x}})^\top \mathrm{Cov}(\boldsymbol{y}|\boldsymbol{X}) \boldsymbol{v}_{\boldsymbol{X},\boldsymbol{x}} .
\end{aligned}$$

Here, the conditional covariance matrix can be lower bounded in terms of the Loewner order (which is defined as $A \succeq B \Leftrightarrow B - A$ positive semi-definite):

$$\mathrm{Cov}(\boldsymbol{y}|\boldsymbol{X}) = \begin{pmatrix} \mathrm{Var}(y_1|\boldsymbol{x}_1) & & \\ & \ddots & \\ & & \mathrm{Var}(y_n|\boldsymbol{x}_n) \end{pmatrix} \succeq \sigma^2 \boldsymbol{I}_n$$

since the labels $y_i$ are conditionally independent given $\boldsymbol{X}$. We therefore obtain

$$\begin{aligned}
\mathbb{E}_D R_P(f_{\boldsymbol{X},\boldsymbol{y}}) - R_P^* &\geq \mathbb{E}_{\boldsymbol{X},\boldsymbol{x}} (\boldsymbol{v}_{\boldsymbol{X},\boldsymbol{x}})^\top \mathrm{Cov}(\boldsymbol{y}|\boldsymbol{X}) \boldsymbol{v}_{\boldsymbol{X},\boldsymbol{x}} \\
&\geq \sigma^2 \mathbb{E}_{\boldsymbol{X},\boldsymbol{x}} \mathrm{tr}((\boldsymbol{v}_{\boldsymbol{X},\boldsymbol{x}})^\top \boldsymbol{v}_{\boldsymbol{X},\boldsymbol{x}}) \\
&= \sigma^2 \mathbb{E}_{\boldsymbol{X},\boldsymbol{x}} \mathrm{tr}(\boldsymbol{v}_{\boldsymbol{X},\boldsymbol{x}}(\boldsymbol{v}_{\boldsymbol{X},\boldsymbol{x}})^\top) . \qquad\square
\end{aligned}$$

**Proposition 5** (**Spectral lower bound**). *Assume that the kernel matrix $k(\boldsymbol{X}, \boldsymbol{X})$ is almost surely positive definite, and that $\mathrm{Var}(y|\boldsymbol{x}) \geq \sigma^2$ for $P_X$-almost all $\boldsymbol{x}$. Then, the expected excess risk satisfies*

$$\mathbb{E}_D R_P(f_{t,\rho}) - R_P^* \geq \frac{\sigma^2}{n} \sum_{i=1}^n \mathbb{E}_{\boldsymbol{X}} \frac{\lambda_i(k_*(\boldsymbol{X}, \boldsymbol{X})/n) \left( 1 - e^{-2t(\lambda_i(k(\boldsymbol{X},\boldsymbol{X})/n)+\rho)} \right)^2}{(\lambda_i(k(\boldsymbol{X}, \boldsymbol{X})/n) + \rho)^2} . \tag{3}$$

*Proof.* Recall from Eq. (1) that

$$f_{t,\rho}(\boldsymbol{x}) = k(\boldsymbol{x}, \boldsymbol{X})\boldsymbol{A}_{t,\rho}(\boldsymbol{X})\boldsymbol{y} ,$$
$$\boldsymbol{A}_{t,\rho}(\boldsymbol{X}) := \left(\boldsymbol{I}_n - e^{-\frac{2}{n}t(k(\boldsymbol{X},\boldsymbol{X})+\rho n\boldsymbol{I}_n)}\right)\left(k(\boldsymbol{X},\boldsymbol{X}) + \rho n\boldsymbol{I}_n\right)^{-1} .$$

By setting $(\boldsymbol{v}_{\boldsymbol{X},\boldsymbol{x}})^\top := k(\boldsymbol{x}, \boldsymbol{X})\boldsymbol{A}_{t,\rho}(\boldsymbol{X})$, we can write $f_{\boldsymbol{X},\boldsymbol{y},t,\rho}(\boldsymbol{x}) := f_{t,\rho}(\boldsymbol{x}) = (\boldsymbol{v}_{\boldsymbol{X},\boldsymbol{x}})^\top\boldsymbol{y}$. Using Lemma F.1, we then obtain

$$\mathbb{E}_D R_P(f_{\boldsymbol{X},\boldsymbol{y},t,\rho}) - R_P^* \geq \sigma^2 \mathbb{E}_{\boldsymbol{X},\boldsymbol{x}}\operatorname{tr}(\boldsymbol{v}_{\boldsymbol{X},\boldsymbol{x}}(\boldsymbol{v}_{\boldsymbol{X},\boldsymbol{x}})^\top)$$
$$= \sigma^2 \mathbb{E}_{\boldsymbol{X},\boldsymbol{x}}\operatorname{tr}\left(\boldsymbol{A}_{t,\rho}(\boldsymbol{X})^\top k(\boldsymbol{X},\boldsymbol{x})k(\boldsymbol{x},\boldsymbol{X})\boldsymbol{A}_{t,\rho}(\boldsymbol{X})\right) .$$

Since

$$(\mathbb{E}_{\boldsymbol{x}}k(\boldsymbol{X},\boldsymbol{x})k(\boldsymbol{x},\boldsymbol{X}))_{ij} = \mathbb{E}_{\boldsymbol{x}}k(\boldsymbol{x}_i,\boldsymbol{x})k(\boldsymbol{x},\boldsymbol{x}_j) = k_*(\boldsymbol{x}_i,\boldsymbol{x}_j) = k_*(\boldsymbol{X},\boldsymbol{X})_{ij} ,$$

we conclude

$$\mathbb{E}_D R_P(f_{\boldsymbol{X},\boldsymbol{y},t,\rho}) - R_P^* \geq \sigma^2 \mathbb{E}_{\boldsymbol{X}}\operatorname{tr}(\boldsymbol{A}_{t,\rho}^\top k_*(\boldsymbol{X},\boldsymbol{X})\boldsymbol{A}_{t,\rho})$$
$$= \sigma^2 \mathbb{E}_{\boldsymbol{X}}\operatorname{tr}(k_*(\boldsymbol{X},\boldsymbol{X})\boldsymbol{A}_{t,\rho}(\boldsymbol{X})\boldsymbol{A}_{t,\rho}(\boldsymbol{X})^\top) .$$

Richter (1958) showed (see also Mirsky, 1959) that for two symmetric matrices $\boldsymbol{B}, \boldsymbol{C}$, we have $\operatorname{tr}(\boldsymbol{BC}) \geq \sum_{i=1}^n \lambda_i(\boldsymbol{B})\lambda_{n+1-i}(\boldsymbol{C})$. We can therefore conclude

$$\mathbb{E}_D R_P(f_{\boldsymbol{X},\boldsymbol{y},t,\rho}) - R_P^* \geq \sigma^2 \mathbb{E}_{\boldsymbol{X}}\sum_{i=1}^n \lambda_i(k_*(\boldsymbol{X},\boldsymbol{X}))\lambda_{n+1-i}(\boldsymbol{A}_{t,\rho}(\boldsymbol{X})\boldsymbol{A}_{t,\rho}(\boldsymbol{X})^\top) .$$

As $\boldsymbol{A}_{t,\rho}(\boldsymbol{X})\boldsymbol{A}_{t,\rho}(\boldsymbol{X})^\top$ is built only out of the matrices $k(\boldsymbol{X},\boldsymbol{X})$ and $\boldsymbol{I}_n$, it is not hard to see that $\boldsymbol{A}_{t,\rho}(\boldsymbol{X})\boldsymbol{A}_{t,\rho}(\boldsymbol{X})^\top$ has the same eigenbasis as $k(\boldsymbol{X},\boldsymbol{X})$ with eigenvalues

$$\tilde{\lambda}_i := \left(\frac{1 - e^{-\frac{2}{n}t(\lambda_i(k(\boldsymbol{X},\boldsymbol{X}))+\rho n)}}{\lambda_i(k(\boldsymbol{X},\boldsymbol{X})) + \rho n}\right)^2 = \frac{1}{n^2}\left(\frac{1 - e^{-2t(\lambda_i(k(\boldsymbol{X},\boldsymbol{X})/n)+\rho)}}{\lambda_i(k(\boldsymbol{X},\boldsymbol{X})/n) + \rho}\right)^2 .$$

It remains to order these eigenvalues correctly. To this end, we observe that for $\lambda > 0$, the function $g(\lambda) := \frac{1-e^{-2t\lambda}}{\lambda}$ satisfies

$$g'(\lambda) = \frac{2t\lambda e^{-2t\lambda} - (1 - e^{-2t\lambda})}{\lambda^2} = \frac{(2t\lambda + 1)e^{-2t\lambda} - 1}{\lambda^2} \leq \frac{e^{2t\lambda}e^{-2t\lambda} - 1}{\lambda^2} = 0 .$$

Therefore, $g$ is nonincreasing, hence the sequence $(\tilde{\lambda}_i)$ is nondecreasing and thus

$$\lambda_{n+1-i}(\boldsymbol{A}_{t,\rho}\boldsymbol{A}_{t,\rho}^\top) = \tilde{\lambda}_i ,$$

from which the claim follows. $\qquad\square$

**Theorem F.2.** *Let $k$ be a kernel on a compact set $\Omega$ and let $P_X$ be supported on $\Omega$. Suppose that $k(\boldsymbol{X},\boldsymbol{X})$ is almost surely positive definite and that $\operatorname{Var}(y|\boldsymbol{x}) \geq \sigma^2$ for $P_X$-almost all $\boldsymbol{x}$. Fix constants $c > 0$ and $q, C \geq 1$. Suppose that $\lambda_i := \lambda_i(T_{k,P_X}) \geq ci^{-q}$. Let $\mathcal{I}(n)$ be the set of all $i \in [n]$ for which*

$$\lambda_i/C \leq \lambda_i(k(\boldsymbol{X},\boldsymbol{X})/n) \leq C\lambda_i \qquad\qquad (F.1)$$
$$\lambda_i^2/C \leq \lambda_i(k_*(\boldsymbol{X},\boldsymbol{X})/n)$$

*both hold at the same time with probability $\geq 1/2$. Moreover, let $I(n) := \max\{m \in [n] \mid [m] \subseteq \mathcal{I}(n)\}$. Then, there exists a constant $c' > 0$ depending only on $c, C$ such that for all $\rho \in [0, \infty)$ and $t \in (0, \infty]$, the following two bounds hold:*

$$\mathbb{E}_D R_P(f_{\boldsymbol{X},\boldsymbol{y},t,\rho}) - R_P^* \geq c'\sigma^2 \frac{1}{1 + (\rho + t^{-1})n^q} \cdot \frac{|\mathcal{I}(n)|}{n} ,$$
$$\mathbb{E}_D R_P(f_{\boldsymbol{X},\boldsymbol{y},t,\rho}) - R_P^* \geq c'\sigma^2 \min\left\{\frac{(\rho + t^{-1})^{-2}}{n}, \frac{(\rho + t^{-1})^{-1/q}}{n}, \frac{I(n)}{n}\right\} .$$

**Remark F.3.** Theorem F.2 provides two lower bounds, one for general "concentration sets" $\mathcal{I}(n)$ and one that applies if concentration holds for a sequence of "head eigenvalues" $\{1, \ldots, I(n)\} \subseteq \mathcal{I}(n)$. If $I(n) \approx |\mathcal{I}(n)|$, the latter bound is stronger for larger regularization levels, and this bound would be particularly suitable for typical forms of relative concentration inequalities for kernel matrices. However, in this paper, we obtain concentration only for "middle eigenvalues" $\mathcal{I}(n) = \{i \mid \varepsilon n \leq i \leq (1 - \varepsilon)n\}$, and therefore we only use the first bound in the proof of Theorem 6. ◀

*Proof of Theorem F.2.* **Step 1: Miscellaneous inequalities.** For $x > 0$,

$$1 - e^{-x} = 1 - \frac{1}{e^x} \geq 1 - \frac{1}{x+1} = \frac{x}{x+1} = \frac{1}{1 + x^{-1}} \ . \tag{F.2}$$

Moreover, since $(1 + a)^2 \leq (1 + a)^2 + (1 - a)^2 = 2 + 2a^2$, we have for $a \neq -1$:

$$\left(\frac{1}{1+a}\right)^2 \geq \frac{1}{2(1 + a^2)} \ . \tag{F.3}$$

**Step 2: Applying the eigenvalue bound.** Define

$$S_i(\boldsymbol{X}) := \frac{\lambda_i(k_*(\boldsymbol{X}, \boldsymbol{X})/n)\left(1 - e^{-2t(\lambda_i(k(\boldsymbol{X}, \boldsymbol{X})/n)+\rho)}\right)^2}{(\lambda_i(k(\boldsymbol{X}, \boldsymbol{X})/n) + \rho)^2} \ .$$

By Proposition 5, we have

$$\mathbb{E}_D R_P(f_{\boldsymbol{X}, \boldsymbol{y}, t, \rho}) - R_P^* \geq \frac{\sigma^2}{n}\sum_{i=1}^n \mathbb{E}_{\boldsymbol{X}} S_i(\boldsymbol{X}) \geq \frac{\sigma^2}{n}\sum_{i \in \mathcal{I}(n)} \mathbb{E}_{\boldsymbol{X}} S_i(\boldsymbol{X}) \ . \tag{F.4}$$

Since $S_i(\boldsymbol{X})$ is almost surely positive, we can focus on the case where (F.1) and (F.2) hold, which is true with probability $\geq 1/2$ by assumption for $i \in \mathcal{I}(n)$. Hence,

$$\mathbb{E}_{\boldsymbol{X}} S_i(\boldsymbol{X}) \geq \frac{1}{2}\frac{\lambda_i^2/C \cdot (1 - e^{-2t(\lambda_i/C+\rho)})^2}{(C\lambda_i + \rho)^2} \overset{\text{(F.2)}}{\geq} \frac{1}{2}\frac{\lambda_i^2/C}{(C\lambda_i + \rho)^2(1 + (2t(\lambda_i/C + \rho))^{-1})^2} \ .$$

We can upper-bound the denominator, using $C \geq 1$, as

$$(C\lambda_i + \rho)\left(1 + \frac{1}{2t(\lambda_i/C + \rho)}\right) \leq C\lambda_i + \rho + \frac{C^2\lambda_i + C\rho}{(\lambda_i + C\rho)t} \leq C\lambda_i + \rho + \frac{C^2\lambda_i + C^3\rho}{(\lambda_i + C\rho)t}$$

$$\leq C^2\left(\lambda_i + \rho + t^{-1}\right) \ ,$$

which yields

$$\mathbb{E}_{\boldsymbol{X}} S_i(\boldsymbol{X}) \geq \frac{1}{2C^5}\frac{\lambda_i^2}{(\lambda_i + \rho + t^{-1})^2} = \frac{1}{2C^5}\frac{1}{\left(1 + \frac{\rho + t^{-1}}{\lambda_i}\right)^2} \overset{\text{(F.3)}}{\geq} \frac{1}{4C^5}\frac{1}{1 + \left(\frac{\rho + t^{-1}}{\lambda_i}\right)^2}$$

$$\geq \frac{1}{4C^5}\frac{1}{1 + \left(\frac{\rho + t^{-1}}{c}i^q\right)^2} \ . \tag{F.5}$$

**Step 3: Analyzing the sum.** We want to analyze the behavior of the sum

$$S(\beta) := \sum_{i \in \mathcal{I}(n)} \frac{1}{1 + (\beta i^q)^2}$$

for $\beta := \frac{\rho + t^{-1}}{c} > 0$. We first obtain the trivial bound

$$S(\beta) \geq |\mathcal{I}(n)|\frac{1}{1 + (\beta n^q)^2} \ .$$

Moreover, we can bound

$$S(\beta) \geq \sum_{i=1}^{I(n)} \frac{1}{1 + (\beta i^q)^2}$$

and distinguish three cases:

(a) If $\beta \geq 1$, we bound

$$S(\beta) \geq \sum_{i=1}^{I(n)} \frac{1}{2(\beta i^q)^2} \geq \frac{1}{2\beta^2} \ .$$

(b) If $\beta \in (I(n)^{-q}, 1)$, we observe that

$$J(\beta) := \lfloor \beta^{-1/q} \rfloor \geq \lceil \beta^{-1/q} \rceil - 1 \geq \frac{1}{2} \lceil \beta^{-1/q} \rceil \geq \frac{\beta^{-1/q}}{2}$$

and therefore

$$S(\beta) \geq \sum_{i=1}^{J(\beta)} \frac{1}{1 + (\beta i^q)^2} \geq \sum_{i=1}^{J(\beta)} \frac{1}{1+1} = \frac{J(\beta)}{2} \geq \frac{\beta^{-1/q}}{4} \ .$$

(c) If $\beta \in (0, I(n)^{-q}]$, we similarly find that

$$S(\beta) \geq \sum_{i=1}^{I(n)} \frac{1}{1+1} = \frac{I(n)}{2} \ .$$

Moreover, there is an absolute constant $c_1 > 0$ such that for any $\beta > 0$,

$$S(\beta) \geq c_1 \min\{\beta^{-2}, \beta^{-1/q}, I(n)\} \ , \tag{F.6}$$

because

(a) $\beta^{-2} = \min\{\beta^{-2}, \beta^{-1/q}, I(n)\}$ for $\beta \geq 1$,
(b) $\beta^{-1/q} = \min\{\beta^{-2}, \beta^{-1/q}, I(n)\}$ for $\beta \in (I(n)^{-q}, 1)$, and
(c) $I(n) = \min\{\beta^{-2}, \beta^{-1/q}, I(n)\}$ for $\beta \in (0, I(n)^{-q}]$.

**Step 4: Putting it together.** Combining the trivial bound in Step 3 with Eq. (F.4) and Eq. (F.5), we obtain

$$\mathbb{E}_D R_P(f_{\boldsymbol{X}, \boldsymbol{y}, t, \rho}) - R_P^* \geq \frac{\sigma^2}{n} \sum_{i \in \mathcal{I}(n)} \mathbb{E}_{\boldsymbol{X}} S_i(\boldsymbol{X}) \geq \frac{\sigma^2}{n} \cdot \frac{1}{4C^5} S(\beta)$$

$$\geq c' \sigma^2 \frac{1}{1 + (\rho + t^{-1}) n^q} \cdot \frac{|\mathcal{I}(n)|}{n} \tag{F.7}$$

for a suitable constant $c' > 0$ depending only on $c$ and $C$.

Moreover, from Eq. (F.6), we obtain

$$S(\beta) \geq \tilde{c}_1 \min\{\beta^{-2}, \beta^{-1/q}, I(n)\} \geq \tilde{c}'' \min\{(\rho + t^{-1})^{-2}, (\rho + t^{-1})^{-1/q}, I(n)\}$$

for a suitable constant $\tilde{c}'' > 0$ depending only on $c$. Again, (F.4) and (F.5) yield

$$\mathbb{E}_D R_P(f_{\boldsymbol{X}, \boldsymbol{y}, t, \rho}) - R_P^* \geq \frac{\sigma^2}{n} \cdot \frac{1}{4C^5} S(\beta)$$

$$\geq \frac{\tilde{c}''}{4C^5} \sigma^2 \min \left\{ \frac{(\rho + t^{-1})^{-2}}{n}, \frac{(\rho + t^{-1})^{-1/q}}{n}, \frac{I(n)}{n} \right\} \ . \qquad \square$$

## F.2  Equivalences of norms and eigenvalues

Later, we will use concentration inequalities for kernel matrix eigenvalues proved for specific kernels, which we then want to transfer to other kernels with equivalent RKHSs. In this subsection, we show that this is possible.

**Definition F.4** (*C*-equivalence of matrices and norms). Let $n \geq 1$ and let $\boldsymbol{K}, \tilde{\boldsymbol{K}} \in \mathbb{R}^{n \times n}$ be symmetric. For $C \geq 1$, we say that $\boldsymbol{K}$ and $\tilde{\boldsymbol{K}}$ are *C*-equivalent if their ordered eigenvalues satisfy

$$C^{-1} \lambda_i(\boldsymbol{K}) \leq \lambda_i(\tilde{\boldsymbol{K}}) \leq C \lambda_i(\boldsymbol{K})$$

for all $i \in [n]$. Moreover, we say that two norms $\|\cdot\|_A, \|\cdot\|_B$ on a vector space $V$ are *C*-equivalent if

$$C^{-1} \|\boldsymbol{v}\|_A \leq \|\boldsymbol{v}\|_B \leq C \|\boldsymbol{v}\|_A$$

for all $\boldsymbol{v} \in V$. ◄

**Lemma F.5.** *Let $n \geq 1$ and let $\boldsymbol{K}, \tilde{\boldsymbol{K}} \in \mathbb{R}^{n \times n}$ be symmetric. Then, $\boldsymbol{K}$ and $\tilde{\boldsymbol{K}}$ are $C$-equivalent iff the Moore-Penrose pseudoinverses $\boldsymbol{K}^+$ and $\tilde{\boldsymbol{K}}^+$ are $C$-equivalent.*

*Proof.* This follows from the fact that if $\boldsymbol{K}$ has eigenvalues $\lambda_1, \ldots, \lambda_n$, then $\boldsymbol{K}^+$ has eigenvalues $1/\lambda_1, \ldots, 1/\lambda_n$, where we define $1/0 := 0$. (A detailed proof would be a bit technical due to the sorting of eigenvalues.) □

**Lemma F.6.** *Let $k : \mathcal{X} \times \mathcal{X} \to \mathbb{R}$ be a kernel on a set $\mathcal{X}$. Then, for any $\boldsymbol{y} \in \mathbb{R}^n$,*

$$\boldsymbol{y}^\top k(\boldsymbol{X}, \boldsymbol{X})^+ \boldsymbol{y} = \|f^*_{k,\boldsymbol{y}}\|^2_{\mathcal{H}_k} \,,$$

*where $\mathcal{H}_k$ is the RKHS associated with $k$ and $f^*_{k,\boldsymbol{y}}$ is the minimum-norm regression solution*

$$f^*_{k,\boldsymbol{y}} := \underset{f \in B}{\mathrm{argmin}} \|f\|^2_{\mathcal{H}_k},$$

$$B := \{f \in \mathcal{H}_k \mid \sum_{i=1}^n (f(\boldsymbol{x}_i) - y_i)^2 = \inf_{\tilde{f} \in \mathcal{H}_k} \sum_{i=1}^n (\tilde{f}(\boldsymbol{x}_i) - y_i)^2\} \,.$$

*Proof.* It is well-known that $f^*_{k,\boldsymbol{y}}(\boldsymbol{x}) = \sum_{i=1}^n \alpha_i k(\boldsymbol{x}, \boldsymbol{x}_i)$, where $\boldsymbol{\alpha} := \boldsymbol{K}^+ \boldsymbol{y}$ (see e.g. Rangamani et al., 2023). We then have

$$\|f^*_{k,\boldsymbol{y}}\|^2_{\mathcal{H}_k} = \left\langle \sum_{i=1}^n \alpha_i k(\boldsymbol{x}_i, \cdot), \sum_{j=1}^n \alpha_j k(\boldsymbol{x}_j, \cdot) \right\rangle_{\mathcal{H}_k} = \sum_{i=1}^n \sum_{j=1}^n \alpha_i \boldsymbol{K}_{ij} \alpha_j$$

$$= \boldsymbol{y}^\top \boldsymbol{K}^+ \boldsymbol{K} \boldsymbol{K}^+ \boldsymbol{y} = \boldsymbol{y}^\top \boldsymbol{K}^+ \boldsymbol{y} \,,$$

where the last step follows from a standard identity for the Moore-Penrose pseudoinverse (see e.g. Section 1.1.1 in Wang et al., 2018). □

**Lemma F.7.** *Let $\mathcal{H}_1$ and $\mathcal{H}_2$ be two RKHSs with $\mathcal{H}_1 \subset \mathcal{H}_2$. Then there exists a constant $C > 0$ such that $\|f\|_{\mathcal{H}_2} \leq C\|f\|_{\mathcal{H}_1}$.*

*Proof.* Let $I : \mathcal{H}_1 \to \mathcal{H}_2$ be the inclusion map, i.e. $I_h := h$ for all $h \in \mathcal{H}_1$. Obviously, $I$ is linear and we need to show that $I$ is bounded. To this end, let $(h_n)_{n \geq 1} \subset \mathcal{H}_1$ be a sequence such that there exist $h \in \mathcal{H}_1$ and $g \in \mathcal{H}_2$ with $h_n \to h$ in $\mathcal{H}_1$ and $Ih_n \to g$ in $\mathcal{H}_2$. This implies $h_n \to h$ pointwise and $h_n = Ih_n \to g$ pointwise, which in turn gives $h = g$. The closed graph theorem, see e.g. (Megginson, 1998, Theorem 1.6.11), then shows that $I$ is bounded. □

Applying Lemma F.7 twice shows that RKHSs $\mathcal{H}_1$ and $\mathcal{H}_2$ with $\mathcal{H}_1 = \mathcal{H}_2$ automatically have $C$-equivalent norms for a suitable constant $C \geq 1$. The following result investigates the corresponding kernels.

**Proposition F.8** (Equivalent kernels have equivalent kernel matrices). *Let $k, \tilde{k} : \mathcal{X} \times \mathcal{X} \to \mathbb{R}$ be kernels such that their RKHSs are equal as sets and the corresponding RKHS-norms are $C$-equivalent as defined in Definition F.4. Then, for any $n \geq 1$ and any $\boldsymbol{x}_1, \ldots, \boldsymbol{x}_n \in \mathcal{X}$, the corresponding kernel matrices $k(\boldsymbol{X}, \boldsymbol{X}), \tilde{k}(\boldsymbol{X}, \boldsymbol{X})$ are $C^2$-equivalent.*

*Proof.* Let $i \in [n]$. For $\boldsymbol{y} \in \mathbb{R}^n$ we have, using the notation of Lemma F.6:

$$\boldsymbol{y}^\top k(\boldsymbol{X}, \boldsymbol{X})^+ \boldsymbol{y} = \|f^*_{k,\boldsymbol{y}}\|^2_{\mathcal{H}_k} \geq C^{-2}\|f^*_{k,\boldsymbol{y}}\|^2_{\mathcal{H}_{\tilde{k}}} \geq C^{-2}\|f^*_{\tilde{k},\boldsymbol{y}}\|^2_{\mathcal{H}_{\tilde{k}}} = C^{-2}\boldsymbol{y}^\top \tilde{k}(\boldsymbol{X}, \boldsymbol{X})^+ \boldsymbol{y} \,.$$

Now, by the Courant-Fischer-Weyl theorem,

$$\lambda_i(k(\boldsymbol{X}, \boldsymbol{X})^+) = \sup_{V: \dim V = i} \inf_{y \in V: \|y\|_2 = 1} y^\top k(\boldsymbol{X}, \boldsymbol{X})^+ y$$

$$\geq C^{-2} \sup_{V: \dim V = i} \inf_{y \in V: \|y\|_2 = 1} y^\top \tilde{k}(\boldsymbol{X}, \boldsymbol{X})^+ y$$

$$= C^{-2} \lambda_i(\tilde{k}(\boldsymbol{X}, \boldsymbol{X})^+) \,.$$

By switching the roles of $k$ and $\tilde{k}$, we obtain that $k(\boldsymbol{X}, \boldsymbol{X})^+$ and $\tilde{k}(\boldsymbol{X}, \boldsymbol{X})^+$ are $C^2$-equivalent. By Lemma F.5 $k(\boldsymbol{X}, \boldsymbol{X})$ and $\tilde{k}(\boldsymbol{X}, \boldsymbol{X})$ are then also $C^2$-equivalent. □

To prove Theorem 6 for arbitrary input distributions $P_X$ with lower and upper bounded densities, we need the following theorem investigating the corresponding eigenvalues of the integral operator.

**Lemma F.9** (Integral operators for equivalent densities have equivalent eigenvalues). *Let $k : \mathcal{X} \times \mathcal{X} \to \mathbb{R}$ be a kernel and let $\mu, \nu$ be finite measures on $\mathcal{X}$ whose support is $\mathcal{X}$ such that $\nu$ has an lower and upper bounded density w.r.t. $\mu$. Then, $\lambda_i(T_{k,\nu}) = \Theta(\lambda_i(T_{k,\mu}))$.*

*Proof.* Let $p$ be such an upper bounded density, that is, $\mathrm{d}\nu = p\,\mathrm{d}\mu$ and there exist $c, C > 0$ such that $c \le p(\boldsymbol{x}) \le C$ for all $\boldsymbol{x} \in \mathcal{X}$. For $f \in L_2(\nu)$, we have

$$\|p \cdot f\|_{L_2(\mu)}^2 = \int f^2 p^2 \,\mathrm{d}\mu \le C \int f^2 p\,\mathrm{d}\mu = C \int f^2\,\mathrm{d}\nu = C\|f\|_{L_2(\nu)}^2 \,.$$

Hence, the linear operator

$$A : L_2(\nu) \to L_2(\mu), f \mapsto p \cdot f$$

is well-defined and continuous. It is also easily verified that $A$ is bijective. Moreover, we have

$$\langle Af, Af \rangle_{L_2(\mu)} = \int f^2 p^2 \,\mathrm{d}\mu \ge c \int f^2 p\,\mathrm{d}\mu = c \int f^2\,\mathrm{d}\nu = c\langle f, f \rangle_{L_2(\nu)} \,.$$

and

$$\langle f, T_{k,\nu}f \rangle_{L_2(\nu)} = \int\int p(\boldsymbol{x})f(\boldsymbol{x})k(\boldsymbol{x},\boldsymbol{x}')f(\boldsymbol{x}')p(\boldsymbol{x}')\,\mathrm{d}\mu(\boldsymbol{x})\,\mathrm{d}\mu(\boldsymbol{x}') = \langle Af, T_{k,\mu}Af \rangle_{L_2(\mu)} \,.$$

Since $T_{k,\mu}$ and $T_{k,\nu}$ are compact, self-adjoint, and positive, we can use the Courant-Fischer minmax principle for operators (see e.g. Bell, 2014) to obtain

$$
\begin{aligned}
\lambda_i(T_{k,\nu}) &= \max_{\substack{V \subseteq L_2(\nu) \text{ subspace} \\ \dim V = i}} \min_{f \in V \setminus \{0\}} \frac{\langle f, T_{k,\nu}f \rangle_{L_2(\nu)}}{\langle f, f \rangle_{L_2(\nu)}} \\
&\ge c \max_{\substack{V \subseteq L_2(\nu) \text{ subspace} \\ \dim V = i}} \min_{f \in V \setminus \{0\}} \frac{\langle Af, T_{k,\mu}Af \rangle_{L_2(\mu)}}{\langle Af, Af \rangle_{L_2(\mu)}} \\
&= c \max_{\substack{\tilde{V} \subseteq L_2(\nu) \text{ subspace} \\ \dim \tilde{V} = i}} \min_{g \in \tilde{V} \setminus \{0\}} \frac{\langle g, T_{k,\mu}g \rangle_{L_2(\mu)}}{\langle g, g \rangle_{L_2(\mu)}} \\
&= c\lambda_i(T_{k,\mu}) \,.
\end{aligned}
$$

Here, we have used that since $A$ is bijective, the subspaces $AV$ for $\dim(V) = i$ are exactly the $i$-dimensional subspaces of $L_2(\mu)$. Our calculation above shows that $\lambda_i(T_{k,\mu}) \le O(\lambda_i(T_{k,\nu}))$. Since $\mathrm{d}\mu = \frac{1}{p}\,\mathrm{d}\nu$ with the lower and upper bounded density $1/p$, we can reverse the roles of $\nu$ and $\mu$ to also obtain $\lambda_i(T_{k,\nu}) \le O(\lambda_i(T_{k,\mu}))$, which proves the claim. $\qquad\square$

**Lemma F.10** (Integral operators of equivalent kernels have equivalent eigenvalues). *Let $k, \tilde{k} : \mathcal{X} \times \mathcal{X} \to \mathbb{R}$ be bounded kernels with RKHSs $\mathcal{H}$ and $\tilde{\mathcal{H}}$ satisfying $\mathcal{H} = \tilde{\mathcal{H}}$ as sets. Moreover, let $C \ge 1$ be a constant such that the corresponding RKHS-norms are $C$-equivalent and let $\nu$ be a finite measure on $\mathcal{X}$. If there exist constants $q > 0$ and $c > 0$ with*

$$\lambda_i(T_{k,\nu}) \le ci^{-q}, \qquad i \ge 1,$$

*then we also have*

$$\lambda_i(T_{\tilde{k},\nu}) \le c \cdot C^2 \cdot K_q \cdot i^{-q}, \qquad i \ge 1,$$

*where $K_q > 0$ is a constant only depending on $q$.*

*Proof.* We follow the ideas outlined in (Steinwart, 2017, Section 3). To this end, let $I_{k,\nu} : \mathcal{H} \to L_2(\nu)$ be the embedding $h \mapsto [h]_\sim$, which is defined and compact since $k$ is bounded and $\nu$ is finite, see e.g. (Steinwart and Scovel, 2012, Lemma 2.3). We write $S_{k,\nu} := I_{k,\nu}^*$ for its adjoint, which in turn gives $I_{k,\nu} = S_{k,\nu}^*$. Then (Steinwart and Scovel, 2012, Lemma 2.2) shows $T_{k,\nu} = S_{k,\nu}^* \circ S_{k,\nu}$. We denote the $i$-th (dyadic) entropy number of $I_{k,\nu}$ by $\varepsilon_i(I_{k,\nu})$, see e.g. Carl and Stephani (1990), (Edmunds and Triebel, 1996, Chapter 1.3.1), or (Steinwart and Christmann, 2008, Chapter 6) for

a definition[3]. Moreover, we denote the $i$-approximation and singular numbers of $I_{k,\nu}$ by $a_i(I_{k,\nu})$, respectively $s_i(I_{k,\nu})$. Since $I_{k,\nu}$ compactly acts between Hilbert spaces, we then have $a_i(I_{k,\nu}) = s_i(I_{k,\nu})$, see e.g. (Pietsch, 1987, Chapter 2.11). This implies

$$\lambda_i(T_{k,\nu}) = a_i^2(I_{k,\nu}) \tag{F.8}$$

for all $i \geq 1$ by the very definition of singular numbers. Finally, analogous definitions and considerations are made for the kernel $\tilde{k}$. From $C^{-1}\|\cdot\|_{\mathcal{H}} \leq \|\cdot\|_{\tilde{\mathcal{H}}} \leq C\|\cdot\|_{\mathcal{H}}$ we then conclude that

$$C^{-1}\varepsilon_i(I_{k,\nu}) \leq \varepsilon_i(I_{\tilde{k},\nu}) \leq C\varepsilon_i(I_{k,\nu})\,, \qquad i \geq 1, \tag{F.9}$$

by the multiplicativity of entropy numbers, see e.g. (Edmunds and Triebel, 1996, Chapter 1.3.1).

Now, (F.8) and our eigenvalue assumption yield

$$a_i(I_{k,\nu}) \leq \sqrt{c} \cdot i^{-q/2}\,, \qquad i \geq 1\,,$$

and Carl's inequality, see e.g. (Carl and Stephani, 1990, Theorem 3.1.1) then gives

$$\varepsilon_i(I_{k,\nu}) \leq \sqrt{c} \cdot \tilde{K}_q \cdot i^{-q/2}\,, \qquad i \geq 1\,,$$

where $\tilde{K}_q$ is a constant only depending on $q$. By (Carl and Stephani, 1990, Inequality (3.0.9)) and (F.9) we then obtain

$$a_i(I_{\tilde{k},\nu}) \leq 2\varepsilon_i(I_{\tilde{k},\nu}) \leq 2C\varepsilon_i(I_{k,\nu}) \leq 2C\sqrt{c} \cdot \tilde{K}_q \cdot i^{-q/2}\,, \qquad i \geq 1\,.$$

Another application of (F.8) then yields the assertion for $K_q := 4\tilde{K}_q^2$. $\qquad\square$

## F.3 Kernel matrix eigenvalue bounds

For upper bounds on the eigenvalues of kernel matrices, we use the following result:

**Proposition F.11** (Kernel matrix eigenvalue upper bound in expectation)**.** *For $m \geq 1$, we have*

$$\mathbb{E}_{\boldsymbol{X}} \sum_{i=m}^{n} \lambda_i(k(\boldsymbol{X}, \boldsymbol{X})/n) \leq \sum_{i=m}^{\infty} \lambda_i(T_k)\,. \tag{F.10}$$

*Proof.* Theorem 7.29 in Steinwart and Christmann (2008) shows that

$$\mathbb{E}_{D \sim \mu^n} \sum_{i=m}^{\infty} \lambda_i(T_{k,D}) \leq \sum_{i=m}^{\infty} \lambda_i(T_{k,\mu})\,, \tag{F.11}$$

where $T_{k,\mu} : L_2(\mu) \to L_2(\mu), f \mapsto \int k(x,\cdot)f(x)\,\mathrm{d}\mu(x)$ is the integral operator corresponding to the measure $\mu$ and $T_{k,D}$ is the corresponding discrete version thereof. We set $\mu := P_X$ and need to show that $k(\boldsymbol{X}, \boldsymbol{X})/n$ has the same eigenvalues as $T_{k,D}$ if $D$ and $\mathcal{X}$ contain the same data points $\boldsymbol{x}_1, \ldots, \boldsymbol{x}_n$. Consider a fixed $D$. Then, we can write $T_{k,D}(f) = n^{-1}ABf$, where

$$A : \mathbb{R}^n \to L_2(D), \boldsymbol{v} \mapsto \sum_{i=1}^{n} v_i k(\boldsymbol{x}_i, \cdot)$$
$$B : L_2(D) \to \mathbb{R}^n, f \mapsto (f(\boldsymbol{x}_1), \ldots, f(\boldsymbol{x}_n))^{\top}\,.$$

Then, $k(\boldsymbol{X}, \boldsymbol{X})/n$ is the matrix representation of $n^{-1}BA$ with respect to the standard basis of $\mathbb{R}^n$. But $AB$ and $BA$ have the same non-zero eigenvalues, which means that

$$\sum_{i=m}^{n} \lambda_i(k(\boldsymbol{X}, \boldsymbol{X})/n) = \sum_{i=m}^{\infty} \lambda_i(T_{k,D})\,,$$

from which the claim follows. $\qquad\square$

---

[3]Usually, dyadic entropy numbers are denoted by $e_i(\cdot)$, but since this symbol is already used for eigenfunctions, we use $\varepsilon_i(\cdot)$ instead.

To obtain a lower bound, we want to leverage the lower bound by Buchholz (2022) for a certain radial basis function kernel with data generated from an open subset of $\mathbb{R}^d$. However, we want to consider different kernels and distributions on the whole sphere. The following theorem bridges the gap by going to subsets of the data on a sphere cap, projecting them to $\mathbb{R}^d$, and using the kernel equivalence results from Appendix F.2:

**Theorem F.12** (Kernel matrix eigenvalue lower bound for Sobolev kernels on the sphere). *Let $k$ be a kernel on $\mathbb{S}^d$ such that its RKHS $\mathcal{H}_k$ is equivalent to a Sobolev space $H^s(\mathbb{S}^d)$ with smoothness $s > d/2$. Moreover, let $P_X$ be a probability distribution on $\mathbb{S}^d$ with lower and upper bounded density. Let the rows of $\boldsymbol{X} \in \mathbb{R}^{n \times d}$ are drawn independently from $P_X$. Then, for $\varepsilon \in (0, 1/20)$, there exists a constant $c > 0$ and $n_0 \in \mathbb{N}$ such that for all $n \geq n_0$,*

$$\lambda_m(k(\boldsymbol{X}, \boldsymbol{X})/n) \geq cn^{-2s/d}$$

*holds with probability $\geq 4/5$ for all $m \in \mathbb{N}$ with $1 \leq m \leq (1 - 11\varepsilon)n$.*

*Proof.* We can choose a suitably large sphere cap $T$ such that $P_X(T) \geq 1 - \varepsilon$. Define the conditional distribution $P_T(\cdot) := P_X(\cdot|T)$. Out of the points $\boldsymbol{X} = (\boldsymbol{x}_1, \ldots, \boldsymbol{x}_n)$, we can consider the submatrix $\boldsymbol{X}_T = (\boldsymbol{x}_{i_1}, \ldots, \boldsymbol{x}_{i_N})^\top$ of the points lying in $T$. Conditioned on $N$, these points are i.i.d. samples from $P_T$. Moreover, by applying Markov's inequality to a Bernoulli distribution, we obtain $N \geq (1 - 10\varepsilon)n$ with probability $\geq 9/10$. We fix a value of $N \geq (1 - 10\varepsilon)n$ in the following and condition on it.

We denote the centered unit ball in $\mathbb{R}^d$ by $B_1(\mathbb{R}^d)$. Using a construction as in Lemma E.1, we can transport $k$ and $P_T$ from $T$ to the unit ball $B_1(\mathbb{R}^d)$ using a rescaled stereographic projection feature map $\phi$, such that we obtain a kernel $k_\phi$ and a distribution $P_\phi = (P_T)_\phi$ on $B_1(\mathbb{R}^d)$ that generate the same distribution of kernel matrices as $k$ with $P_T$, and such that $\mathcal{H}_{k_\phi} \cong H^s(B_1(\mathbb{R}^d))$. The rows of $\boldsymbol{X}_\phi := \phi(\boldsymbol{X}_T)$ are i.i.d. samples from $P_\phi$. Moreover, we know that $P_\phi$ has an lower and upper bounded density w.r.t. the Lebesgue measure on $B_1(\mathbb{R}^d)$.

In order to apply the results from Buchholz (2022), we define a translation-invariant reference kernel on $\mathbb{R}^d$ through the Fourier transform

$$\hat{k}_{\mathrm{ref}}(\xi) = (1 + |\xi|^2)^{-2s} \, ,$$

see Eq. (3) in Buchholz (2022). The RKHS of $k_{\mathrm{ref}}$ on $\mathbb{R}^d$ is equivalent to the Sobolev space $H^s(\mathbb{R}^d)$. Therefore, the RKHS of $k_{\mathrm{ref}}|_{B_1(\mathbb{R}^d), B_1(\mathbb{R}^d)}$ is $H^s(B_1(\mathbb{R}^d))$, cf. the remarks in Appendix B.1 and Lemma F.7.

Now, let $1 \leq m \leq (1 - 11\varepsilon)n$, which implies

$$1 \leq m \leq (1 - 11\varepsilon)n \leq (1 - \varepsilon)(1 - 10\varepsilon)n \leq (1 - \varepsilon)N \, .$$

We apply Theorem 12 by Buchholz (2022) with bandwidth $\gamma = 1$ and $\alpha = 2s$ to $\lambda_m$ and obtain with probability at least $1 - 2/N$:

$$\lambda_m(k_{\mathrm{ref}}(\boldsymbol{X}_\phi, \boldsymbol{X}_\phi))^{-1} \leq c_3 \left( \frac{N^{2(\alpha-d)/d}}{(N - m)^{(\alpha-d)/d}} + 1 \right) \leq c_3 \left( \frac{N^{2(\alpha-d)/d}}{(\varepsilon N)^{(\alpha-d)/d}} + 1 \right)$$
$$\leq c_4(n^{\alpha/d-1} + 1)$$

as long as $N$ is large enough such that $(1 - \varepsilon)N < N - 32 \ln(N)$, which is the case if $n$ is large enough. Here, the constant $c_3$ from Buchholz (2022) does not depend on $N$ or $m$, but only on $\alpha$, $d$, and the upper and lower bounds on the density, which in our case depend on $\varepsilon$ through the choice of $T$. Since $\alpha = 2s > d$, we have $n^{\alpha/d-1} > 1$ and therefore

$$\lambda_m(k_{\mathrm{ref}}(\boldsymbol{X}_\phi, \boldsymbol{X}_\phi)/n) \geq c_5 n^{-\alpha/d} = c_5 n^{-2s/d} \, .$$

Now, we want to translate this to the kernel $k$. Since the RKHSs of $k_\phi$ and $k_{\mathrm{ref}}$ on $B_1(\mathbb{R}^d)$ are both equivalent to $H^s(B_1(\mathbb{R}^d))$, the kernels themselves are $C$-equivalent for some constant $C \geq 1$ as defined in Definition F.4. Therefore, Proposition F.8 shows that the corresponding kernel matrices are $C^2$-equivalent, which implies

$$\lambda_m(k_{*,\phi}(\boldsymbol{X}_\phi, \boldsymbol{X}_\phi)/n) \geq c_5 C^{-2} n^{-2s/d} \, .$$

By Cauchy's interlacing theorem, we therefore have

$$\lambda_m(k_*(\boldsymbol{X}, \boldsymbol{X})/n) \geq \lambda_m(k_*(\boldsymbol{X}_T, \boldsymbol{X}_T)/n) = \lambda_m(k_{*,\phi}(\boldsymbol{X}_\phi, \boldsymbol{X}_\phi)/n) \geq c_5 C^{-2} n^{-2s/d} .$$

Denoting the event where $\lambda_m(k_*(\boldsymbol{X}, \boldsymbol{X})/n) \geq c_5 C^{-2} n^{-2s/d}$ by $A$, we thus have

$$
\begin{aligned}
P(A) &= P(A|N \geq (1-10\varepsilon)n)P(N \geq (1-10\varepsilon)n) \geq \frac{9}{10} P(A|N \geq (1-10\varepsilon)n) \\
&= \frac{9}{10} \sum_{\hat{N}=\lceil (1-10\varepsilon)n \rceil}^{n} P(N = \hat{N}|N \geq (1-10\varepsilon)n)P(A|N = \hat{N}) \\
&\geq \frac{9}{10} \sum_{\hat{N}=\lceil (1-10\varepsilon)n \rceil}^{n} P(N = \hat{N}|N \geq (1-10\varepsilon)n)(1-2/N) \\
&\geq \frac{9}{10} \left(1 - \frac{2}{(1-10\varepsilon)n}\right) \sum_{\hat{N}=\lceil (1-10\varepsilon)n \rceil}^{n} P(N = \hat{N}|N \geq (1-10\varepsilon)n) \\
&= \frac{9}{10} \left(1 - \frac{2}{(1-10\varepsilon)n}\right) \geq \frac{4}{5} ,
\end{aligned}
$$

where the last step holds for sufficiently large $n$. $\qquad\square$

### F.4 Spectral lower bound for dot-product kernels on the sphere

An application of the spectral generalization bound in Proposition 5 requires a lower bound on eigenvalues of the kernel matrix $k_*(\boldsymbol{X}, \boldsymbol{X})$. To achieve this, we need to understand the properties of the convolution kernel $k_*$. Since the eigenvalues of $T_{k_*, P_X}$ are the squared eigenvalues of $T_{k, P_X}$, one might hope that if $\mathcal{H}_k$ is equivalent to a Sobolev space $H^s$, then $\mathcal{H}_{k_*}$ is equivalent to a Sobolev space $H^{2s}$. Unfortunately, this is not the case in general, as $\mathcal{H}_{k_*}$ might be a smaller space that involves additional boundary conditions (Schaback, 2018). However, perhaps since the sphere is a manifold without boundary, the desired characterization of $\mathcal{H}_{k_*}$ holds for dot-product kernels on the sphere:

**Lemma F.13** (RKHS of convolution kernels). *Let $k$ be a dot-product kernel on $\mathbb{S}^d$ such that its RKHS $\mathcal{H}_k$ is equivalent to a Sobolev space $H^s(\mathbb{S}^d)$ with smoothness $s > d/2$, and let $P_X$ be a distribution on $\mathbb{S}^d$ with lower and upper bounded density. Then, the RKHS $\mathcal{H}_{k_*}$ of the kernel*

$$k_* : \mathbb{S}^d \times \mathbb{S}^d \to \mathbb{R}, k_*(\boldsymbol{x}, \boldsymbol{x}') := \int k(\boldsymbol{x}, \boldsymbol{x}'')k(\boldsymbol{x}'', \boldsymbol{x}') \, \mathrm{d}P_X(\boldsymbol{x}'')$$

*is equivalent to the Sobolev space $H^{2s}(\mathbb{S}^d)$.*

*Proof.* Define

$$k_{*,\mathrm{unif}}(\boldsymbol{x}, \boldsymbol{x}') = \int k(\boldsymbol{x}, \boldsymbol{x}'')k(\boldsymbol{x}'', \boldsymbol{x}') \, \mathrm{d}\mathcal{U}(\mathbb{S}^d)(\boldsymbol{x}'') .$$

For the corresponding integral operator, we have

$$T_{k_{*,\mathrm{unif}}, \mathcal{U}(\mathbb{S}^d)} = T^2_{k, \mathcal{U}(\mathbb{S}^d)} .$$

This means that the corresponding eigenvalues are the squares of the eigenvalues of the corresponding integral operator of $k$. Especially, we obtain the Mercer representations

$$k(\boldsymbol{x}, \boldsymbol{x}') = \sum_{l=0}^{\infty} \mu_l \sum_{i=1}^{N_{l,d}} Y_{l,i}(\boldsymbol{x})Y_{l,i}(\boldsymbol{x}') ,$$

$$k_{*,\mathrm{unif}}(\boldsymbol{x}, \boldsymbol{x}') = \sum_{l=0}^{\infty} \mu_l^2 \sum_{i=1}^{N_{l,d}} Y_{l,i}(\boldsymbol{x})Y_{l,i}(\boldsymbol{x}') ,$$

where Lemma B.1 yields $\mu_l = \Theta((l+1)^{-2s})$, hence $\mu_l^2 = \Theta((l+1)^{-4s})$ and hence $\mathcal{H}_{k_{*,\mathrm{unif}}} \cong H^{2s}(\mathbb{S}^d)$.

Next, we show the equality of the ranges of the integral operators:

$$R(T_{k,\mathcal{U}(\mathbb{S}^d)}) = R(T_{k,P_X}) .$$

Let $p_X$ be a density of $P_X$ w.r.t. the uniform distribution $\mathcal{U}(\mathbb{S}^d)$. If $f \in R(T_{k,\mathcal{U}(\mathbb{S}^d)})$, there exists $g \in L_2(\mathcal{U}(\mathbb{S}^d))$ with $f = T_{k,\mathcal{U}(\mathbb{S}^d)}g$. But then, since $p_X$ is lower bounded, we have $g/p_X \in L_2(P_X)$ and therefore

$$f = T_{k,P_X}(g/p_X) \in R(T_{k,P_X}) .$$

An analogous argument shows that $R(T_{k,P_X}) \subseteq R(T_{k,\mathcal{U}(\mathbb{S}^d)})$ since $p_X$ is upper bounded.

The equality of the ranges yields for the RKHSs (as sets)

$$\mathcal{H}_{k_*,\mathrm{unif}} = R(T_{k,\mathcal{U}(\mathbb{S}^d)}) = R(T_{k,P_X}) = \mathcal{H}_{k_*} ,$$

Applying Lemma F.7 twice then shows $\mathcal{H}_{k_*} \cong H^{2s}(\mathbb{S}^d)$. $\qquad\square$

**Theorem 6 (Inconsistency for Sobolev dot-product kernels on the sphere).** *Let $k$ be a dot-product kernel on $\mathbb{S}^d$, i.e., a kernel of the form $k(\boldsymbol{x}, \boldsymbol{x}') = \kappa(\langle \boldsymbol{x}, \boldsymbol{x}' \rangle)$, such that its RKHS $\mathcal{H}_k$ is equivalent to a Sobolev space $H^s(\mathbb{S}^d)$, $s > d/2$. Moreover, let $P$ be a distribution on $\mathbb{S}^d \times \mathbb{R}$ such that $P_X$ has a lower and upper bounded density w.r.t. the uniform distribution $\mathcal{U}(\mathbb{S}^d)$, and such that $\mathrm{Var}(y|\boldsymbol{x}) \geq \sigma^2 > 0$ for $P_X$-almost all $\boldsymbol{x} \in \mathbb{S}^d$. Then, for every $C > 0$, there exists $c > 0$ independent of $\sigma^2$ such that for all $n \geq 1$, $t \in (C^{-1}n^{2s/d}, \infty]$, and $\rho \in [0, Cn^{-2s/d})$, the expected excess risk satisfies*

$$\mathbb{E}_D R_P(f_{t,\rho}) - R_P^* \geq c\sigma^2 > 0 .$$

*Proof.* **Step 0: Preparation.** Since the Sobolev space $H^{2s}(\mathbb{S}^d)$ is dense in the space of continuous functions $\mathbb{S}^d \to \mathbb{R}$, the kernel $k$ is universal. Applying (Steinwart and Christmann, 2008, Corollary 5.29 and Corollary 5.34) for the least squares loss thus shows that $k$ is strictly positive definite. If we have mutually distinct $\boldsymbol{x}_1, \ldots, \boldsymbol{x}_n$, the corresponding Gram matrix $k((\boldsymbol{x}_i, \boldsymbol{x}_j))_{i,j=1}^n$ is therefore invertible. Now, our assumptions on $P$ guarantee that $\boldsymbol{X}$ consists almost surely of mutually distinct observations, and therefore $k(\boldsymbol{X}, \boldsymbol{X})$ is almost surely invertible.

By Proposition 5, we know that

$$\mathbb{E}_D \mathcal{R}_P(f_{\boldsymbol{X},\boldsymbol{y},t,\rho}) - \mathcal{R}_P^* \geq \frac{\sigma^2}{n} \sum_{i=1}^{n} \mathbb{E}_{\boldsymbol{X}} \frac{\lambda_i(k_*(\boldsymbol{X},\boldsymbol{X})/n)\left(1 - e^{-2t(\lambda_i(k(\boldsymbol{X},\boldsymbol{X})/n)+\rho)}\right)^2}{(\lambda_i(k(\boldsymbol{X},\boldsymbol{X})/n) + \rho)^2}$$

$$\geq \frac{\sigma^2}{n} \sum_{i=1}^{n} \mathbb{E}_{\boldsymbol{X}} \frac{\lambda_i(k_*(\boldsymbol{X},\boldsymbol{X})/n)\left(1 - e^{-2C^{-1}n^{2s/d}(\lambda_i(k(\boldsymbol{X},\boldsymbol{X})/n)+0)}\right)^2}{(\lambda_i(k(\boldsymbol{X},\boldsymbol{X})/n) + Cn^{-2s/d})^2}$$

$$\geq c_n \sigma^2$$

for a suitable constant $c_n > 0$ depending on $n$ but not on $\sigma^2, t, \rho$, since the kernel matrix eigenvalues are nonzero almost surely. It is therefore sufficient to show the desired statement (with $c$ independent of $n, \sigma^2, t, \rho$) for sufficiently large $n$.

In the following, we assume $n \geq 40$ and set $\varepsilon := 1/100$.

**Step 1: Eigenvalue decay for the integral operator.** From Lemma B.1, we know that

$$\lambda_i(T_{k,\mathcal{U}(\mathbb{S}^d)}) = \Theta(i^{-2s/d}) .$$

Therefore, by Lemma F.9, we know that

$$\lambda_i(T_{k,P_X}) = \Theta(i^{-2s/d}) .$$

**Step 2: Eigenvalue upper bound.** Next, we want to upper-bound suitable eigenvalues of the form $\lambda_i(k(\boldsymbol{X}, \boldsymbol{X})/n)$ using Proposition F.11. Using Step 1, we derive

$$\sum_{i=m}^{\infty} \lambda_i(T_{k,P_X}) \leq C_1 \sum_{i=m}^{\infty} i^{-2s/d} \leq C_2 \int_m^{\infty} x^{-2s/d} \, \mathrm{d}x = C_3 m^{1-2s/d}$$

with constants independent of $m \geq 1$. For sufficiently large $n$, we can choose $m \in \mathbb{N}_{\geq 1}$ such that $\varepsilon n \leq m \leq 2\varepsilon n$. Then, Proposition F.11 yields

$$\mathbb{E}_{\boldsymbol{X}} \sum_{i=m}^{n} \lambda_i(k(\boldsymbol{X}, \boldsymbol{X})/n) \leq \sum_{i=m}^{\infty} \lambda_i(T_k) \leq C_3 m^{1-2s/d} \leq C_4 n^{1-2s/d} .$$

Since $\mathbb{E}_{\boldsymbol{X}} \lambda_i(k(\boldsymbol{X}, \boldsymbol{X})/n)$ is decreasing with $i$, we have for $i \geq 4\varepsilon n \geq 2m$:

$$\mathbb{E}_{\boldsymbol{X}} \lambda_i(k(\boldsymbol{X}, \boldsymbol{X})/n) \leq C_4 n^{1-2s/d}/m \leq C_5 n^{-2s/d} \leq C_6 \lambda_i(T_{k,P_X}) .$$

**Step 3: Eigenvalue lower bounds.** From Lemma F.13, we know that $\mathcal{H}_{k_*} \cong H^{2s}(\mathbb{S}^d)$. Therefore, we can apply Lemma F.13 to both $k$ and $k_*$ and obtain for sufficiently large $n$ and suitable constants $c_1, c_2 > 0$ that

$$\lambda_i(k(\boldsymbol{X}, \boldsymbol{X})/n) \geq c_1 n^{-2s/d}$$
$$\lambda_i(k_*(\boldsymbol{X}, \boldsymbol{X})/n) \geq c_2 n^{-4s/d}$$

individually hold with probability $\geq 4/5$ for all $i \in \mathbb{N}$ with $1 \leq i \leq (1 - 11\varepsilon)n$. By the union bound, both bounds hold at the same time with probability $\geq 3/5$.

**Step 4: Final result.** Now, using the value of $m$ from Step 2, consider an index $i$ with $2m \leq i \leq (1 - 11\varepsilon)n$. Since $2m \leq 4\varepsilon n$ and $\varepsilon = 1/100$, there are at least $n/2$ such indices. By combining Step 3 and Step 1, we have

$$\lambda_i(k(\boldsymbol{X}, \boldsymbol{X})/n) \geq c_3 \lambda_i(T_{k,P_X})$$
$$\lambda_i(k_*(\boldsymbol{X}, \boldsymbol{X})/n) \geq c_4 \lambda_i(T_{k,P_X})^2$$

with probability $\geq 3/5$. By applying Markov's inequality to Step 2, we obtain

$$\lambda_i(k(\boldsymbol{X}, \boldsymbol{X})/n) \leq 10 C_6 \lambda_i(T_{k,P_X})$$

with probability $\geq 9/10$. Therefore, by the union bound, all three inequalities hold simultaneously with probability $\geq 1/2$. Moreover, for $q = 2s/d$, we have $\lambda_i(T_{k,P_X}) \geq c_5 i^{-q}$ by Step 1. We can thus apply the first lower bound from Theorem F.2 to obtain

$$\begin{aligned}
\mathbb{E}_D R_P(f_{\boldsymbol{X}, \boldsymbol{y}, t, \rho}) - R_P^* &\geq c'\sigma^2 \frac{1}{1 + (\rho + t^{-1})n^{2s/d}} \cdot \frac{|\mathcal{I}(n)|}{n} \\
&\geq c'\sigma^2 \frac{1}{1 + (Cn^{-2s/d} + Cn^{-2s/d})n^{2s/d}} \cdot \frac{n/2}{n} \\
&= \frac{c'}{2 + 2C}\sigma^2 .
\end{aligned}$$ $\qquad \square$

# G    Proof of Theorem 8

Here we denote the solution of kernel ridge regression on $D$ with the kernel function $k$ and regularization parameter $\rho > 0$ as

$$\hat{f}_\rho^k(\boldsymbol{x}) = k(\boldsymbol{x}, \boldsymbol{X})\left(k(\boldsymbol{X}, \boldsymbol{X}) + \rho \boldsymbol{I}\right)^{-1} \boldsymbol{y},$$

and write $\hat{f}_0^k(\boldsymbol{x}) = k(\boldsymbol{x}, \boldsymbol{X})k(\boldsymbol{X}, \boldsymbol{X})^+ \boldsymbol{y}$ for the minimum-norm interpolant in the RKHS of $k$.

While Theorem 1 states that overfitting kernel ridge regression using Sobolev kernels is always inconsistent as long as the derivatives remain bounded by the derivatives of the minimum-norm interpolant of the fixed kernel (Assumption (N)), here we show that consistency over a large class of distributions is achievable by designing a kernel sequence, which can have Sobolev RKHS, that consists of a smooth component for generalization and a spiky component for interpolation.

Recall that $\tilde{k}$ denotes any universal kernel function for the smooth component, and $\check{k}_\gamma$ denotes the kernel function of the spiky component with bandwidth $\gamma$. Then we define the $\rho$-*regularized spiky-smooth kernel with spike bandwidth* $\gamma$ as

$$k_{\rho,\gamma}(\boldsymbol{x}, \boldsymbol{x}') = \tilde{k}(\boldsymbol{x}, \boldsymbol{x}') + \rho \cdot \check{k}_\gamma(\boldsymbol{x}, \boldsymbol{x}').$$

Let $B_t(\boldsymbol{x}) := \{\boldsymbol{y} \in \mathbb{R}^d \mid |\boldsymbol{x} - \boldsymbol{y}| \leq t\}$ denote the Euclidean ball of radius $t \geq 0$ around $\boldsymbol{x} \in \mathbb{R}^d$.

(D2) There exists a constant $\beta_X > 0$ and a continuous function $\phi : [0, \infty) \to [0, 1]$ with $\phi(0) = 0$ such that $P_X(B_t(\boldsymbol{x})) \leq \phi(t) = O(t^{\beta_X})$ for all $\boldsymbol{x} \in \Omega$ and all $t \geq 0$.

The kernel $\check{k}_\gamma$ of the spiky component should fulfill the following weak assumption on its decay behaviour. For example, Laplace, Matérn, and Gaussian kernels all fulfill Assumption (SK).

(SK) There exists a function $\varepsilon : (0, \infty) \times [0, \infty) \to [0, 1]$ such that for any bandwidth $\gamma > 0$ and any $\delta > 0$ it holds that
   (i) $\varepsilon(\gamma, 0) = 1$,
   (ii) $\varepsilon(\gamma, \delta)$ is monotonically increasing in $\gamma$,
   (iii) For all $\boldsymbol{x}, \boldsymbol{y} \in \Omega$, if $|\boldsymbol{x} - \boldsymbol{y}| \geq \delta$ then $|\check{k}_\gamma(\boldsymbol{x}, \boldsymbol{y})| \leq \varepsilon(\gamma, \delta)$,
   (iv) For any rates $\beta_X, \beta_k > 0$ there exists a rate $\beta_\gamma > 0$ such that, if $\delta_n = \Omega(n^{-\beta_X})$ and $\gamma_n = O(n^{-\beta_\gamma})$, then $\varepsilon(\gamma_n, \delta_n) = O(n^{-\beta_k})$.

**Theorem G.1 (Consistency of spiky-smooth ridgeless kernel regression).** *Assume that the training set $D$ consists of $n$ i.i.d. pairs $(\boldsymbol{x}, y) \sim P$ such that the marginal $P_X$ fulfills (D2) and $\mathbb{E}y^2 < \infty$. Let the kernel components satisfy:*

- *$\tilde{k}$ denotes an arbitrary universal kernel, and $\rho_n \to 0$ and $n\rho_n^4 \to \infty$.*
- *$\check{k}_{\gamma_n}$ denotes a kernel function that fulfills Assumption (SK) with a sequence of positive bandwidths $(\gamma_n)$ fulfilling $\gamma_n = O(\exp(-\beta n))$ for some arbitrary $\beta > 0$.*

*Then the minimum-norm interpolant of the $\rho_n$-regularized spiky-smooth kernel sequence $k_n := k_{\rho_n, \gamma_n}$ is consistent for $P$.*

**Remark G.2 (Spike bandwidth scaling).** Under stronger assumptions on $\phi$ and $\varepsilon$ in assumptions (D2) and (SK), the spike bandwidths $\gamma_n$ can be chosen to converge to 0 at a much slower rate. For example, if we choose $\check{k}_\gamma$ to be the Laplace kernel, choosing bandwidths $0 < \gamma_n \leq \frac{\delta}{\beta \ln n}$ yields, for separated points $|\boldsymbol{x} - \boldsymbol{y}| \geq \delta$,

$$\check{k}_{\gamma_n}(\boldsymbol{x}, \boldsymbol{y}) \leq \exp\left(-\frac{\delta}{\gamma_n}\right) \leq n^{-\beta}.$$

For probability measures with upper bounded Lebesgue density, we can choose $\delta_n = n^{-\frac{2+\alpha}{d}}$ and $\beta = \frac{9}{4} + \frac{\alpha}{2}$ for consistency or $\beta = \frac{11}{4} + \frac{\alpha}{2}$ for optimal convergence rates, for any fixed $\alpha > 0$, in the proof of Theorem 8. Hence the Laplace kernel only requires a slow bandwidth decay rate of $\gamma_n = \Omega\left(\frac{n^{-\frac{2+\alpha}{d}}}{\alpha \ln(n)}\right)$, where $\alpha > 0$ arbitrary. For the Gaussian kernel an analogous argument yields $\gamma_n = \Omega\left(\frac{n^{-\frac{4+2\alpha}{d}}}{\alpha \ln(n)}\right)$. The larger the dimension $d$, the slower the required bandwidth decay. ◄

**Remark G.3 (Generalizations).** If one does not care about continuous kernels, one could simply take a Dirac kernel as the spike and then obtain consistency for all atom-free $P_X$. However, we need a continuous kernel to be able to translate it to an activation function for the NTK. Beyond kernel regression, the spike component $\check{k}_\gamma$ does not even need to be a kernel, it just needs to fulfill Assumption (SK) or a similar decay criterion. Then one could still use the 'quasi minimum-norm estimator' $\boldsymbol{x} \mapsto (\tilde{k} + \rho_n \check{k}_{\gamma_n})(\boldsymbol{x}, \boldsymbol{X}) \cdot (\tilde{\boldsymbol{K}} + \rho_n \check{\boldsymbol{K}}_{\gamma_n})^+ \boldsymbol{y}$. ◄

**Remark G.4 (Consistency with a single kernel function).** Without resorting to kernel sequences as we do, there seems to be no rigorous proof showing that ridgeless kernel regression can be consistent in fixed dimension. In future work, can an analytical expression of such a kernel be found? According to the semi-rigorous results in Mallinar et al. (2022) a spectral decay like $\lambda_k = \Theta(k^{-1} \cdot \log^\alpha(k))$, $\alpha > 1$ could lead to such a kernel. ◄

*Proof of Theorem G.1.* Given any universal kernel, (Steinwart, 2001, Theorem 3.11 or Example 4.6) implies universal consistency of kernel ridge regression if $\rho_n \to 0$ and $n\rho_n^4 \to \infty$. Hence, for any $\varepsilon > 0$ it holds that

$$\lim_{n \to \infty} P^n\left(D \in (\mathbb{R}^d \times \mathbb{R})^n \mid R_P(\hat{f}_{\rho_n}^{\tilde{k}}) - R_P(f_P^*) = \mathbb{E}_{\boldsymbol{x}}(\hat{f}_{\rho_n}^{\tilde{k}}(\boldsymbol{x}) - f_P^*(\boldsymbol{x}))^2 \geq (\varepsilon/2)^2\right) = 0.$$

Due to the triangle inequality in $L_2(P_X)$, we know

$$R_P(\hat{f}_0^{k_n}) - R_P(f_P^*) = \mathbb{E}_{\boldsymbol{x}}(\hat{f}_0^{k_n}(\boldsymbol{x}) - f_P^*(\boldsymbol{x}))^2$$

$$\leq \left( \left( \mathbb{E}_{\boldsymbol{x}}(\hat{f}_0^{k_n}(\boldsymbol{x}) - \hat{f}_{\rho_n}^{\tilde{k}}(\boldsymbol{x}))^2 \right)^{1/2} + \left( \mathbb{E}_{\boldsymbol{x}}(\hat{f}_{\rho_n}^{\tilde{k}}(\boldsymbol{x}) - f_P^*(\boldsymbol{x}))^2 \right)^{1/2} \right)^2.$$

It is left to show that $k_n$ fulfills

$$\lim_{n\to\infty} P^n \left( D \in (\mathbb{R}^d \times \mathbb{R})^n \mid \mathbb{E}_{\boldsymbol{x}}(\hat{f}_0^{k_n}(\boldsymbol{x}) - \hat{f}_{\rho_n}^{\tilde{k}}(\boldsymbol{x}))^2 \geq (\varepsilon/2)^2 \right) = 0.$$

For this purpose we decompose the above difference into the difference of $\check{\boldsymbol{K}}_{\gamma_n} := \check{k}_{\gamma_n}(\boldsymbol{X}, \boldsymbol{X})$ and $\boldsymbol{I}_n$ and a remainder term depending on $\check{k}_{\gamma_n}$. We denote the 2-operator norm by $\|\cdot\|$ and the Euclidean norm in $\mathbb{R}^n$ by $|\cdot|$. For any $\boldsymbol{x} \in \mathbb{R}^d$ it holds that

$$|\hat{f}_0^k(\boldsymbol{x}) - \hat{f}_{\rho_n}^{\tilde{k}}(\boldsymbol{x})| \leq \left| (\tilde{k} + \rho_n \check{k}_{\gamma_n})(\boldsymbol{x}, \boldsymbol{X}) \cdot (\tilde{\boldsymbol{K}} + \rho_n \check{\boldsymbol{K}}_{\gamma_n})^{-1} \boldsymbol{y} - \tilde{k}(\boldsymbol{x}, \boldsymbol{X}) \cdot (\tilde{\boldsymbol{K}} + \rho_n \boldsymbol{I}_n)^{-1} \boldsymbol{y} \right|$$

$$\leq \left| \tilde{k}(\boldsymbol{x}, \boldsymbol{X}) \left( (\tilde{\boldsymbol{K}} + \rho_n \check{\boldsymbol{K}}_{\gamma_n})^{-1} - (\tilde{\boldsymbol{K}} + \rho_n \boldsymbol{I}_n)^{-1} \right) \boldsymbol{y} \right|$$

$$+ \rho_n \cdot \left| \check{k}_{\gamma_n}(\boldsymbol{x}, \boldsymbol{X})(\tilde{\boldsymbol{K}} + \rho_n \check{\boldsymbol{K}}_{\gamma_n})^{-1} \boldsymbol{y} \right|$$

$$\leq \|\tilde{k}(\boldsymbol{x}, \boldsymbol{X})\| \cdot \left\| (\tilde{\boldsymbol{K}} + \rho_n \check{\boldsymbol{K}}_{\gamma_n})^{-1} - (\tilde{\boldsymbol{K}} + \rho_n \boldsymbol{I}_n)^{-1} \right\| \cdot |\boldsymbol{y}|$$

$$+ \rho_n \cdot \|\check{k}_{\gamma_n}(\boldsymbol{x}, \boldsymbol{X})\| \cdot \|(\tilde{\boldsymbol{K}} + \rho_n \check{\boldsymbol{K}}_{\gamma_n})^{-1}\| \cdot |\boldsymbol{y}|.$$

Consequently we get

$$\mathbb{E}_{\boldsymbol{x}}(\hat{f}_0^k(\boldsymbol{x}) - \hat{f}_{\rho_n}^{\tilde{k}}(\boldsymbol{x}))^2 \leq 2 \, \mathbb{E}_{\boldsymbol{x}} \|\tilde{k}(\boldsymbol{x}, \boldsymbol{X})\|^2 \cdot \left\| (\tilde{\boldsymbol{K}} + \rho_n \check{\boldsymbol{K}}_{\gamma_n})^{-1} - (\tilde{\boldsymbol{K}} + \rho_n \boldsymbol{I}_n)^{-1} \right\|^2 \cdot |\boldsymbol{y}|^2 \quad \text{(G.1)}$$

$$+ 2 \, \rho_n^2 \cdot \mathbb{E}_{\boldsymbol{x}} \|\check{k}_{\gamma_n}(\boldsymbol{x}, \boldsymbol{X})\|^2 \cdot \|(\tilde{\boldsymbol{K}} + \rho_n \check{\boldsymbol{K}}_{\gamma_n})^{-1}\|^2 \cdot |\boldsymbol{y}|^2. \quad \text{(G.2)}$$

We now bound the individual terms in Eq. (G.1) and (G.2). To this end, fix any $\alpha > 0$.

**Bounding Eq. (G.1):**

Since we assumed $y_i$ i.i.d. and $\mathbb{E}y_1^2 < \infty$, the Markov inequality implies, with $b_n = \mathbb{E}y_1^2 \cdot n^\alpha$,

$$P(|\boldsymbol{y}|^2 \geq b_n n) \leq \frac{\mathbb{E}y_1^2}{b_n} = n^{-\alpha}.$$

Stated differently, with probability at least $1 - n^{-\alpha}$ it holds that $|\boldsymbol{y}|^2 \leq \mathbb{E}y_1^2 \cdot n^{1+\alpha}$.

In order to bound the spectrum of $\check{\boldsymbol{K}}_{\gamma_n}$, Lemma G.7 implies that there exists a positive sequence $\delta_\alpha(n) = n^{-\frac{2+\alpha}{\beta_X}}$ such that with probability at least $1 - O(n^{-\alpha})$ it holds that

$$\min_{i,j\in[n]:i\neq j} |\boldsymbol{x}_i - \boldsymbol{x}_j| \geq \delta_\alpha(n).$$

Since $(\gamma_n)$ fulfills $\gamma_n = O(n^{-\beta_\gamma})$ for any $\beta_\gamma > 0$, by Assumption (SK) there exists a sequence $\varepsilon_n = o(\rho_n n^{-2-\frac{\alpha}{2}})$ such that $\varepsilon(\gamma_n, \delta_\alpha(n)) \leq \varepsilon_n$. Assumption (SK) further implies that whenever $\min_{i,j\in[n]:i\neq j} |\boldsymbol{x}_i - \boldsymbol{x}_j| \geq \delta_\alpha(n)$ it holds that $(\check{\boldsymbol{K}}_{\gamma_n})_{ii} = 1$ and $0 \leq (\check{\boldsymbol{K}}_{\gamma_n})_{ij} \leq \varepsilon(\gamma_n, \delta_\alpha(n)) \leq \varepsilon_n$ for $i \neq j$. Then Gershgorin's theorem (Gerschgorin, 1931) implies that for all eigenvalues of $\check{\boldsymbol{K}}_{\gamma_n}$

$$|\lambda_i(\check{\boldsymbol{K}}_{\gamma_n}) - 1| \leq (n-1)\varepsilon_n \text{ for all } i \in [n].$$

This in turn implies

$$\|\check{\boldsymbol{K}}_{\gamma_n} - \boldsymbol{I}_n\| \leq (n-1)\varepsilon_n, \qquad \lambda_{\max}(\check{\boldsymbol{K}}_{\gamma_n}) \leq 1 + (n-1)\varepsilon_n, \qquad \lambda_{\min}(\check{\boldsymbol{K}}_{\gamma_n}) \geq 1 - (n-1)\varepsilon_n.$$

Using $\|(\tilde{\boldsymbol{K}} + \rho_n \boldsymbol{I}_n)^{-1}\| \leq \frac{1}{\lambda_{\min}(\tilde{\boldsymbol{K}}) + \rho_n} \leq \rho_n^{-1}$ and $\|\check{\boldsymbol{K}}_{\gamma_n} - \boldsymbol{I}_n\| \leq (n-1)\varepsilon_n$, Lemma G.8 implies

$$\left\| (\tilde{\boldsymbol{K}} + \rho_n \check{\boldsymbol{K}}_{\gamma_n})^{-1} - (\tilde{\boldsymbol{K}} + \rho_n \boldsymbol{I}_n)^{-1} \right\| \leq \frac{\|(\tilde{\boldsymbol{K}} + \rho_n \boldsymbol{I}_n)^{-1}\|^2 \cdot \rho_n \|\check{\boldsymbol{K}}_{\gamma_n} - \boldsymbol{I}_n\|}{1 - \|(\tilde{\boldsymbol{K}} + \rho_n \boldsymbol{I}_n)^{-1}\| \cdot \rho_n \|\check{\boldsymbol{K}}_{\gamma_n} - \boldsymbol{I}_n\|}$$

$$\leq \frac{\rho_n^{-1}(n-1)\varepsilon_n}{1 - (n-1)\varepsilon_n}.$$

Using $|\tilde{k}(\boldsymbol{x}, \boldsymbol{X}_i)| \leq 1$ for all $i \in [n]$ yields the naive bound $\|\tilde{k}(\boldsymbol{x}, \boldsymbol{X})\|^2 \leq n$.

Combining all terms in Eq. (G.1) yields its convergence to 0 as the product satifies the rate $O(n^{4+\alpha}\rho_n^{-2}\varepsilon_n^2) = o(1)$ with probability at least $1 - 2n^{-\alpha}$.

**Bounding Eq. (G.2):**

The analysis below is restricted to the event of probability at least $1 - 2n^{-\alpha}$, on which the bound on Eq. (G.1) holds.

Since $(n-1)\varepsilon_n \to 0$, for any $C > 1$ it holds for $n$ large enough,

$$\rho_n \cdot \|(\tilde{\boldsymbol{K}} + \rho_n \check{\boldsymbol{K}}_{\gamma_n})^{-1}\| \leq \frac{\rho_n}{\lambda_{\min}(\tilde{\boldsymbol{K}}) + \rho_n(1 - (n-1)\varepsilon_n)} \leq \frac{1}{(1 - (n-1)\varepsilon_n)} \leq C.$$

Finally we show $\sup_{\boldsymbol{x}' \in \mathbb{R}^d} \mathbb{E}_{\boldsymbol{x}} \check{k}_{\gamma_n}(\boldsymbol{x}, \boldsymbol{x}')^2 \leq 2n^{-(2+\alpha)}$ for $n$ large enough.

Fix an arbitrary $\boldsymbol{x}' \in \mathbb{R}^d$. Then by construction of $\delta_\alpha(n)$ and $\varepsilon_n$ it holds that

$$\begin{aligned}
\mathbb{E}_{\boldsymbol{x}} \check{k}_{\gamma_n}(\boldsymbol{x}, \boldsymbol{x}')^2 &\leq 1 \cdot P_X(\{\boldsymbol{x} \in \mathbb{R}^d : \check{k}_{\gamma_n}(\boldsymbol{x}, \boldsymbol{x}')^2 \geq \varepsilon_n^2\}) + \varepsilon_n^2 \\
&\leq P_X(\{\boldsymbol{x} \in \mathbb{R}^d : |\boldsymbol{x} - \boldsymbol{x}'| < \delta_\alpha(n)\}) + \varepsilon_n^2 \\
&\leq \phi(\delta_\alpha(n)) + \varepsilon_n^2 \leq n^{-(2+\alpha)} + \varepsilon_n^2.
\end{aligned}$$

Since $\varepsilon_n^2 = o(\rho_n^2 n^{-4-\alpha})$, we get $\mathbb{E}_{\boldsymbol{x}} \check{k}_{\gamma_n}(\boldsymbol{x}, \boldsymbol{x}')^2 \leq 2n^{-(2+\alpha)}$ for $n$ large enough.

Combining all terms in Eq. (G.2) yields its convergence to 0 with the rate $O(n^{-(2+\alpha)} \cdot 1 \cdot n^{1+\alpha}) = O(n^{-1})$ with probability at least $1 - 2n^{-\alpha}$, which concludes the proof. $\square$

The following theorem shows that the minimum-norm interpolants of the spiky-smooth kernel sequence can achieve optimal convergence rates for Sobolev target functions, as long as $\rho_n$ is properly chosen. We therefore introduce Assumption (D3), which resembles Assumption (D1) but allows more general target functions $f^* \in H^{s^*}(\Omega) \backslash \{0\}$, $s^* > 0$, that may lie outside of the RKHS.

(D3) Let $\Omega = \mathbb{S}^d$ or let $\Omega \subseteq \mathbb{R}^d$ be a bounded open Lipschitz domain. Let $P_X$ be a distribution on $\Omega$ with lower- and upper-bounded Lebesgue density. Consider i.i.d. data sets $D = \{(\boldsymbol{x}_1, y_1), \ldots, (\boldsymbol{x}_n, y_n)\} \subseteq \Omega \times \mathbb{R}$, where $\boldsymbol{x}_i \sim P_X$, $f^*(\boldsymbol{x}) = \mathbb{E}[y|\boldsymbol{x}] \in H^{s^*}(\Omega) \backslash \{0\}$, $s^* > 0$, with $\|f^*\|_{L^\infty(P_X)} < B_\infty$ for some constant $B_\infty > 0$, $\mathbb{E}y^2 < \infty$ and there are constants $\sigma, L > 0$ such that

$$\mathbb{E}\Big[|y - f^*(\boldsymbol{x})|^m \,\Big|\, \boldsymbol{x}\Big] \leq \frac{1}{2}m!\, \sigma^2\, L^{m-2},$$

for $P_X$-almost all $\boldsymbol{x} \in \Omega$ and all $m \geq 2$.

The above moment condition holds for additive Gaussian noise with variance $\sigma^2 > 0$. Hence Assumption (D1) is strictly stronger than Assumption (D3). The spike components $\check{k}_{\gamma_n}$ can also be chosen as in Theorem G.1.

**Theorem G.5.** *Assume Assumption (D3) holds and that the kernel components satisfy:*

- *the RKHS $\tilde{\mathcal{H}}$ of $\tilde{k}$ satisfies $\tilde{\mathcal{H}} = H^s$ as sets with $s > \max(s^*, d/2)$,*
- *$\check{k}_{\gamma_n}$ denotes the Laplace kernel with a sequence of positive bandwidths $(\gamma_n)$ fulfilling $\gamma_n \leq n^{-\frac{2+\alpha}{d}} \left((\frac{11}{4} + \frac{\alpha}{2})\ln n\right)^{-1}$, where $\alpha > 0$ is arbitrary.*

*Then there exists a constant $C > 0$ independent of $n$ and there is a sequence $(\rho_n)_{n \in \mathbb{N}}$ of order $n^{-s/(s^*+d/2)}$ such that the minimum-norm interpolant $f_{\rho_n, \gamma_n}$ of the $\rho_n$-regularized spiky-smooth kernel sequence $k_n := k_{\rho_n, \gamma_n}$ fulfills, with probability at least $1 - 6n^{-(1 \wedge \alpha)}$, for $n$ large enough,*

$$R_P(f_{\rho_n, \gamma_n}) - R_P(f^*) \leq C\, n^{-\frac{s^*}{(s^*+d/2)}} \log^2(n).$$

*Proof.* **Step 1: Kernel ridge regression $\hat{f}_{\rho_n}^{\tilde{k}}$ with optimal regularization achieves the desired convergence rate, with high probability.**

We slightly modify the proof of Theorem 8. Instead of using (Steinwart, 2001, Theorem 3.11 or Example 4.6), we use results of Fischer and Steinwart (2020). Here we first note that in the case

$\mathcal{H} = H^s(\Omega), \Omega \subseteq \mathbb{R}^d$ as RKHSs, we could directly use (Fischer and Steinwart, 2020, Corollary 5). Since we only have equivalent norms and also want to consider $\Omega = \mathbb{S}^d$, see Lemma F.7, we need to resort to the underlying more general result (Fischer and Steinwart, 2020, Theorem 1). To this end, we first need to verify its Assumptions (EMB), (EVD), (SRC), and (MOM).

**Step 1.1: Verifying (MOM).** The moment condition (MOM) on the noise distributions holds since we assumed it in Assumption (D3).

**Step 1.2: Simpler equivalent spaces.** We verify the remaining conditions by analyzing them for a nicer equivalent RKHS $\mathcal{H}$ and with uniform distribution $\nu$ on $\Omega$. For the non-spherical case $\Omega \subseteq \mathbb{R}^d$, we choose $\mathcal{H} := H^s(\Omega)$. For the case $\Omega = \mathbb{S}^d$, we choose $\mathcal{H}$ as an RKHS associated to a dot-product kernel $k$ with $\mu_l = \Theta((l+1)^{-2s})$, such that $\mathcal{H} \cong H^s \cong \tilde{\mathcal{H}}$ by Lemma B.1. In each case, $\mathcal{H} = \tilde{\mathcal{H}}$ with equivalent norms, and $L_2(\nu) = L_2(P_X)$ with equivalent norms since we assumed in (D3) that $P_X$ has an upper- and lower-bounded density.

**Step 1.3: Verifying (EVD+).** It suffices to verify the eigenvalue decay condition (EVD+) for $\mathcal{H}$ and $\nu$, since Lemma F.10 and Lemma F.9 then allow to transfer it to $\tilde{\mathcal{H}}$ and $P_X$.

For $\Omega \subseteq \mathbb{R}^d$, as pointed out in front of (Fischer and Steinwart, 2020, Corollary 5), it is well-known that $\mathcal{H}$ satisfies the polynomial eigenvalue decay assumption (EVD+) for $p := \frac{d}{2s}$.

For $\Omega = \mathbb{S}^d$, our definition of $\mathcal{H}$ together with Lemma B.1 directly yields (EVD+) for $\mathcal{H}$ and $\nu$ with $p := \frac{d}{2s}$.

**Step 1.4: Verifying (EMB) and (SRC).** The remaining two conditions of (Fischer and Steinwart, 2020, Theorem 1) are stated in terms of so-called power spaces, which in turn can be described by interpolation spaces of the real method. We therefore quickly recall these spaces. To this end, let us assume that we have two Banach spaces $E$ and $F$ such that $F \subset E$ and the corresponding inclusion map is continuous. Then the so-called $K$-functional of an $x \in E$ is defined by

$$K(x, t, E, F) := \inf_{y \in F} \left( t\|y\|_F + \|x - y\|_E \right), \qquad t > 0.$$

For $q \in (0, 1)$ and $x \in E$ we then define

$$\|x\|_{q,2}^2 := \int_0^\infty t^{-2q-1} K^2(x, t, E, F) \, dt$$

and $[E, F]_{q,2} := \{x \in E : \|x\|_{q,2} < \infty\}$. Let us now consider the cases $(E, F) = (L_2(\nu), \mathcal{H})$ and $(E, F) = (L_2(P_X), \tilde{\mathcal{H}})$. Now, for a suitable constant $C$, $\mathcal{H}$ and $\tilde{\mathcal{H}}$ are $C$-equivalent, and $L_2(\nu)$ and $L_2(P_x)$ are also $C$-equivalent. We then find

$$C^{-1}K(f, t, L_2(\Omega), \mathcal{H}) \leq K(f, t, L_2(\Omega), \tilde{\mathcal{H}}) \leq CK(f, t, L_2(\Omega), \mathcal{H})$$

for all $t > 0$ and $f \in L_2(\Omega)$, and consequently we have

$$[L_2(\nu), \mathcal{H}]_{q,2} = [L_2(P_X), \tilde{\mathcal{H}}]_{q,2}$$

for all $q \in (0, 1)$ with $C$-equivalent norms. Now, (Steinwart and Scovel, 2012, Theorem 4.6) shows that the power space $[\mathcal{H}]_\nu^q$ defined in (Steinwart and Scovel, 2012, Equation (36)) satisfies

$$[\mathcal{H}]_\nu^q = [L_2(\nu), \mathcal{H}]_{q,2}$$

with equivalent norms, and an analogous result is true for $\tilde{\mathcal{H}}$ and $P_X$. Moreover, $\mathcal{H}$ is dense in $L_2(\nu)$, and therefore (Steinwart and Scovel, 2012, Equations (36) and (18)) together with (Steinwart and Scovel, 2012, Lemma 2.2) show $[\mathcal{H}]_\nu^0 = L_2(\nu)$ as spaces. Again, we analogously find $[\tilde{\mathcal{H}}]_{P_X}^0 = L_2(P_X)$ Consequently, for all $0 \leq q < 1$ we have

$$[\mathcal{H}]_\nu^q = [\tilde{\mathcal{H}}]_{P_X}^q$$

with equivalent norms. From this we easily deduce that the Assumptions (EMB) and (SRC) are satisfied for $(\tilde{\mathcal{H}}, P_X)$ if and only if they are satisfied for $(\mathcal{H}, \nu)$.

**Step 1.4.1: Non-spherical case.** Now, let $\Omega \subseteq \mathbb{R}^d$. Then, (EMB) and (SRC) are satisfied for $(\mathcal{H}, \nu)$ for $\beta := s^*/s$ and an arbitrary but fixed $\alpha \in (p, \min\{1, p + \beta\})$ as outlined in front of (Fischer and Steinwart, 2020, Corollary 5). Applying Part ii) of (Fischer and Steinwart, 2020, Theorem 1) for

$\gamma = 0$ then shows that there exists a sequence $(\rho_n)_{n\in\mathbb{N}}$ of order $n^{-s/(s^*+d/2)}$ and a constant $K_1 > 0$ independent of $n$ such that, for $n$ large enough,

$$P^n\left(D \in (\mathbb{R}^d \times \mathbb{R})^n \mid \mathbb{E}_{\boldsymbol{x}}(\hat{f}^{\tilde{k}}_{\rho_n}(\boldsymbol{x}) - f^*(\boldsymbol{x}))^2 \geq K_1 n^{-\frac{s^*}{(s^*+d/2)}} \log^2(n)\right) \leq 4n^{-1}. \quad (G.3)$$

**Step 1.4.2: Spherical case.** Suppose $\Omega = \mathbb{S}^d$. Using the differently normalized spherical harmonics $Y_{l,i}$ and $\tilde{Y}_{l,i}$ from Appendix B.3, we obtain for the power spaces:

$$
\begin{aligned}
[\mathcal{H}]_\nu^q &= \left\{ \sum_{l=0}^\infty \sum_{i=1}^{N_{l,d}} a_{li} \tilde{\mu}_i^{q/2} [\tilde{Y}_{l,i}]_\sim \mid (a_{li}) \in \ell_2 \right\} \\
&= \left\{ \sum_{l=0}^\infty \sum_{i=1}^{N_{l,d}} a_{l,i}(l+1)^{-qs} [Y_{l,i}]_\sim \mid (a_{li}) \in \ell_2 \right\} \\
&= \left\{ \sum_{l=0}^\infty \sum_{i=1}^{N_{l,d}} b_{l,i} [Y_{l,i}]_\sim \mid \sum_{l=0}^\infty \sum_{i=1}^{N_{l,d}} b_{l,i}^2 (l+1)^{2qs} < \infty \right\} \\
&= H^{qs}(\mathbb{S}^d),
\end{aligned}
$$

where the first equation follows from (Steinwart and Scovel, 2012, Eq. (36)), the second one from our definition of the $\mu_l$ in Step 1.2, and the last one from (Hubbert et al., 2023, Section 3). Again, we can choose an arbitrary but fixed $\alpha \in (p, \min\{1, p+\beta\})$. Then, $\alpha s > ps = d/2$, which means that $[\mathcal{H}]_\nu^\alpha = H^{\alpha s}$ is an RKHS with bounded kernel (De Vito et al., 2021, Theorem 8), hence the embedding condition (EMB) holds. Similarly, (SRC) holds for $\beta := s^*/s$ and the result follows as above.

**Step 2: $\hat{f}_0^{k_n}$ and $\hat{f}_{\rho_n}^{\tilde{k}}$ are close in $L_2(P_X)$, with high probability.**

Since $\frac{s^*}{(s^*+d/2)} < 1$, it suffices to show that $k_n$ fulfills, for some constant $K_2 > 0$,

$$P^n\left(D \in (\mathbb{R}^d \times \mathbb{R})^n \mid \mathbb{E}_{\boldsymbol{x}}(\hat{f}_0^{k_n}(\boldsymbol{x}) - \hat{f}_{\rho_n}^{\tilde{k}}(\boldsymbol{x}))^2 \geq K_2 n^{-1}\right) \leq 2n^{-\alpha}. \quad (G.4)$$

Since $\gamma_n \leq n^{-\frac{2+\alpha}{d}}\left(\left(\frac{11}{4} + \frac{\alpha}{2}\right)\ln n\right)^{-1}$, it holds that $|\check{k}_{\gamma_n}(\boldsymbol{x}, \boldsymbol{y})| \leq \varepsilon_n := \rho_n n^{-\frac{5+\alpha}{2}}$ (cf. Remark G.2). Then the product of all terms in Eq. (G.1) satisfies $O(n^{4+\alpha}\rho_n^{-2}\varepsilon_n^2) = o(n^{-1})$ with probability at least $1 - 2n^{-\alpha}$. On the same event, the bound on Eq. (G.2) remains of order $O(n^{-1})$, which shows Eq. (G.4). Combining (G.3) and (G.4) with the triangle inequality in $L_2(P_X)$ concludes the proof. $\quad\square$

**Remark G.6** (Optimality of the rates)**.** In the setting of Theorem G.5, we can apply (Fischer and Steinwart, 2020, Theorem 2) in order to obtain lower bounds on the achievable rates. We have already verified the conditions (MOM), (EVD+), (EMB), and (SRC) in the proof of Theorem G.5. In the case $s^* > d/2$, we have $\beta = s^*/s > d/(2s) = p$ and can therefore choose $\alpha \in (p, \beta)$ such that $\beta > \alpha$. Then, (Fischer and Steinwart, 2020, Theorem 2) yields a lower bound on the rate of the form (with constant probability)

$$n^{-\frac{\beta}{\beta+p}} = n^{-\frac{s^*}{s^*+d/2}},$$

which matches the rates in Theorem G.5 up to log terms. $\quad\blacktriangleleft$

### G.1 Auxiliary results for the proof of Theorem 8

The distributional Assumption (D2) immediately implies that the training points are separated with high probability.

**Lemma G.7.** *Assume (D2) is fulfilled with $\beta_X > 0$. Then with probability at least $1 - O(n^{-\alpha})$,*

$$\min_{i,j\in[n]:i\neq j} |\boldsymbol{x}_i - \boldsymbol{x}_j| \geq n^{-\frac{2+\alpha}{\beta_X}}.$$

*Proof.* For any $i \in [n]$, the union bound implies

$$P\left(\min_{j\in[n]:i\neq j} |\boldsymbol{x}_i - \boldsymbol{x}_j| \leq \delta\right) = P\left(\bigcup_{j\in[n]:j\neq i} \{\boldsymbol{x}_j \in B_\delta(\boldsymbol{x}_i)\}\right) \leq (n-1)\phi(\delta).$$

Another union bound yields

$$P(\min_{i,j\in[n]:i\neq j}|\boldsymbol{x}_i - \boldsymbol{x}_j| \leq \delta) \leq n(n-1)\phi(\delta).$$

Choosing $\delta_\alpha(n) = n^{-\frac{2+\alpha}{\beta_X}}$ yields $\phi(\delta_\alpha(n)) = O(\frac{1}{n^{2+\alpha}})$, which concludes the proof. $\quad\square$

The following lemma bounds $\|\boldsymbol{A}^{-1} - \boldsymbol{B}^{-1}\|$ via $\|\boldsymbol{A}^{-1}\|$ and $\|\boldsymbol{A} - \boldsymbol{B}\|$. Similar results can for example be found in (Horn and Johnson, 2013, Section 5.8).

**Lemma G.8.** *Let* $\boldsymbol{A}, \boldsymbol{B} \in \mathbb{R}^{n\times n}$ *be invertible matrices and let* $\|\cdot\|$ *be a submultiplicative matrix norm with* $\|\boldsymbol{I}_n\| = 1$. *If* $\boldsymbol{A}$ *and* $\boldsymbol{B}$ *fulfill* $\|\boldsymbol{A}^{-1}\|\|\boldsymbol{A} - \boldsymbol{B}\| < 1$, *then it holds that*

$$\|\boldsymbol{B}^{-1} - \boldsymbol{A}^{-1}\| \leq \frac{\|\boldsymbol{A}^{-1}\|^2 \cdot \|\boldsymbol{A} - \boldsymbol{B}\|}{1 - \|\boldsymbol{A}^{-1}\| \cdot \|\boldsymbol{A} - \boldsymbol{B}\|}.$$

*Proof.* Because of $\|\boldsymbol{A}^{-1}(\boldsymbol{A} - \boldsymbol{B})\| \leq \|\boldsymbol{A}^{-1}\|\|\boldsymbol{A} - \boldsymbol{B}\| < 1$ we get

$$\|\boldsymbol{I} - \boldsymbol{A}^{-1}(\boldsymbol{A} - \boldsymbol{B})\| \geq 1 - \|\boldsymbol{A}^{-1}\|\|\boldsymbol{A} - \boldsymbol{B}\|.$$

Writing $\boldsymbol{B} = \boldsymbol{A}(\boldsymbol{I} - \boldsymbol{A}^{-1}(\boldsymbol{A} - \boldsymbol{B}))$ yields $\boldsymbol{B}^{-1} = (\boldsymbol{I} - \boldsymbol{A}^{-1}(\boldsymbol{A} - \boldsymbol{B}))^{-1}\boldsymbol{A}^{-1}$ which implies

$$\|\boldsymbol{B}^{-1}\| \leq \frac{\|\boldsymbol{A}^{-1}\|}{1 - \|\boldsymbol{A}^{-1}\|\|\boldsymbol{A} - \boldsymbol{B}\|}.$$

Now write $\boldsymbol{B}^{-1} - \boldsymbol{A}^{-1} = \boldsymbol{A}^{-1}(\boldsymbol{A} - \boldsymbol{B})\boldsymbol{B}^{-1}$ to get

$$\|\boldsymbol{B}^{-1} - \boldsymbol{A}^{-1}\| \leq \|\boldsymbol{A}^{-1}\|\|\boldsymbol{A} - \boldsymbol{B}\|\|\boldsymbol{B}^{-1}\|.$$

Combining the last two inequalities concludes the proof. $\quad\square$

## G.2 RKHS norm bounds

Here we show that if $\tilde{k}$ and $\check{k}_\gamma$ have RKHS equivalent to some Sobolev space $H^s$, $s > d/2$, then the RKHS of the spiky-smooth kernel $k_{\rho,\gamma}$ is also equivalent to $H^s$, for any fixed $\rho, \gamma > 0$. Hence all members of the spiky-smooth kernel sequence may have RKHS equivalent to a Sobolev space $H^s$ and are individually inconsistent due to Theorem 1; yet the sequence is consistent. This shows that when arguing about generalization properties based on RKHS equivalence, the constants matter and the narrative that depth does not matter in the NTK regime as in Bietti and Bach (2021) is too simplified.

The following proposition states that the sum of kernels with equivalent RKHS yields an RKHS that is equivalent to the RKHS of the summands. For example, the spiky-smooth kernel with Laplace components possesses an RKHS equivalent to the RKHS of the Laplace kernel.

**Proposition G.9.** *Let* $\mathcal{H}_1$ *and* $\mathcal{H}_2$ *denote the RKHS of* $k_1$ *and* $k_2$ *respectively. If* $\mathcal{H}_1 = \mathcal{H}_2$ *then the RKHS* $\mathcal{H}$ *of* $k = k_1 + k_2$ *fulfills* $\mathcal{H} = \mathcal{H}_1$. *Moreover, if* $C \geq 1$ *is a constant with* $\frac{1}{C}\|f\|_{\mathcal{H}_2} \leq \|f\|_{\mathcal{H}_1} \leq C\|f\|_{\mathcal{H}_2}$, *then we have* $\frac{1}{\sqrt{2}C}\|f\|_{\mathcal{H}_1} \leq \|f\|_{\mathcal{H}} \leq \|f\|_{\mathcal{H}_1}$.

*Proof.* The RKHS of $k = k_1 + k_2$ is given by $\mathcal{H} = \mathcal{H}_1 + \mathcal{H}_2$ with norm

$$\|f\|_{\mathcal{H}}^2 = \min\{\|f_1\|_{\mathcal{H}_1}^2 + \|f_2\|_{\mathcal{H}_2}^2 : f = f_1 + f_2, f_1 \in \mathcal{H}_1, f_2 \in \mathcal{H}_2\}.$$

To see this we consider the map $\Phi : X \to \mathcal{H}_1 \times \mathcal{H}_2$ defined by $\Phi(\boldsymbol{x}) := (\Phi_1(\boldsymbol{x},\cdot), \Phi_2(\boldsymbol{x},\cdot))$ for all $\boldsymbol{x} \in X$, where $X$ is the set, the spaces $\mathcal{H}_i$ live on and $\Phi_i(\boldsymbol{x}) := k_i(\boldsymbol{x},\cdot)$. The reproducing property of $k_1$ and $k_2$ immediately ensures that $\Phi$ is a feature map of $k_1 + k_2$ and Theorem E.3 then shows

$$\mathcal{H} = \left\{\langle w, \Phi(\cdot)\rangle_{\mathcal{H}_1 \times \mathcal{H}_2} : w \in \mathcal{H}_1 \times \mathcal{H}_2\right\}$$
$$= \left\{\langle w_1, \Phi_1(\cdot)\rangle_{\mathcal{H}_1} + \langle w_2, \Phi_2(\cdot)\rangle_{\mathcal{H}_2} : w_1 \in \mathcal{H}_1, w_2 \in \mathcal{H}_2\right\} = \mathcal{H}_1 + \mathcal{H}_2$$

as well as the formula for the norm on $\mathcal{H}$. Now let $f \in \mathcal{H}$. Considering the decomposition $f = f_1 + 0$ then gives $\|f\|_{\mathcal{H}} \leq \|f\|_{\mathcal{H}_1}$. Moreover, for $f = f_1 + f_2$ with $f_i \in \mathcal{H}_i$ we have

$$\|f\|_{\mathcal{H}_1} \leq \|f_1\|_{\mathcal{H}_1} + \|f_2\|_{\mathcal{H}_1} \leq \|f_1\|_{\mathcal{H}_1} + C\|f_2\|_{\mathcal{H}_2} \leq \sqrt{2}C\left(\|f_1\|_{\mathcal{H}_1}^2 + \|f_2\|_{\mathcal{H}_1}^2\right)^{1/2}.$$

Taking the infimum over all decomposition then yields the estimate $\|f\|_{\mathcal{H}_1} \leq \sqrt{2}C\|f\|_{\mathcal{H}}$. $\quad\square$

# H  Spiky-smooth activation functions induced by Gaussian components

Here we explore the properties of the NNGP and NTK activation functions induced by spiky-smooth kernels with Gaussian components.

To offer some more background, it is well-known that NNGPs and NTKs on the sphere $\mathbb{S}^d$ are dot-product kernels, i.e., kernels of the form $k_d(\boldsymbol{x}, \boldsymbol{x}') = \kappa(\langle \boldsymbol{x}, \boldsymbol{x}' \rangle)$, where the function $\kappa$ has a series representation $\kappa(t) = \sum_{i=0}^{\infty} b_i t^i$ with $b_i \geq 0$ and $\sum_{i=0}^{\infty} b_i < \infty$. The function $\kappa$ is independent of the dimension $d$ of the sphere. Conversely, all such kernels can be realized as NNGPs or NTKs (Simon et al., 2022, Theorem 3.1).

As dot-product kernel $k(\boldsymbol{x}, \boldsymbol{y}) = \kappa(\langle \boldsymbol{x}, \boldsymbol{y} \rangle)$ on the sphere, the Gaussian kernel has the simple analytic expression,

$$\kappa_\gamma^{Gauss}(z) = \exp\left(\frac{2(z-1)}{\gamma}\right),$$

with Taylor expansion

$$\kappa_\gamma^{Gauss}(z) = \sum_{i=0}^{\infty} \underbrace{\frac{2^i}{\gamma^i i!} \exp(-2/\gamma)}_{b_i^{Gauss}} \; z^i.$$

For spiky-smooth kernels $k = \tilde{k} + \rho \check{k}_\gamma$ with Gaussian components $\tilde{k}$ and $\check{k}_\gamma$ of width $\tilde{\gamma}$ and $\gamma$ respectively, we get Taylor series coefficients

$$b_i = \frac{\exp(-2/\tilde{\gamma})}{i!}\left(\frac{2}{\tilde{\gamma}}\right)^i + \rho\frac{\exp(-2/\gamma)}{i!}\left(\frac{2}{\gamma}\right)^i. \tag{H.1}$$

Now Theorem 11 states that as soon as $\kappa$ induces a dot-product kernel for every input dimension $d$, then the dot-product kernels can be written as the NNGP kernel of a 2-layer fully-connected network without biases and with the induced activation function

$$\phi_{NNGP}^\kappa(x) = \sum_{i=0}^{\infty} s_i b_i^{1/2} h_i(x),$$

or as the NTK of a 2-layer fully-connected network without biases and with the induced activation function

$$\phi_{NTK}^\kappa(x) = \sum_{i=0}^{\infty} s_i \left(\frac{b_i}{i+1}\right)^{1/2} h_i(x),$$

where $h_i$ denotes the $i$-th Probabilist's Hermite polynomial normalized such that $\|h_i\|_{L_2(\mathcal{N}(0,1))} = 1$ and $s_i \in \{-1, +1\}$ are arbitrarily chosen for all $i \in \mathbb{N}_0$.

Now we can study the induced activation functions if we know the kernel's Taylor coefficients $(b_i)_{i \in \mathbb{N}_0}$. If infinitely many $b_i > 0$, then infinitely many activation functions induce the same dot-product kernel, with different choices of the signs $s_i$. For alternating signs $s_i = (-1)^i$, the symmetry property $h_i(-x) = (-1)^i h_i(x)$ of the Hermite polynomials implies

$$\phi_{NNGP,+-}(x) = \phi_{NNGP,+}(-x), \qquad \phi_{NTK,+-}(x) = \phi_{NTK,+}(-x).$$

To form an orthonormal basis of $L_2(\mathcal{N}(0,1))$ the unnormalized Probabilist's Hermite polynomials $He_i$ have to be normalized by $h_i(x) = \frac{1}{\sqrt{i!}} He_i(x)$. We can use the identity $\exp(xt - \frac{t^2}{2}) = \sum_{i=0}^{\infty} He_i(x)\frac{t^i}{i!}$ with $t = \sqrt{2/\gamma}$ to analytically express the NNGP activation of the Gaussian kernel with all $s_i = +1$ as the exponential function

$$\phi_{NNGP,+}^{Gauss}(x) = \exp(-1/\gamma)\sum_{i=0}^{\infty}\frac{1}{i!}\left(\frac{2}{\gamma}\right)^{\frac{i}{2}} h_i(x) = \exp\left(\left(\frac{2}{\gamma}\right)^{\frac{1}{2}} \cdot x - \frac{2}{\gamma}\right). \tag{H.2}$$

Remarkably, the Gaussian kernel can not only be induced by an exponential activation function, but also by a single shifted sine activation function. This is shown in the following proposition.

**Proposition H.1** (**Trigonometric Gaussian NNGP activation functions**). *For any $\gamma > 0$ and the bi-alternating choice of signs $\{(-1)^{\lfloor i/2 \rfloor}\}_{i=0,1,2,...}$, the Gaussian kernel of bandwidth $\gamma$ can be realized as the NNGP kernel of a two-layer fully-connected network without biases and with activation function*

$$\phi_{NNGP,++--}^{Gauss}(x) = \sin((2/\gamma)^{1/2}x) + \cos((2/\gamma)^{1/2}x).$$

*Proof.* We write $c = 2/\gamma$. We need to show that

$$\sin(c^{1/2}x) + \cos(c^{1/2}x) = e^{-c/2} \sum_{i=0}^{\infty} (-1)^{\lfloor i/2 \rfloor} \frac{c^{i/2}}{i!} He_i(x).$$

We will use the fact that

$$e^{2xz-z^2} = \sum_{i=0}^{\infty} \frac{z^i}{i!} H_i(x),$$

with the choices $z_1 = i\sqrt{c/2}$ and $z_2 = -i\sqrt{c/2}$. Now, using $e^{iax+b} = e^b(\cos(ax) + i\sin(ax))$, observe that

$$\sin(\sqrt{c}x) = \sin(\sqrt{2c}x/\sqrt{2}) = \frac{1}{2ie^{c/2}} \left( e^{ix\sqrt{c}+c/2} - e^{ix\sqrt{c}+c/2} \right)$$

$$= \frac{1}{2ie^{c/2}} \left( \sum_{i=0}^{\infty} \frac{(i\sqrt{c/2})^i}{i!} H_i(x/\sqrt{2})(1-(-1)^i) \right)$$

$$= e^{-c/2} \sum_{i=0}^{\infty} (-1)^i \frac{(\sqrt{c/2})^{2i+1}}{(2i+1)!} H_{2i+1}(x/\sqrt{2}).$$

An analogous calculation yields

$$\cos(c^{1/2}x) = e^{-c/2} \sum_{i=0}^{\infty} (-1)^i \frac{(\sqrt{c/2})^{2i}}{(2i)!} H_{2i}(x/\sqrt{2}).$$

Finally, using $H_i(x/\sqrt{2}) = 2^{i/2} He_i(x)$, we get

$$\sin(c^{1/2}x) + \cos(c^{1/2}x) = e^{-c/2} \sum_{i=0}^{\infty} (-1)^{\lfloor i/2 \rfloor} \frac{(c/2)^{i/2}}{i!} H_i(x/\sqrt{2})$$

$$= e^{-c/2} \sum_{i=0}^{\infty} (-1)^{\lfloor i/2 \rfloor} \frac{c^{i/2}}{i!} He_i(x).$$

$\square$

For $\phi_{NNGP}(x) = \sum_{i=0}^{\infty} s_i \sqrt{b_i} h_i(x)$, we get $\|\phi\|_{L_2(\mathcal{N}(0,1))}^2 = \sum_{i=0}^{\infty} b_i$ invariant to the choice $\{s_i\}_{i\in\mathbb{N}}$. For Gaussian NNGP activation components with bandwidth $\gamma > 0$ this yields

$$\|\phi_{NNGP}^{Gauss}\|_{L_2(\mathcal{N}(0,1))}^2 = \exp(-2/\gamma) \sum_{i=0}^{\infty} \frac{1}{i!} \left( \frac{2}{\gamma} \right)^i = 1, \tag{H.3}$$

because $\{h_i\}_{i\in\mathbb{N}_0}$ is an ONB of $L_2(\mathcal{N}(0,1))$. Analogously, for Gaussian NTK activation components, we get

$$\|\phi_{NTK}^{Gauss}\|_{L_2(\mathcal{N}(0,1))}^2 = \exp(-2/\gamma) \sum_{i=0}^{\infty} \frac{1}{(i+1)!} \left( \frac{2}{\gamma} \right)^i$$

$$= \exp(-2/\gamma) \frac{\gamma}{2} \sum_{i=1}^{\infty} \frac{1}{i!} \left( \frac{2}{\gamma} \right)^i = \frac{\gamma}{2} \left( 1 - \exp\left( -\frac{2}{\gamma} \right) \right). \tag{H.4}$$

This implies that the average amplitude of NNGP activation functions does not depend on $\gamma$, while the average amplitude of NTK activation functions decays with $\gamma \to 0$.

By the fact $h'_n(x) = \sqrt{n}h_{n-1}(x)$, we know that any activation function $\phi(x) = \sum_{i=0}^{\infty} s_i a_i h_i(x)$ has the derivative $\phi'(x) = \sum_{i=0}^{\infty} s_i a_{i+1}\sqrt{i+1} \cdot h_i(x)$ as long as $\sum_{i=0}^{\infty} |a_{i+1}\sqrt{i+1}| < \infty$.

The following proposition formalizes the additive approximation $\phi^k \approx \phi^{\tilde{k}} + \rho^{1/2}\phi^{\check{k}_\gamma}$, and quantifies the necessary scaling of $\gamma$ for any demanded precision of the approximation.

**Proposition H.2.** *Fix $\tilde{\gamma}, \rho > 0$ arbitrary. Let $k = \tilde{k} + \rho\check{k}_\gamma$ denote the spiky-smooth kernel where $\tilde{k}$ and $\check{k}_\gamma$ are Gaussian kernels of bandwidth $\tilde{\gamma}$ and $\gamma$, respectively. Assume that we choose the activation functions $\phi^k_{NTK}$, $\phi^{\tilde{k}}_{NTK}$ and $\phi^{\check{k}_\gamma}_{NTK}$ as in [Theorem 11](#) with same signs $\{s_i\}_{i\in\mathbb{N}}$. Then, for $\gamma > 0$ small enough, it holds that*

$$\|\phi^k_{NTK} - (\phi^{\tilde{k}}_{NTK} + \sqrt{\rho}\cdot\phi^{\check{k}_\gamma}_{NTK})\|^2_{L_2(\mathcal{N}(0,1))} \leq 2^{1/2}\rho\gamma^{3/2}\exp\left(-\frac{1}{\gamma}\right) + \frac{4\pi\gamma(1+\tilde{\gamma})}{\tilde{\gamma}},$$

$$\|\phi^k_{NNGP} - (\phi^{\tilde{k}}_{NNGP} + \sqrt{\rho}\cdot\phi^{\check{k}_\gamma}_{NNGP})\|^2_{L_2(\mathcal{N}(0,1))} \leq 2^{3/2}\rho\gamma^{1/2}\exp\left(-\frac{1}{\gamma}\right) + \frac{8\pi\gamma(1+\tilde{\gamma})}{\tilde{\gamma}^2}.$$

*Proof.* Let $b_{i,\gamma} = \frac{2^i}{\gamma^i i!}\exp(-2/\gamma)$ denote the Taylor coefficients of the Gaussian kernel. All considered infinite series converge absolutely.

$$\|\phi^{\tilde{\gamma};\gamma;\rho}_{NNGP} - (\phi^{\tilde{\gamma}}_{NNGP} + \sqrt{\rho}\cdot\phi^{\gamma}_{NNGP})\|^2_{L_2(\mathcal{N}(0,1))}$$

$$=\|\sum_{i=0}^{\infty} s_i\sqrt{b_{i,\tilde{\gamma}} + \rho b_{i,\gamma}}\,h_i(x) - \sum_{i=0}^{\infty} s_i(\sqrt{b_{i,\tilde{\gamma}}} + \sqrt{\rho b_{i,\gamma}})h_i(x)\|^2_{L_2(\mathcal{N}(0,1))}$$

$$= \sum_{i=0}^{\infty}\left(\sqrt{b_{i,\tilde{\gamma}} + \rho b_{i,\gamma}} - (\sqrt{b_{i,\tilde{\gamma}}} + \sqrt{\rho b_{i,\gamma}})\right)^2$$

$$\leq 2\underbrace{\sum_{i=0}^{I}(\sqrt{b_{i,\tilde{\gamma}} + \rho b_{i,\gamma}} - b_{i,\tilde{\gamma}}^{1/2})^2}_{(I)} + 2\rho\underbrace{\sum_{i=0}^{I} b_{i,\gamma}}_{(II)} + 2\underbrace{\sum_{i=I+1}^{\infty}(\sqrt{b_{i,\tilde{\gamma}} + \rho b_{i,\gamma}} - \rho^{1/2}b_{i,\gamma}^{1/2})^2}_{(III)} + 2\underbrace{\sum_{i=I+1}^{\infty} b_{i,\tilde{\gamma}}}_{(IV)},$$

for any $I \in \mathbb{N}$. To bound $(I)$ observe

$$\sum_{i=0}^{I}(\sqrt{b_{i,\tilde{\gamma}} + \rho b_{i,\gamma}} - b_{i,\tilde{\gamma}}^{1/2})^2 = \sum_{i=0}^{I}\left(\rho b_{i,\gamma} + 2b_{i,\tilde{\gamma}}\left(1 - \sqrt{1 + \frac{\rho b_{i,\gamma}}{b_{i,\tilde{\gamma}}}}\right)\right) \leq \rho\sum_{i=0}^{I} b_{i,\gamma}.$$

An analogous calculation for $(III)$ yields

$$\sum_{i=I+1}^{\infty}(\sqrt{b_{i,\tilde{\gamma}} + \rho b_{i,\gamma}} - \rho^{1/2}b_{i,\gamma}^{1/2})^2 \leq \sum_{i=I+1}^{\infty} b_{i,\tilde{\gamma}}.$$

So overall we get the bound

$$\|\phi^{\tilde{\gamma};\gamma;\rho}_{NNGP} - (\phi^{\tilde{\gamma}}_{NNGP} + \sqrt{\rho}\cdot\phi^{\gamma}_{NNGP})\|^2_{L_2(\mathcal{N}(0,1))} \leq 4\rho\sum_{i=0}^{I} b_{i,\gamma} + 4\sum_{i=I+1}^{\infty} b_{i,\tilde{\gamma}}. \quad\text{(H.5)}$$

Now, defining $c := 2/\gamma$,

$$\sum_{i=0}^{I} b_{i,\gamma} = \exp(-c)\sum_{i=0}^{I}\frac{1}{i!}c^i = \frac{\Gamma(I+1,c)}{I!},$$

where $\Gamma(k+1,c)$ denotes the upper incomplete Gamma function. Choosing $I = \lfloor\frac{c}{2\pi}\rfloor$, (Pinelis, 2020, Theorem 1.1) yields, for $c \geq 121$,

$$\frac{\Gamma(I+1,c)}{I!} \leq \exp(-c)\frac{(c + (I+1)!^{1/I})^{I+1}}{(I+1)!\cdot(I+1)!^{1/I}} \leq \frac{\exp(-c)(c+I)^{I+1}}{(I+1)!^{(I+1)/I}}$$

$$\leq \frac{\exp(-c)(c+I)^{I+1}}{(2\pi(I+1))^{1/2}\left(\frac{I+1}{e}\right)^{(I+1)^2/I}}$$

$$\leq \frac{1}{\sqrt{2\pi(I+1)}} \exp\Big(-c + (I+1)\big(\ln(c+I) - \ln(I+1) + 1\big)\Big)$$

$$\leq \frac{1}{\sqrt{c}} \exp\Big(-c + \big(\frac{c}{2\pi}+1\big)\big(\ln(\frac{2\pi+1}{2\pi}c) - \ln(\frac{c}{2\pi}) + 1\big)\Big)$$

$$\leq \frac{1}{\sqrt{c}} \exp\Big(-c + \big(\frac{c}{2\pi}+1\big)\big(\ln(2\pi+1) + 1\big)\Big) \leq \frac{\exp\big(-\frac{c}{2}\big)}{\sqrt{c}}, \qquad \text{(H.6)}$$

where we used $(I+1)!^{1/I} \leq I$ for $I \geq 3$ in the first line, Stirling's approximation in the second line, and $\big(\frac{c}{2\pi}+1\big)\big(\ln(2\pi+1)+1\big) \leq c/2$ for $c \geq 121$ in the last line.

It is obvious that

$$\sum_{i=I+1}^{\infty} b_{i,\tilde\gamma} \to 0, \quad \text{for } I = \lfloor \frac{c}{2\pi} \rfloor \to \infty.$$

To quantify the rate of convergence, we use the bound $\Gamma(I+1, c_0) \geq e^{-c_0} I!(1 + c_0/(I+1))^I$, which follows from applying Jensen's inequality to $\Gamma(I+1, c_0) = e^{-c_0} I! \mathbb{E}(1 + c_0/G)^I$, where $G \sim \Gamma(I+1, 1)$ and $\mathbb{E}G = I+1$. Defining $c_0 = 2/\tilde\gamma$, it holds that

$$\sum_{i=I+1}^{\infty} b_{i,\tilde\gamma} \leq 1 - \frac{\Gamma(I+1, c_0)}{I!} \leq 1 - e^{-c_0}\Big(1 + \frac{c_0}{I+1}\Big)^I \leq 1 - e^{-c_0}\Big(1 + \frac{c_0}{I+1}\Big)^{I+1}.$$

Taking the first two terms of the Laurent series expansion of $n \mapsto \big(1 + \frac{c_0}{n}\big)^n$ about $n = \infty$ yields $\big(1 + \frac{c_0}{I+1}\big)^{I+1} > e^{c_0}\big(1 - \frac{c_0^2}{2(I+1)}\big)$ for $I$ large enough (where we demand $\gamma \in o(\tilde\gamma^2)$), thus

$$\sum_{i=I+1}^{\infty} b_{i,\tilde\gamma} \leq 1 - e^{-c_0}\Big(1 + \frac{c_0}{I+1}\Big)^{I+1} \cdot \Big(1 + \frac{c_0}{I+1}\Big)^{-1}$$

$$\leq \frac{c_0/(I+1) + c_0^2/(2(I+1))}{1 + c_0/(I+1)} \leq \frac{c_0}{I+1} + \frac{c_0^2}{2(I+1)} \leq \frac{4\pi}{\tilde\gamma c} + \frac{4\pi}{\tilde\gamma^2 c}. \qquad \text{(H.7)}$$

Plugging (H.6) and (H.7) into (H.5) yields, for $\gamma \leq 1/61$,

$$\|\phi_{NNGP}^{\tilde\gamma,\gamma,\rho} - (\phi_{NNGP}^{\tilde\gamma} + \sqrt{\rho} \cdot \phi_{NNGP}^{\gamma})\|_{L_2(\mathcal{N}(0,1))}^2 \leq 2^{3/2}\rho\gamma^{1/2}\exp\Big(-\frac{1}{\gamma}\Big) + \frac{8\pi\gamma(1+\tilde\gamma)}{\tilde\gamma^2}.$$

For the NTK we get

$$\|\phi_{NTK}^{\tilde\gamma,\gamma,\rho} - (\phi_{NTK}^{\tilde\gamma} + \sqrt{\rho} \cdot \phi_{NTK}^{\gamma})\|_{L_2(\mathcal{N}(0,1))}^2$$

$$= \|\sum_{i=0}^{\infty} s_i \sqrt{\frac{b_{i,\tilde\gamma} + \rho b_{i,\gamma}}{i+1}} h_i(x) - \sum_{i=0}^{\infty} s_i \Big(\sqrt{\frac{b_{i,\tilde\gamma}}{i+1}} + \sqrt{\frac{\rho b_{i,\gamma}}{i+1}}\Big) h_i(x)\|_{L_2(\mathcal{N}(0,1))}^2$$

$$= \sum_{i=0}^{\infty} \frac{1}{i+1}\Big(\sqrt{b_{i,\tilde\gamma} + \rho b_{i,\gamma}} - (\sqrt{b_{i,\tilde\gamma}} + \sqrt{\rho b_{i,\gamma}})\Big)^2.$$

We can proceed exactly as for the NNGP, but choose $I = \lfloor \frac{c}{2\pi} \rfloor - 1$ to get

$$\sum_{i=0}^{I} \frac{b_{i,\gamma}}{i+1} = \exp(-c)\sum_{i=0}^{I} \frac{c^i}{(i+1)!} = \frac{\exp(-c)}{c}\Big(\sum_{i=0}^{I+1} \frac{c^i}{i!} - 1\Big) \leq \frac{\exp(-c/2)}{c^{3/2}} - \frac{\exp(-c)}{c},$$

and replace (H.7) with

$$\sum_{i=I+1}^{\infty} \frac{b_{i,\tilde\gamma}}{i+1} = \frac{\exp(-c_0)}{c_0}\sum_{i=I+2}^{\infty} \frac{c_0^i}{i!} \leq \frac{1}{I+2} + \frac{c_0}{2(I+2)} \leq \frac{\pi\gamma(1+\tilde\gamma)}{\tilde\gamma}. \qquad \Box$$

# I Additional experimental results

The code to reproduce all our experiments is provided in the supplementary material and under

<p style="text-align:center">https://github.com/moritzhaas/mind-the-spikes</p>

Our implementations rely on PyTorch (Paszke et al., 2019) for neural networks and mpmath (Johansson et al., 2023) for high-precision calculations.

## I.1 Experimental details of Figure 1

For the kernel experiment (Figure 1a), we used the Laplace kernel with bandwidth $0.4$ and the spiky-smooth kernel (4) with Laplace components with $\rho = 1, \tilde{\gamma} = 1, \gamma = 0.01$.

For the neural network experiment (Figure 1b,c) we initialize 2-layer networks with NTK parametrization (Jacot et al., 2018) and He initialization (He et al., 2015). Using the antisymmetric initialization trick from Zhang et al. (2020) doubles the network width from 10000 to 20000 and helps to prevent errors induced by the random initialization function. It might also be helpful to increase the initialization variance (Chizat et al., 2019). We train the network with stochastic gradient descent of batch size 1 over the 15 training samples with learning rate $0.04$ for 2500 epochs. Training with gradient descent and learning rate $0.4$ produces similar results. We use the spiky-smooth activation function given by $x \mapsto ReLU(x) + 0.01 \cdot (\sin(100x) + \cos(100x))$, which corresponds to $x \mapsto ReLU(x) + \omega_{NTK}(x, \frac{1}{5000})$, including both even and uneven Hermite coefficients.

## I.2 Disentangling signal from noise in neural networks with spiky-smooth activation functions

Since our spiky-smooth activation function has the additive form $\sigma_{spsm}(x) = ReLU(x) + \omega_{NTK}(x; \frac{1}{5000})$, we can dissect the learned neural network

$$f_{spsm}(\boldsymbol{x}) = \boldsymbol{W}_2 \cdot \sigma_{spsm}(\boldsymbol{W}_1 \cdot \boldsymbol{x} + \boldsymbol{b}_1) + b_2 = f_{ReLU}(\boldsymbol{x}) + f_{spikes}(\boldsymbol{x}) \tag{I.1}$$

into its $ReLU$-component

$$f_{ReLU}(\boldsymbol{x}) = \boldsymbol{W}_2 \cdot ReLU(\boldsymbol{W}_1 \cdot \boldsymbol{x} + \boldsymbol{b}_1) + b_2,$$

and its spike component

$$f_{spikes}(\boldsymbol{x}) = \boldsymbol{W}_2 \cdot \omega_{NTK}(\boldsymbol{W}_1 \cdot \boldsymbol{x} + \boldsymbol{b}_1; \frac{1}{5000}).$$

If the analogy to the spiky-smooth kernel holds and $f_{spikes}$ fits the noise in the labels while having a small $L_2$-norm, then $f_{ReLU}$ would have learned the signal in the data. Indeed Figure I.1 demonstrates that this simple decomposition is useful to disentangle the learned signal from the spike component in our setting. The figure also suggests that the oscillations in the activations of the hidden layer constructively interfere to interpolate the training points, while the differing frequencies and phases approximately destructively interfere on most of the remaining covariate support. Figure I.2 shows some of the functions generated by the hidden layer neurons of the spike component $f_{spikes}$. Both the phases and frequencies vary. Destructive interference in sums of many oszillations occurs, for example, under a uniform phase distribution.

An exciting direction of future work will be to understand when and why the neural networks with spiky-smooth activation functions learn the target function well, and when the decomposition into $ReLU$- and spike component succeeds to disentangle the noise from the signal. Particular challenges will be to design architectures and learning algorithms that provably work on complex data sets and to determine their statistical convergence rates. A different line of work could evaluate whether there exist useful spike components for deep and narrow networks beyond the pure infinite-width limit. Maybe for deep architectures is suffices to apply spiky-smooth activation functions only between the penultimate and the last layer.

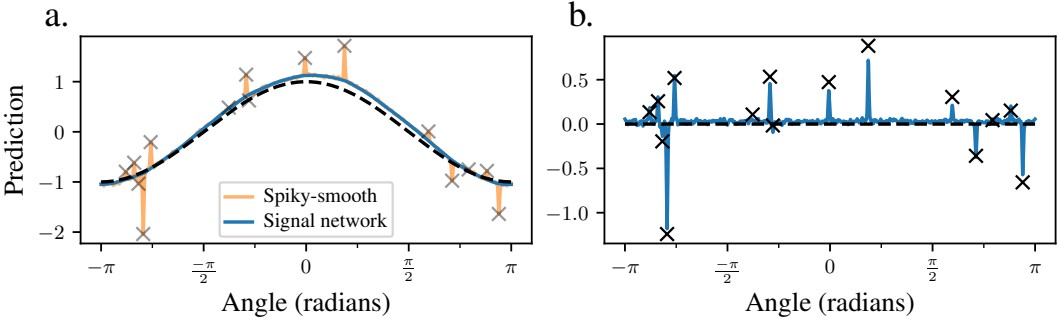

Figure I.1: **a.** The $ReLU$-component $f_{ReLU}$ (blue) and the full spiky-smooth network $f_{spsm}$ (orange) of the learned neural network from Figure 1. **b.** The spike component $f_{spikes}$ of the learned neural network from Figure 1 against the label noise in the training set, derived by subtracting the signal from the training points. Observe that the $ReLU$-component has learned the signal, while the spike component has fitted the noise in the data while regressing to $0$ between data points.

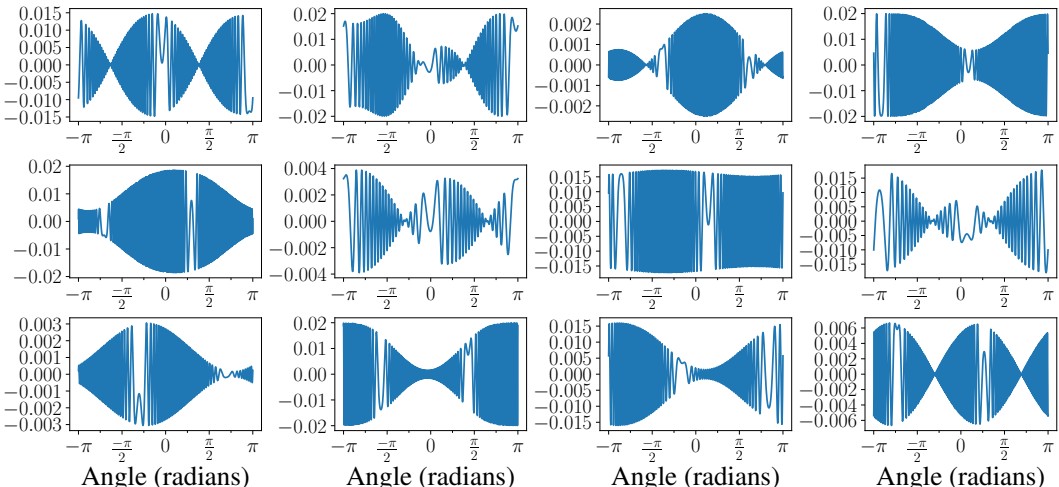

Angle (radians)       Angle (radians)       Angle (radians)       Angle (radians)

Figure I.2: Here we plot the functions learned by 12 random hidden layer neurons of the spike component network $f_{spikes}$ corresponding to Figure 1.

Of course an analogous additive decomposition exists for the minimum-norm interpolant $\hat{f}_0^k$ of the spiky-smooth kernel,

$$\hat{f}_0^k(\boldsymbol{x}) = (\tilde{k} + \rho_n \check{k}_{\gamma_n})(\boldsymbol{x}, \boldsymbol{X}) \cdot (\tilde{\boldsymbol{K}} + \rho_n \check{\boldsymbol{K}}_{\gamma_n})^{-1} \boldsymbol{y} = f_{signal}(\boldsymbol{x}) + f_{spikes}(\boldsymbol{x}), \qquad (\text{I.2})$$

where

$$f_{signal}(\boldsymbol{x}) = \tilde{k}(\boldsymbol{x}, \boldsymbol{X}) \cdot (\tilde{\boldsymbol{K}} + \rho_n \check{\boldsymbol{K}}_{\gamma_n})^{-1} \boldsymbol{y}, \qquad f_{spikes}(\boldsymbol{x}) = \rho_n \check{k}_{\gamma_n}(\boldsymbol{x}, \boldsymbol{X}) \cdot (\tilde{\boldsymbol{K}} + \rho_n \check{\boldsymbol{K}}_{\gamma_n})^{-1} \boldsymbol{y}.$$

We plot the results in Figure I.3. Observe that the spikes $f_{spikes}$ regress to $0$ more reliably than in the neural network.

Although spiky-smooth estimators can be consistent, any method that interpolates noise cannot be adversarially robust. The signal component $f_{signal}$ may be a simple correction towards robust estimators. Figure I.4 suggests that the signal components of spiky-smooth estimators behave more robustly than ReLU networks or minimum-norm interpolants of Laplace kernels in terms of finite-sample variance.

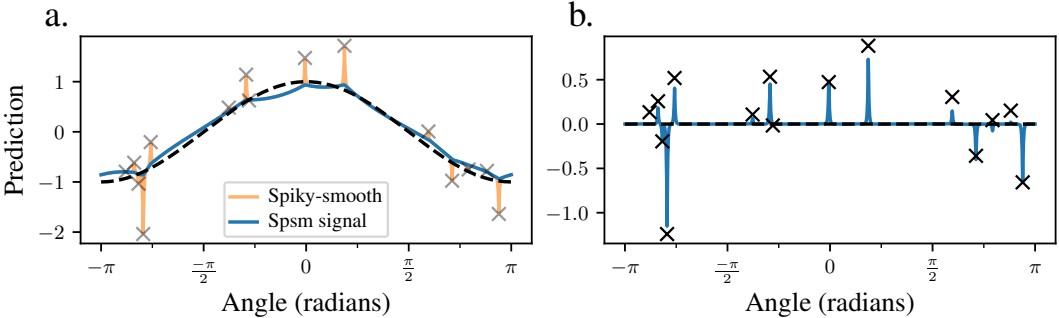

Figure I.3: **a.** The signal component $f_{signal}$ (blue) and the full minimum-norm interpolant $\hat{f}_0^k$ (orange) of the spiky-smooth kernel from Figure 1. **b.** The spike component $f_{spikes}$ of the spiky-smooth kernel from Figure 1 against the label noise in the training set, derived by subtracting the signal from the training points.

## I.3 Repeating the finite-sample experiments

We repeat the experiment from Figure 1 100 times, both randomizing with respect to the training set and with respect to neural network initialization.

For the kernels (Figure I.4a), observe that all minimum-norm kernel interpolants are biased towards 0. While the Laplace kernel and the signal component (I.2) of the spiky-smooth kernel have similar averages, the spiky-smooth kernel has a slightly larger bias. However, both the spiky-smooth kernel as well as its signal component produces lower variance estimates than the Laplace kernel.

Considering the trained neural networks (Figure I.4b), the ReLU networks are approximately unbiased, but have large variance. The neural networks with spiky-smooth activation function as well as the extracted signal network (I.1) are similar on average: They are slighly biased towards 0, but have much smaller variance than the ReLU networks.

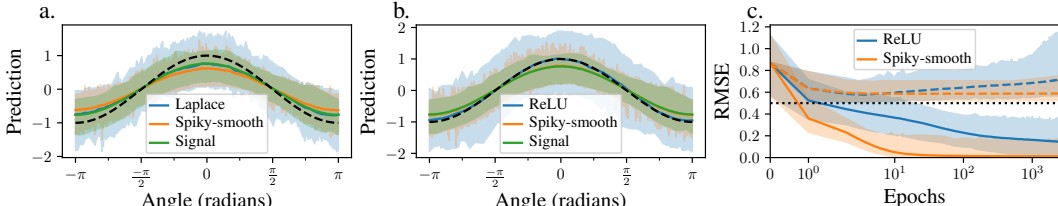

Figure I.4: We repeat the experiment from Figure 1 100 times and report the mean values (lines). Confidence bands denote the interval between the empirical 2.5%- and 97.5%-quantiles from the 100 independent runs.

The training curves (Figure I.4c) offer similar conclusions as Figure 1: While the ReLU networks harmfully overfit over the course of training, the neural networks with spiky-smooth activation function quickly overfit to 0 training error with monotonically decreasing test error, which on average is almost optimal, already with only 15 training points. The spiky-smooth networks have smaller confidence bands, indicating increased robustness compared to the ReLU networks. If the ReLU networks would be early-stopped with perfect timing, they would generalize similarly well as the networks with spiky-smooth activation function.

## I.4 Spiky-smooth activation functions

In Figures I.5 and I.6 we plot the 2-layer NTK activation functions induced by spiky-smooth kernels with Gaussian components, where $\tilde{k}$ has bandwidth 1, and in the first figure $\rho = 1$ while in the second figure $\rho = 0.1$.

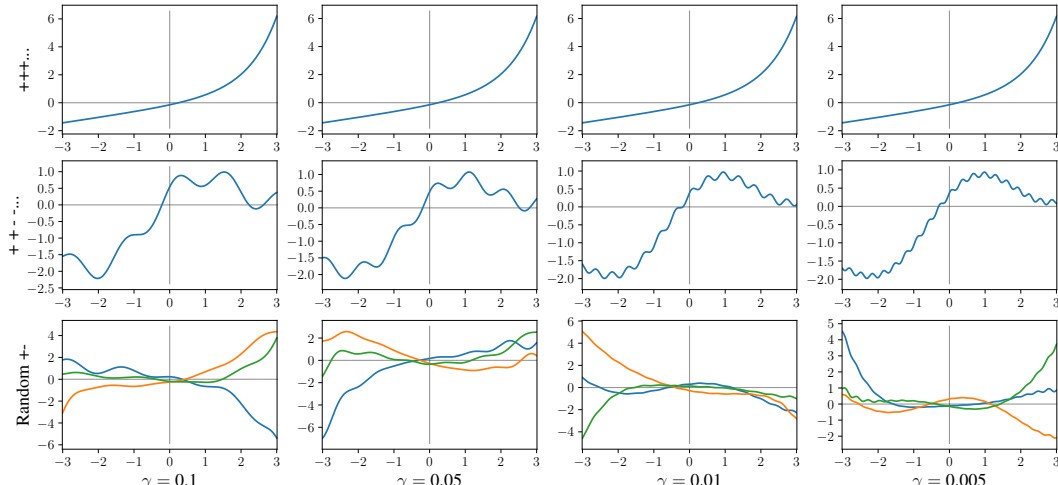

Figure I.5: The 2-layer NTK activation functions for Gaussian-Gaussian spiky-smooth kernels with varying $\gamma$ (columns) with $k_{max} = 1000$, $\tilde{k}$ has bandwidth 1, $\rho = 1$. Top: all $s_i = +1$, middle: $+, +, -, -, +, +, ...$, bottom: Random $+1$ and $-1$. Although the activation function induced by the spiky-smooth kernel is not exactly the sum of the activation functions induced by its components, this approximation is accurate because the spike components approximately live in a subspace of higher frequency in the Hermite basis orthogonal to the low-frequency subspace of the smooth component.

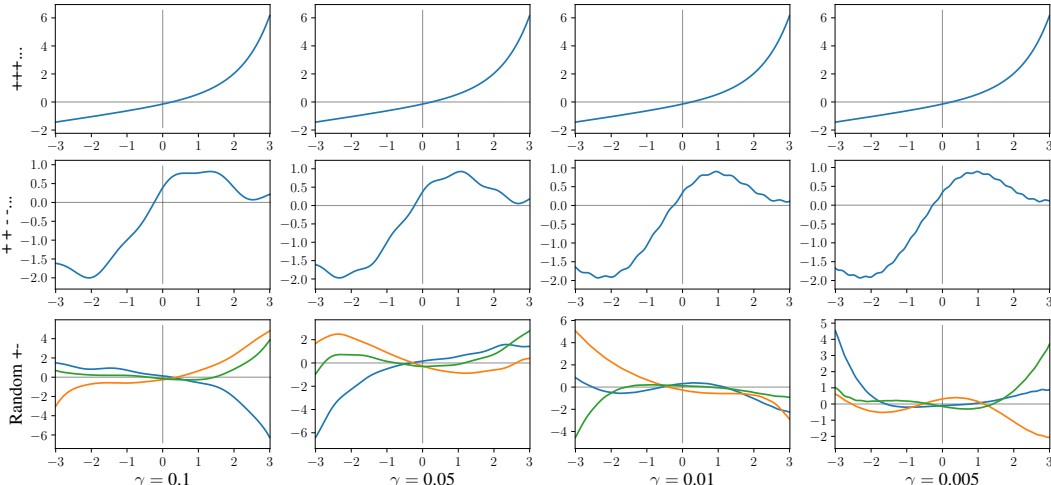

Figure I.6: Same as above but $\rho = 0.1$. The high-frequency fluctuations are much smaller compared to Figure I.5.

In Figure I.7 we plot the corresponding 2-layer NNGP activation functions with $\rho = 1$. In contrast to the NTK activation functions the amplitudes of the fluctuations only depends on $\rho$ and not on $\gamma$. Our intuition is the following: Since the first layer weights are not learned in case of the NNGP, the first layer cannot learn constructive interference, so that the oscillations in the activation function need to be larger.

The additive approximation $\phi^k \approx \phi^{\tilde{k}} + \rho^{1/2}\phi^{\tilde{k}_\gamma}$ remains accurate in all considered cases (Appendix I.6).

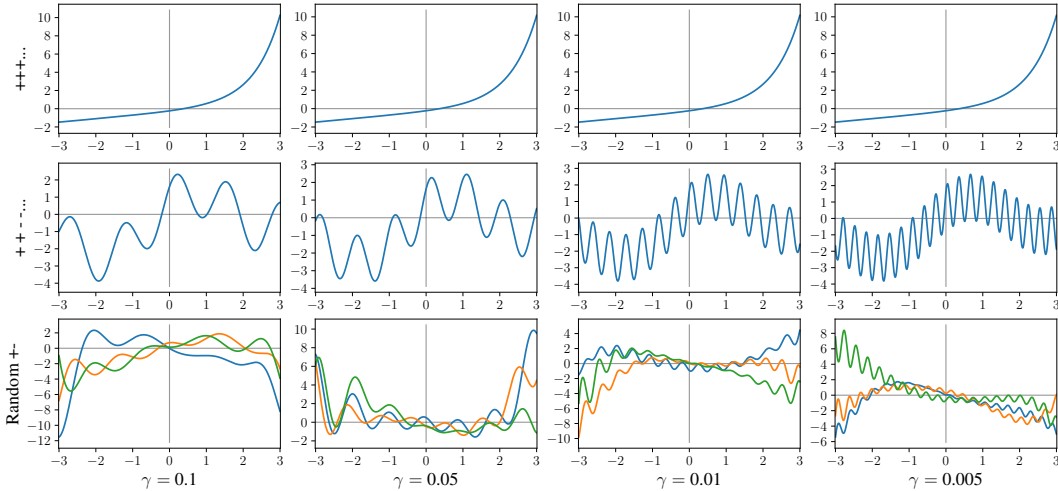

Figure I.7: Same as above but NNGP and $\rho = 1$. As expected from the isolated spike plots: Spikes essentially add fluctuations that increase in frequency and stay constant in amplitude for $\gamma \to 0$, $\rho$ regulates the amplitude.

## I.5 Isolated spike activation functions

Figure I.8 is the equivalent of Figure 3 for the NNGP.

By plotting the NTK activation components corresponding to Gaussian spikes $\phi^{\check{k}_\gamma}$ with varying choices of the signs $s_i$ in Figure I.9, we observe the following properties:

1. All $s_i = +1$ leads to exponentially exploding activation functions, cf. Eq. (H.2).
2. If the signs $s_i$ alternate every second $i$, i.e. $s_i = +1$ iff $\lfloor \frac{i}{2} \rfloor$ even, $\phi^{\check{k}_\gamma}$ is approximately a single shifted sin-curve with increasing frequency and decreasing/constant amplitude for NTK/NNGP activation functions, cf. Eq. (6).
3. If $s_i$ is chosen uniformly at random, with high probability, $\phi^{\check{k}_\gamma}$ both oscillates at a high frequency around 0 and explodes for $|x| \to \infty$.

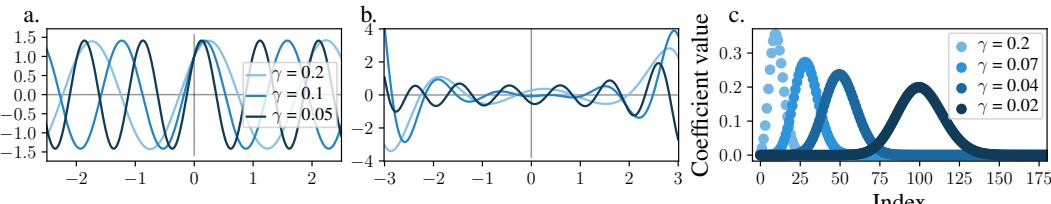

Figure I.8: Same as Figure 3 but for the NNGP. In contrast to the NTK, the amplitudes of the oscillations in a. do not shrink with $\gamma \to 0$. Otherwise the behaviour is analogous. For example, the Hermite coefficients peak at $2/\gamma$. The squared coefficients sum to 1 (Eq. (6)).

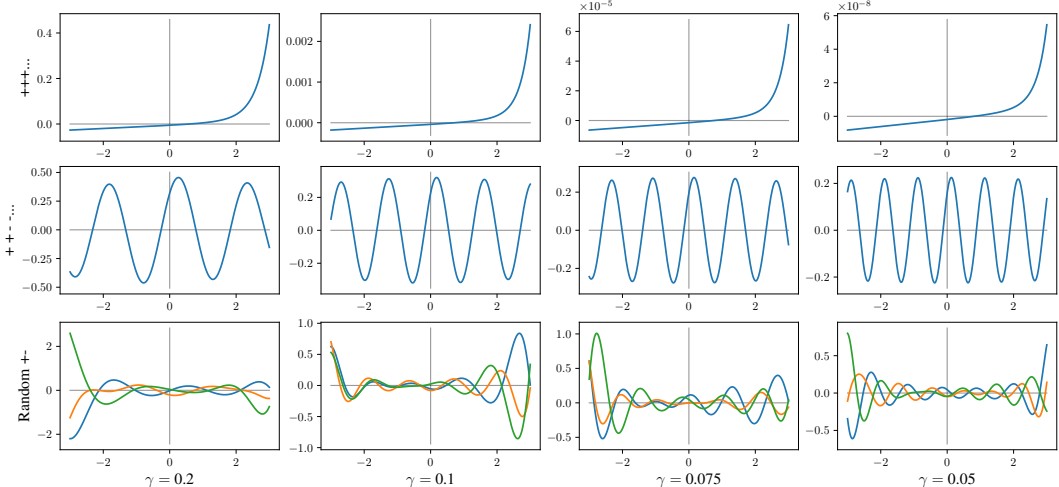

Figure I.9: The spike activation components of 2-layer NTK for Gaussian spikes with varying $\gamma$ (columns), $k_{max} = 1000$, top: all $s_i = +1$, middle: signs alternate every second index, bottom: 3 draws from uniformly random signs. With $\gamma \to 0$, the amplitude shrinks, while the frequency increases.

Figure I.10 visualizes NNGP activation functions induced by Gaussian spikes with varying bandwidth $\gamma$. Observe similar behaviour as for the NTK but amplitudes invariant to $\gamma$ as predicted by Eq. (6). For smaller $\gamma$ the explosion of (all+) activation functions starts at larger $x$, but appears sharper as can be seen in the analytic expression (H.2).

Figure I.11 resembles Figure I.9 but plotted on a larger range to visualize the exploding behaviour for $|x| \to \infty$.

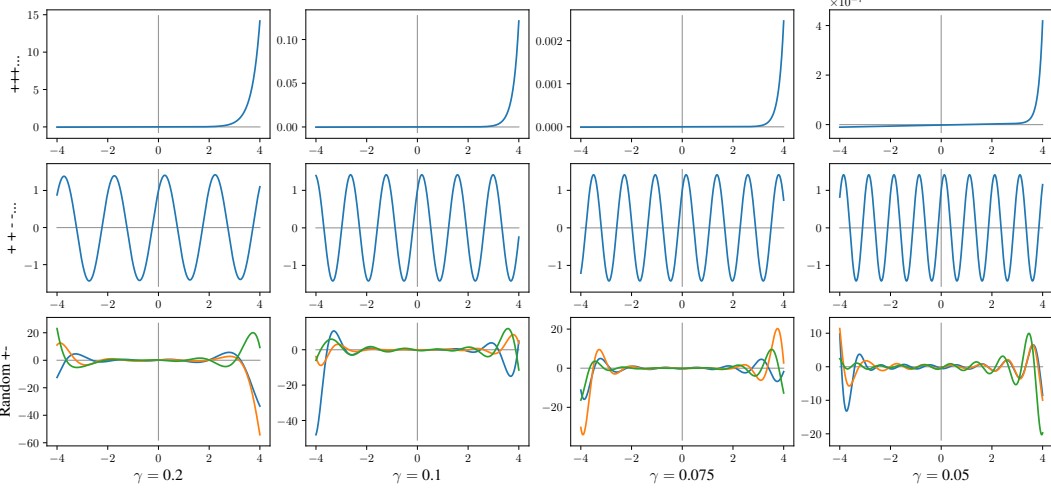

Figure I.10: Spike activation components as in Figure I.9, but for the NNGP with $x$ between $[-4, 4]$.

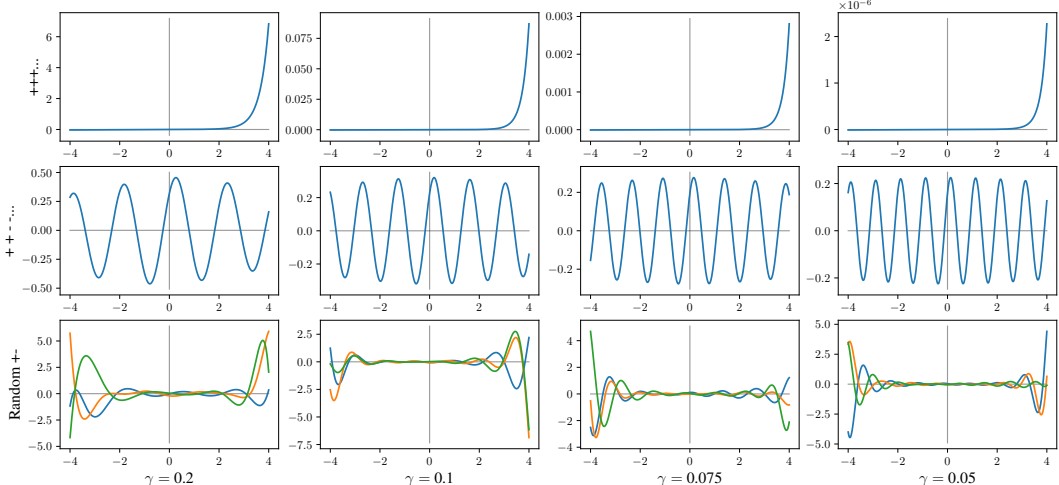

Figure I.11: Spike NTK activation components as in Figure I.9 but with $x$ between $[-4, 4]$. The all+ NTK activation explodes exponentially. While random sign activations explode as well, $+ + - -$- activations remain stable $\sin$-fluctuations with slowly decaying amplitude for $|x| \to \infty$.

## I.6    Additive decomposition and $\sin$-fit

Here we quantify the error of the $\sin$-approximation (8) of Gaussian NTK activation components. The additive decomposition $\phi^k \approx \phi^{\tilde{k}} + \rho^{1/2}\phi^{\check{k}_\gamma}$ quickly becomes accurate in the limit $\gamma \to 0$ (Figures I.12 and I.13), the $\sin$-approximation seems to converge pointwise at rate $\Theta(|x|\gamma)$, where a good approximation can be expected when $|x| \ll 1/\gamma$. The error at large $|x|$ arises because the spike component decays for $|x| \to \infty$. For $O(1)$ inputs, we conjecture that this inaccuracy does not dramatically affect the test error of neural networks when $\gamma$ is chosen to be small.

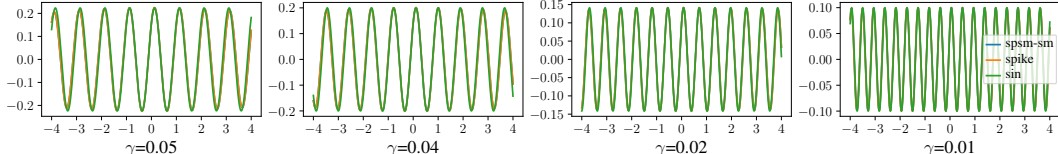

Figure I.12: The isolated NTK spike activation function (orange), the difference between spiky-smooth and smooth activation function (blue) and a fitted $\sin$-curve (8) (green). All curves roughly align, in particular for $\gamma \to 0$.

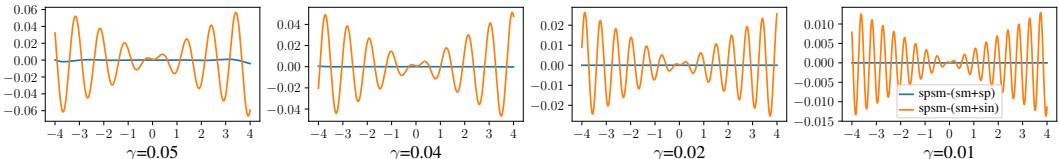

Figure I.13: The error of the additive decomposition $\phi^k \approx \phi^{\tilde{k}} + \rho^{1/2}\phi^{\check{k}_\gamma}$ (blue) and the $\sin$-fit (8) (orange) for the NTK. While the additive decomposition makes errors of order $10^{-3}, 10^{-4}, 10^{-9}, 10^{-15}$ (from left to right) in the domain $[-4, 4]$, the $\sin$-fit is increasingly inaccurate for $|x| \to \infty$, and increasingly accurate for $\gamma \to 0$.

Now we evaluate the numerical approximation quality of the $\sin$-fits (7) and (8) to the isolated spike activation components $\phi^{\check{k}_\gamma}$. As expected by Proposition H.1, the NNGP oscillating activation function $\phi^{\check{k}_\gamma}$ of a Gaussian spike component corresponds to Eq. (7) up to numerical errors. Both for the NNGP and for the NTK, the approximations become increasingly accurate with smaller bandwidths $\gamma \to 0$ (Figure I.14). Again the approximation quality suffers for $|x| \to \infty$, since $\phi^{\check{k}_\gamma}_{NTK}$ slowly decay to 0 for $|x| \to \infty$.

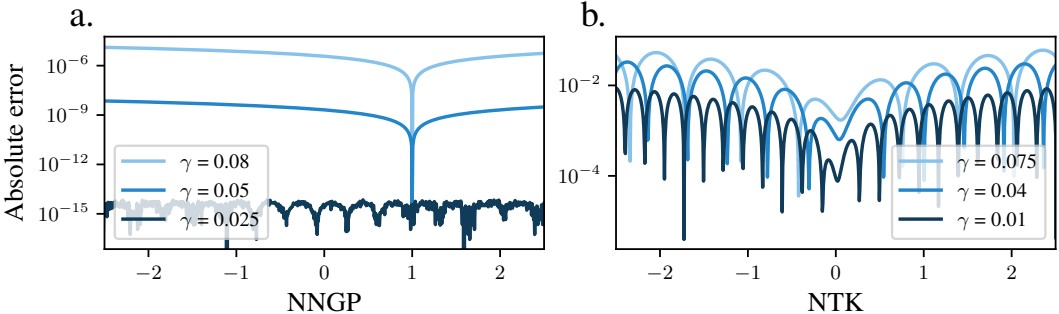

Figure I.14: Absolute numerical error between the oscillating activation function $\phi^{\check{k}_\gamma}$ of a Gaussian spike component and **a.** its analytical expression Eq. (7) for the NNGP and **b.** the approximation Eq. (8) for the NTK with varying bandwidth $\gamma$.

## I.7 Spiky-smooth kernel hyper-parameter selection

To understand the empirical performance of spiky-smooth kernels on finite data sets, we generate i.i.d. data where $\boldsymbol{x} \sim \mathcal{U}(\mathbb{S}^d)$ and

$$y = \boldsymbol{x}_1 + \boldsymbol{x}_2^2 + \sin(\boldsymbol{x}_3) + \prod_{i=1}^{d+1} \boldsymbol{x}_i + \varepsilon,$$

with $\varepsilon \sim \mathcal{N}(0, \sigma^2)$ independent of $\boldsymbol{x}$ and evaluate the least squares excess risk of the minimum-norm interpolant. Figure I.15 shows that

- the smaller the spike bandwidth $\gamma$, the better. At some point, the improvement saturates,
- $\rho$ should be carefully tuned, it has large impact. As with $\gamma \to 0$ ridgeless regression with the spiky-smooth kernel approximates ridge regression with $\tilde{k}$ and regularization $\rho$, simply choose the optimal regularization $\rho^{opt}$ of ridge regression.
- The spiky-smooth kernel with Gaussian components exhibits catastrophic overfitting, when $\gamma$ is too large (cf. Mallinar et al. (2022)), the Laplace kernel is more robust with respect to $\gamma$.
- With sufficiently thin spikes and properly tuned $\rho$, spiky-smooth kernels with Gaussian components outperform the Laplace counterparts.

We repeat the experiment in Figure I.16 with a slightly more complex generating function and come to the same conclusions.

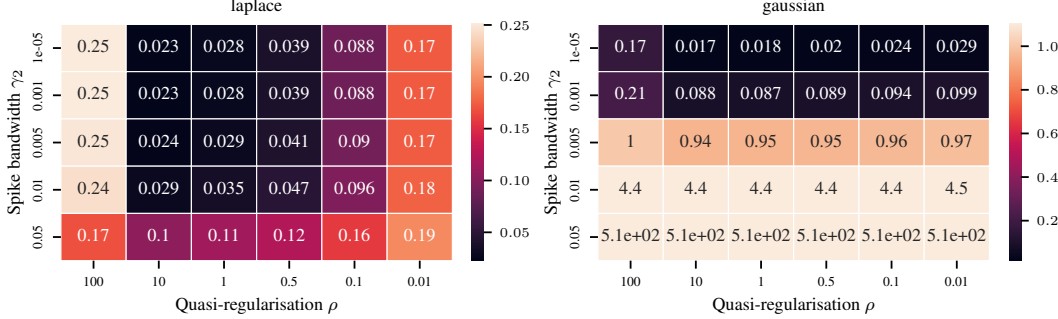

Figure I.15: Least squares excess risk for spiky-smooth kernel ridgeless regression with Laplace components (*left*) and Gaussian components (*right*), with $n = 1000, d = 2$, estimated on 10000 independent test points, $\sigma^2 = 0.5, \tilde{\gamma} = 1$. The smaller the spike bandwidth $\gamma$, the better. Properly tuning $\rho$ is important.

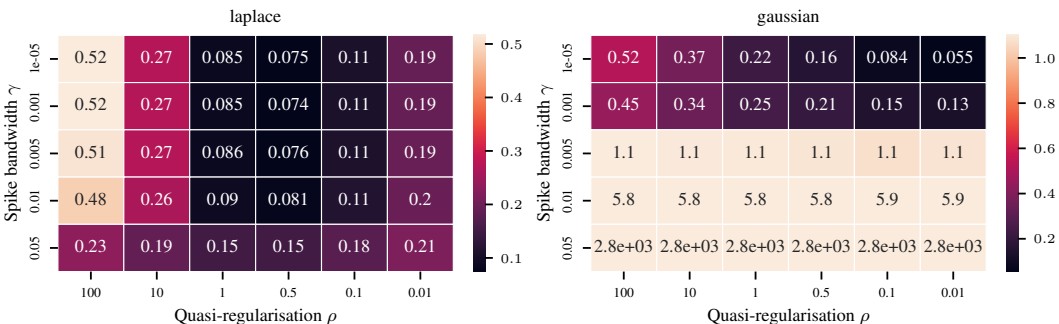

Figure I.16: Same as Figure I.15 but with the more complex generating function $y = |\boldsymbol{x}_1| + \boldsymbol{x}_2^2 + \sin(2\pi\boldsymbol{x}_3) + \prod_{i=1}^{d+1} \boldsymbol{x}_i + \varepsilon$. The errors are larger compared to Figure I.15 and the optimal values of $\rho$ are smaller, but the conceptual conclusions remain the same.

