# OpenReview forum: "Mind the spikes: Benign overfitting of kernels and neural networks in fixed dimension"
_NeurIPS.cc/2023/Conference — NeurIPS 2023 poster_

### Official Review · Reviewer_Si3s · 2023-06-29

**Soundness:** 3 good
**Presentation:** 3 good
**Contribution:** 3 good
**Rating:** 8
**Confidence:** 4

**Summary:**

This paper generalize existing inconsistency results to non-interpolating models and more kernels to show that benign overfitting with moderate derivatives is impossible in fixed dimension.
Moreover, this paper proves that interpolation with spiky-smooth kernels can be consistent and such kernels can be induced by certain activation functions.
There are also experiments supporting the authors' claims.

**Strengths:**

The paper is well-written and easy to follow.
It also provides us with novel understandings of the benign overfitting phenomenon: the smoothness of the estimators, and not the dimension, matters.
In particular, by considering spiky-smooth kernels, the authors find that interpolation with kernels can be consistent in fixed dimension.
This new result is very interesting.

On the technical side, the authors improve previous inconsistency results for RKHS equivalent to Sobolev RKHS of smoothness $s > d/2$ on the sphere $\mathbb{S}^d$. The techniques here are solid.



**Weaknesses:**

N/A

**Questions:**

1. Can Theorem 5 be extended to $H^s(\Omega)$, $\Omega \subset \mathbb{R}^d$ ?
If not, could the authors explain the difficulties here?

2. Is it possible to derive optimal rates for interpolation with spiky-smooth kernels?

**Limitations:**

The authors have adequately addressed the limitations.

---

> ### Author Rebuttal · Authors · 2023-08-09
>
>
> We thank you for carefully reading our paper and for providing detailed feedback.
>
> **Question 1: Can Theorem 5 be extended to $H^s(\Omega)$, $\Omega\subset \mathbb{R}^d$?**
>
> There is a technical obstacle to generalizing Theorem 5 to open bounded sets $\Omega \subseteq \mathbb{R}^d$, which could perhaps be overcome with additional technical contributions. This is discussed before Proposition F.12 in the appendix. Our proof uses that if $H_k \cong H^s$, then the RKHS $H_{k_*} \cong H^{2s}$, which is true for dot-product kernels on the sphere as shown in Proposition F.12. This allows us to apply kernel matrix eigenvalue bounds for Sobolev kernels to $k_*$. However, for kernels on open bounded sets $\Omega \subseteq \mathbb{R}^d$, this is in general not true, as the space $H_{k_*}$ will be slightly smaller than $H^{2s}$ due to some extra ``boundary conditions''. Intuitively, the sphere is a manifold without boundary and therefore less problematic in this regard. However, with a better understanding of the spaces $H_{k_*}$ for such $\Omega$, it might be possible to extend our proof to this setting. We will add a short discussion on this issue in the updated version of our manuscript.
>
> **Question 2: Is it possible to derive optimal rates for interpolation with spiky-smooth kernels?**
>
> Yes, very good point. Since minimum-norm interpolation with spiky-smooth kernels mimics ridge regression with smooth kernels, the rates from kernel ridge regression can be easily transferred. We agree that this is a valuable addition to the paper and will include this result in the updated version. More details can be found in the global response.

---

### Official Review · Reviewer_GSDW · 2023-07-01

**Soundness:** 3 good
**Presentation:** 3 good
**Contribution:** 3 good
**Rating:** 6
**Confidence:** 2

**Summary:**

This work studies the generalization behavior of overfitting methods in terms of the smoothness of the estimator, showing that only non-smooth estimators can interpolate benignly. They give a discussion of this result in the context of NTKs and their corresponding infinite-width architectures.

**Strengths:**

Originality and quality: this work seems to shed some new light on a well-studied problem. I'm not very familiar with background results on asymptotic risk w.r.t. function smoothness, so it's difficult for me to assess the first several theorem statements. I'm quite familiar with the latter results on NTKs, and to my knowledge this observation about spikiness in activation functions is new, and I find it well-explained.

Clarity: the paper seems pretty clear and well-written.

Significance: the math here is nice, and it seems like this might lead to some more clarity regarding exotic activation functions.

**Weaknesses:**

On significance: it's sort of unclear to me how this matters, or where we go from here. For example: the results of Mallinar et al. suggest that early stopping in nets (or including a ridge parameter in KRR, as shown by prior work too) is enough to make fitting consistent. These questions like "how do we design our model so it overfits benignly?" seem like they're actually missing some motivation -- why do we need to do that? Can't we just use a ridge parameter or optimal stopping?

The results around spike-inducing kernels seem quite hacky. The fact that the spike has to get smaller with dataset size makes it clear that this is just effectively adding a ridge. (The authors acknowledge this, but it's still not a surprising conclusion in my view.) It's also not useful -- the authors do not try to train their Hermite-polynomial nets, and (based on my own attempts to do such things) I suspect this may be because they're unstable, even at moderately large width!

As stated above, I'm unfamiliar with prior art around function smoothness and overfitting behavior, so there's a fair bit of uncertainty in my assessment here.

**Questions:**

The authors argue that the key object to consider is estimator smoothness, not input dimension. Given that there exists a body of results that do find that input dimension is important, is it clear how to reconcile the author's new results with these old ones? For example, do the smoothness requirements get more relaxed as dimension increases?

Note: I think an activation fn corresponding to the Gaussian NNGP kernel can be written down in closed form. (I do not know if this is true for the NTK, and would be curious to know if it can.)

Note: a high-frequency sinusoidal activation function can also give you a spiky kernel!

Note for future research: this smooth + spiky kernel thing seems like a hack. However, the results of Mallinar et al. suggest that if you could write down any kernel with an eigendecay like $\lambda_i \sim i^{-1} \log^{-\alpha} i$ for $\alpha > 1$, you'd get a consistent estimator without the need to scale down the kernel. It seems plausible to me that by taking a sort of exponent-equals-zero limit of Bietti and Bach's analysis, you could construct a kernel that actually gives you this decay (even on the unit circle), and it'd be consistent while being (a) a continuous function and (b) not needing to be scaled with dataset size. Could be an interesting thing to look into.

---

> ### Author Rebuttal · Authors · 2023-08-09
>
> We thank you for carefully reading our paper and for providing detailed feedback.
>
> **Remark 1: Benign overfitting lacks motivation.**
>
> The considerable interest in benign overfitting is mainly motivated by the great successes of overfitted NNs. In this sense, overfitting is motivated by empirical results, even though this overfitting might not be consistent or benign in a theoretical sense. A more detailed discussion of related work is provided in Appendix A. Since the community does not yet understand precisely why overfitting is often beneficial for NNs, it seems premature to dismiss approaches that can enable benign overfitting. On the other hand, it is not our main goal to improve overfitting in NNs, but rather to understand the conditions under which benign overfitting can occur. To the best of our knowledge, this paper is the first to show benign overfitting of a neural model in the challenging regime of low dimension, and rigorously establishing both consistency and inconsistency results for kernel regression posed several technical challenges. In the future work section, we acknowledge that we have not yet provided a neural method that is meant for practical use.
>
> **Remark 2: Spiky-smooth kernels seem hacky and not surprising.**
>
> Exploiting the connection to ridge regression is precisely what enables optimal nonparametric convergence rates. We will include this result in the updated version. More details can be found in the global response. Constructing our spiky-smooth kernel sequence as we do allows us to plug in Gaussian kernels and apply Theorem 3.1 of Simon et al. (2022). Spiky-smooth kernels are a simple idea and might not be surprising in hindsight, but we would argue that the consistency properties they can achieve are not obvious, and the simplicity is helpful in understanding their properties and the resulting implications, for example in deriving optimal rates for benign overfitting.
>
> **Remark 3: We do not train Hermite polynomial nets.**
>
> The Hermite polynomial basis only serves us as a tool to understand which activation functions give rise to benign overfitting. Since the shifted sine function emerges as a more practical solution, there is no need to train neural networks with Hermite polynomial activation functions. When training fully connected networks with the spiky-smooth activation function in Figure 1, we observe quite stable training trajectories at large widths. However, we acknowledge in the future work section that the spiky-smooth activation functions do not yet scale to large datasets. Identifying the right architectural inductive bias for moderate widths also poses an exciting opportunity for future work. Maybe this involves more than purely adapting the activation function. The goal of this theoretical study is rather to rigorously establish when and how benign overfitting with kernels and neural networks is possible in fixed dimension.
>
> **Question 1: What is the interplay between smoothness and input dimension?**
>
> Let us discuss this question from several angles.
>
> 1) **Order of smoothness.** Other papers find that input dimension is important for a "typical'' sequence of kernels $(k_d)_{d \in \mathbb{N}}$. Our results show that the dimension is not the single decisive quantity because one can also achieve benign overfitting in fixed dimension, even in Sobolev RKHS of arbitrary smoothness. Mallinar et al. (2022) consider, for example, Laplace-like kernels and semi-rigorously show that they overfit more benignly (tempered with decreasing constant in $d$) in high dimension. However, it is unclear how to compare their smoothness across dimensions: The Sobolev space smoothness $s = (d+1)/2$ increases with $d$, but the eigenvalue decay $\lambda_n = \Theta(n^{-(d+1)/d})$ gets slower.
>
> 2) **Smoothness in the sense of the magnitude of derivatives.** Here we refer to the global response. In short, the spike bandwidth is allowed to decay slower in high dimension. The order of smoothness of the RKHS is irrelevant, we achieve benign overfitting even by composing Gaussian kernels.
>
> **Remark 4: An activation function corresponding to the Gaussian NNGP kernel can be written down in closed form.**
>
> Thank you for pointing this out. We will include the derivation of Eq. (7) in the updated version. It just requires $e^{2xz-z^2}=\sum_{n=0}^\infty z^n H_n(x)/n!$ with $z=\pm i\sqrt{1/\gamma}$ and some elementary calculations. When choosing all signs in the Hermite expansion positive, the corresponding activation function was stated in closed from in Eq. (H.2). The problem with the spiky-smooth activation functions is that the corresponding Hermite coefficients are of the form $\pm\sqrt{a_n + b_n}$, where $a_n$ and $b_n$ are the Taylor coefficients of the smooth and spiky parts, respectively. So we cannot just add up the spiky and smooth activation functions. We do not see how one could write Gaussian NTK activation functions in closed form.
>
> **Remark 5: A high-frequency sinusoidal activation function induces a spiky kernel.**
>
> Yes, a high-frequency sinusoidal activation function induces a spiky kernel. This is the idea behind adding it to a low-frequency activation function to get a spiky-smooth kernel. Similar to the small bandwidth case in the proof of Theorem 1, a pure spiky kernel will not overfit benignly.
>
> **Remark 6: Consistency with a single kernel function.**
>
> Indeed this is an interesting question, though it might well be that the resulting convergence rates would be terrible. We have mentioned this idea in Remark G.4.
>
> **References:**
>
> Mallinar et al. "Benign, Tempered, or Catastrophic: A Taxonomy of Overfitting", NeurIPS 2022.
>
> Simon et al. "Reverse engeneering the neural tangent kernel", ICML 2022.

---

> > ### Comment · Reviewer_GSDW · 2023-08-12
> > **Response to response**
> >
> > I thank the authors for their detailed and thoughtful response. I'm more convinced that these results will prove interesting and clarifying for the community, and I've thus increased my score slightly.

---

### Official Review · Reviewer_jpYR · 2023-07-04

**Soundness:** 3 good
**Presentation:** 3 good
**Contribution:** 3 good
**Rating:** 6
**Confidence:** 3

**Summary:**

This paper extends previous results on the inconsistency of ridgeless kernel regression in fixed dimension by showing that non-interpolating estimators whose norm grows comparably to the minimum-norm interpolator are also inconsistent. On the other hand, it is shown that so-called spiky-smooth kernels whose derivatives grow with the number of samples are able to interpolate training data consistently. These results are further specialized to Neural Tangent Kernels and Neural Network Gaussian Processes, and experimental results support the theory.

**Strengths:**

* The paper is generally well-written with understandable and insightful theorem statements and proof ideas, even for non-experts.
* Modifying the kernel to achieve consistency in fixed dimension via inspirations from high-dimensional benign overfitting seems novel and interesting.
* The assumptions seem to be transparent and clearly stated in the main text.

**Weaknesses:**

* Having explicit rate estimates in addition to asymptotic consistency guarantees could provide further insight into how the spiky-smooth kernels/activations must be designed to achieve optimal performance.
* While mathematically exciting and valuable, the results might have limited applicability in modern settings with typically high-dimensional datasets.
* The results concerning the NTK can be applied to networks in the kernel regime (a.k.a. "lazy training"), which can only provide a partial picture as there is increasing evidence that in certain settings, neural networks can generalize better in the "feature learning" regime, see e.g. [1].

[1] Malach et al. "Quantifying the benefit of using differentiable learning over tangent kernels." ICML 2021.


**Questions:**

* I believe it can be helpful to add a discussion on the possibility of initializing the weights of a neural network such that training remains in the NTK regime, while the initial predictor is 0, which is an assumption throughout the work.

* In Equation (2) of Theorem 1, is there a hidden dependence on $\sigma$ in $c$? In particular, is this intuition incorrect that in the noiseless setting of $\sigma = 0$, consistent estimation ($c = 0$) must be possible?

* In line 232, it is mentioned that Section 5 achieves consistency by violating Assumption (N). To my understanding, it seems like Section 5 takes advantage of having the norm (of any estimator) depend on the number of samples. On the other hand, the specific statement of Assumption (N) seems to still hold in Theorem 7, as it is stated for the minimum-norm interpolant (hence $C_\text{norm} = 1$).

Minor questions/comments:

* In line 88, it is mentioned that training infinitely wide NNs with gradient flow corresponds to learning with the NTK. Perhaps it is worth mentioning that this is only true under a certain scaling that leads to the "lazy training" regime [2], and other smaller initialization scalings that lead to the "mean field/feature learning" regime can have better generalization performance [1].

* In line 95, rotation-invariant kernels are suddenly discussed without prior context. It could be useful to mention their importance given that under standard initializations the NTK is rotation-invariant.

[1] Malach et al. "Quantifying the benefit of using differentiable learning over tangent kernels." ICML 2021.

[2] Chizat et al. "On Lazy Training in Differentiable Programming." NeurIPS 2019.

**Limitations:**

Please see "Weaknesses".

---

> ### Author Rebuttal · Authors · 2023-08-09
>
> We thank you for carefully reading our paper and for providing detailed feedback. We agree that all of your suggestions are important clarifications and will include them in the updated version. They will certainly improve the updated version of the paper. Concretely, we will:
>
> - add a discussion on how we configure our NNs to stay close to the NTK limit, with an initial predictor that is zero, and mention feature learning limits.
>
> - add the sentence "The above kernels as well as NTKs and NNGPs of standard fully-connected neural networks are rotationally invariant." before mentioning rotation-invariant kernels.
>
> **Remark 1: Explicit convergence rates.**
>
> We agree that providing explicit convergence rates constitutes a valuable addition to the paper. Since the minimum-norm estimator of our spiky-smooth kernel sequence mimics kernel ridge regression, it achieves optimal convergence rates for Sobolev target functions. See the global response for more details. We agree that an analysis beyond the asymptotic limit will constitute an interesting line of future work, in particular to understand when and how finite neural networks can overfit benignly on small datasets.
>
> **Remark 2: Limited applicability on high-dimensional datasets.**
>
> Note that the task of benign overfitting is more challenging in fixed dimension, and that this paper is the first one to show that it can be achieved by a neural network. We show that standard ReLU networks (in the NTK parametrization) cannot overfit benignly in this setting and present first ideas on how to enable nearly optimal generalization while overfitting to noise.
>
> **Remark 3: The results only cover the kernel regime.**
>
> We agree that analysing feature learning neural networks constitutes an interesting line of future work beyond the scope for this paper. We will add a sentence in the future work section (see global response).
>
> At least for kernel regression, we will show in the updated version that the minimum-norm interpolant w.r.t. our spiky-smooth kernel sequence achieves optimal convergence rates for Sobolev spaces $H^s, s > d/2$. Hence, in fixed dimension, rate-optimal benign overfitting can already be achieved without feature learning.
>
> From a theoretical standpoint, it is not clear which function classes to study that represent real datasets more accurately to be able resolve the question how to design feature learning that outperforms kernel regression.
>
> You can find another perspective in our corresponding answer to Reviewer P771.
>
> **Question 1: Hidden dependence on $\sigma$ in $c$?**
>
> Without adaptive bandwidths, as in the formulation of Theorem 1 and for NTKs and NNGPs, or in the large bandwidth case, the dependence is of the form $c\sigma^2$. We will make the dependence on $\sigma$ explicit in the updated paper. However, for the more general bandwidth-independent version in Theorem D.1, the low-bandwidth case lower-bounds the estimator bias instead of the estimator variance, and this bias should in principle not depend on $\sigma^2$ but rather on something like $||f^*||$.
>
> **Question 2: Consistency in the noiseless setting?**
>
> We mostly study the estimator variance due to label noise in our lower bounds, which of course is zero when $\sigma^2 = 0$, but this does not say anything about the estimator bias. Whether kernel interpolation is consistent in the noiseless setting is, to our knowledge, an open question. While the kernel interpolation community often studies deterministic covariates $\boldsymbol{x}_i$, it should be possible to get consistency under relatively weak assumptions from classical theory (see e.g. Wendland, 2005) as long as $f^* \in \mathcal{H}_k$. There are also some results for functions $f^*$ in Sobolev spaces that are slightly less smooth than the RKHS (e.g. Theorem 4.2 in Narcowich et al., 2006), but these results require stronger assumptions on the separation distance of the $\boldsymbol{x}_i$, and it is unclear to us if this would be satisfied for random samples.
>
> **Remark 5: Violating Assumption (N).**
>
> Indeed we have $C_{\mathrm{norm}} = 1$ when comparing to the RKHS norm of the corresponding spiky-smooth kernel that depends on the number of samples, but $C_{\mathrm{norm}}$ grows unbounded when comparing to a fixed Sobolev norm. In the updated version, we will replace the sentence "The key is to violate Assumption (N) and allow for quickly exploding derivatives" by "The key is to violate Assumption (N) for every fixed Sobolev RKHS norm $||\cdot||_{H_k}$ and introduce an inductive bias towards learning spiky-smooth functions".
>
> **References:**
>
> Fischer and Steinwart "Sobolev Norm Learning Rates for Regularized Least-Squares", JMLR 2020.
>
> Narcowich et al. "Sobolev error estimates and a Bernstein inequality for scattered data interpolation via radial basis functions", Constructive Approximation, 2006.
>
> Wendland ``Scattered Data Approximation'', Cambridge University Press, 2005.

---

> > ### Comment · Reviewer_jpYR · 2023-08-18
> >
> > Thank you for your thorough response. I will maintain my positive evaluation of the work.

---

### Official Review · Reviewer_P771 · 2023-07-07

**Soundness:** 3 good
**Presentation:** 3 good
**Contribution:** 2 fair
**Rating:** 6
**Confidence:** 4

**Summary:**

In this paper, the authors studied the problem of benign overfitting for kernels and wide neural networks (in kernel regime) in fixed dimension. The authors showed that benign overfitting is possible if and only if the learner model has large derivatives. This implies that benign overfitting is not possible for those models with small derivatives (including ReLU NTK). On the other hand, they showed that for certain spiky-smooth activations/kernels, benign overfitting is achievable. Experiments are provided in the paper to verify the results.

**Strengths:**

1.	The paper is clearly written and easy-to-follow. The proof sketch is given to help the readers to understand the proof easier.
2.	Understanding benign overfitting phenomenon is an important problem for deep learning. The current paper focuses on the kernel regression setting and connects to neural networks via the neural tangent kernel (NTK) theory
3.	The idea of introducing spiky terms in NTK seem to be interesting. The observation in the proof that such spiky term in NTK behaves like regularization term so that the whole solution is now approximating kernel ridge regression also seems to be interesting.



**Weaknesses:**

1.	The current paper focuses on kernel regression and neural networks in the kernel regime. It would be more interesting to go beyond the kernel regime to see if similar results would also hold (e.g., in feature learning regime).

**Questions:**

1.	I was wondering what the dependency on dimension $d$ would be in the results such as Theorem 1.

**Limitations:**

The limitation is discussed in the paper. This is a theoretical work and therefore does not seem to have negative societal impact.

---

> ### Author Rebuttal · Authors · 2023-08-09
>
> We thank you for carefully reading our paper and for providing detailed feedback.
>
> **Remark: The results only cover the kernel regime.**
>
> We would like to point out that this paper is the first to establish benign overfitting with a neural model in the challenging regime of low dimension, and the first to rigorously establish benign overfitting in fixed dimension for kernel regression. This regime is very different from the high-dimensional setting, in which feature-learning neural networks usually excel. For Sobolev function spaces, kernel ridge regression achieves optimal non-parametric convergence rates, so that asymptotically there is no need to propose a feature learning neural method, but it was unclear whether benign overfitting can be achieved. In the updated version, we will show that our method matches the convergence rate of kernel ridge regression (see the global response for more details). This shows that benign overfitting is not only possible with kernels and wide neural networks, but can even be rate-optimal.
>
> We agree that benign overfitting results beyond the kernel regime would also constitute an interesting line of future work, but they are beyond the scope of this paper. We will mention the feature learning regime in the future work section (see the global response). You can find another perspective in our response to Reviewer jpYR.
>
> **Question: What is the dependency on the dimension $d$?**
>
> See the global response.

---

> > ### Comment · Reviewer_P771 · 2023-08-13
> >
> > Thanks for the response. I will keep my score.

---

### Official Review · Reviewer_m3mo · 2023-07-09

**Soundness:** 4 excellent
**Presentation:** 3 good
**Contribution:** 3 good
**Rating:** 7
**Confidence:** 3

**Summary:**

This paper explains that benign overfitting when the dimensionality of data is fixed is possible if one looks for estimators differently from conventional minimal norm. If one allows the estimator to be spiky, benign overfitting can still be possible. These results are extended via NTK to two-layer infinite-width neural networks. The authors showed that a nearly imperceptible sinusoid added to a RELU nonlinearity allows the trained neural network to overfit noisy data without compromising generalization.

**Strengths:**

(Please note that I could not have proofread the submission as well as I would have liked, due to various external obligations.) This paper extends the results on consistency / inconsistency for infinite-dimensional data to the finite dimensional data case. Although the main points - allowing the estimator to go out of its "smooth" way to fit noisy data - could have been argued to be expected, actually seeing this established is still a contribution of remark. The other main point - replacing commonly used non-linearity with one that contains tiny level of sinusoidal perturbation - is not expected by me, that's a very nice way of understanding the non-linearity for neural networks.

**Weaknesses:**

(Please note that I could not have proofread the submission as well as I would have liked, due to various external obligations.) This paper promotes benign overfitting, but it still leaves open whether not allowing overfitting is better or worse than going into benign overfitting, and within the space of benign overfitting how to reach good results, for a given (n,d).

In Section 6.2 I felt that the noise standard deviation of 0.25 and the neural tangent kernel of gamma = 1/5000, these two quantities are somehow linked. It could turn out that claiming that such a network can be trained to overfit without regularization might be a dangerous overstatement.

**Questions:**

N/A

---

> ### Author Rebuttal · Authors · 2023-08-09
>
> We thank you for carefully reading our paper and for providing detailed feedback.
>
> **Q1: Is regularization better than benign overfitting?**
>
> We would like to emphasize that we do not promote overfitting. Instead this paper is the first to show that benign overfitting of kernels and neural networks is possible in the classical limit $d$ fixed, $n\to\infty$, in which traditional statistics had suggested that training and test error should be balanced via explicit regularization.
>
> Indeed the question of when overfitting to noise can be helpful is an interesting one, but difficult to answer. From a theoretical perspective, when considering convergence rates for Sobolev target functions, kernel ridge regression already achieves optimal rates, as discussed in the detailed related work (Appendix A). In the updated version, we will show that our estimator achieves the same convergence rates if the spike bandwidth $\gamma$ converges to zero slightly faster, achieving benign overfitting with optimal rates in Sobolev spaces. This is easy to see since our estimator already mimics kernel ridge regression. From this perspective, we show that both explicitly regularized as well as interpolating estimators can be optimal and bad generalization is not implied by $0$ training error alone.
>
> From a practical perspective, there are two aspects: For kernel regression, our spiky-smooth kernels mimic ridge regularization and we do not expect them to perform better than kernel ridge regression.
> For neural networks, our approach to benign overfitting is not yet practical on realistic data set sizes, more complex architectures, or outside of the kernel regime (as we mention in the future work section), therefore a comparison would be premature.
>
> **Q2: Within the space of benign overfitting, how should one reach good results for given $(n, d)$?**
>
> The quasi-regularization $\rho$ can be chosen as in kernel ridge regression, for example via cross-validation, as we discuss in Appendix A. Since the typical distance between training points scales as $n^{-1/d}$ for continuous random variables in $\mathbb{R}^d$ (see for example Lemma 7 in Buchholz, 2022), the spike bandwidth $\gamma$ can always be chosen as for the most challenging case $d=1$. With increasing dimension it can be chosen to decay slower as $\gamma_{n,d}=\Omega(n^{-(2+\alpha)/d})$, $\alpha>0$, as explained in Remark G.2 (see also our global response). This leaves open some parameters such as layer width and learning rate, but this paper should rather be seen as a first possibility result than proposing a fully practical neural method.
>
> Regarding neural networks (Section 6.2), the value $\gamma = 1/5000$ still determines the width of the spike in the NTK. The spike should be thin enough but it still needs to be possible to approximate it with finite-width neural networks. The optimal width of the spike is not influenced by the noise standard deviation, but the ``regularization'' parameter $\rho$ that corresponds to the height of the spike should be.

---

### Author Rebuttal · Authors · 2023-08-09

We want to thank all reviewers for their detailed feedback. The following remarks have been raised multiple times and we will include a discussion in the revised version of our paper:

**Question 1: Can benign overfitting with spiky-smooth kernels achieve optimal rates?**

Since minimum norm interpolation with the spiky-smooth kernel sequence mimics kernel ridge regression with the smooth kernel component, it is easy to see that our estimator achieves the same convergence rates as kernel ridge regression, if the spike bandwidth $\gamma$ converges to zero slightly faster. By choosing the quasi-regularisation $\rho$ as for kernel ridge regression, we can therefore achieve benign overfitting with optimal nonparametric rates in Sobolev spaces (up to a $\log^2(n)$ term) (Fischer and Steinwart, 2020). As discussed in the detailed related work Appendix A, $\rho$ can for example be chosen via cross-validation. We will include this result in the updated version, which resolves the question whether benign overfitting with optimal nonparametric convergence rates is possible with kernel regression in fixed dimension.

**Question 2: What is the dependency on the dimension $d$?**

This question can be posed for our inconsistency results and for our consistency result.

For our inconsistency results, the constant $c$ in the lower bound can depend on the dimension $d$ as well as the specific choice of the kernel $k = k_d$ for different dimensions $d$. For specific sequences of kernels $(k_d)_{d \in \mathbb{N}}$, there are results by Liang and Rakhlin (2020) and Liang et al. (2020) which show a convergence to zero for $d \to \infty$, and a more quantitative but semi-rigorous result by Mallinar et al. (2022). In our case, obtaining an explicit dependency on $d$ would require at least stronger assumptions and very careful reworking of previous works. Specifically, it would require at least

- to consider how the constants of equivalence behave for varying $d$ in assumption (K)

- to fix one definition of the Sobolev norm, since the constants of equivalence between different equivalent Sobolev norm definitions might grow with $d$,

- to track the dependence of the constants on $d$ in our proofs and all previous works, especially Buchholz (2022) and the literature on Sobolev spaces he builds upon.

Now we discuss our consistency result. For continuous covariate distributions with upper bounded Lebesgue density, the typical distance between training points scales as $n^{-1/d}$ (see e.g. Lemma 7 in Buchholz (2022)). The spike bandwidth $\gamma$ has to decay to $0$ faster than this typical distance. For the example of the Laplace kernel, we can choose the spike bandwidth $\gamma_{n,d}=\Omega(n^{-(2+\alpha)/d})$, $\alpha>0$, as previously explained in Remark G.2. Hence, the spikes are indeed allowed to be less sharp with increasing dimension. When $d$ and $n$ have the same order of magnitude, the minimum-norm interpolants of common kernels suffice to achieve benign overfitting - under stronger, favorable distributional assumptions (Liang and Rakhlin, 2020). Irrespective of the dimension $d$, we achieve benign overfitting with estimators in RKHS of arbitrary degrees of smoothness.

**Limitation: Our results only cover the kernel regime.**

Since our analysis only covers the kernel regime, we will add the following sentence in the future work section: "Finite-sample analyses of moderate-width neural networks with feature learning parametrizations and other initializations could enable to understand how to induce a spiky-smooth inductive bias in feature learning neural architectures."

**References:**

S. Buchholz "Kernel interpolation in Sobolev spaces is not consistent in low dimensions", COLT 2022.

Fischer and Steinwart "Sobolev Norm Learning Rates for Regularized Least-Squares", JMLR 2020.

Liang and Rakhlin "Just Interpolate: Kernel “Ridgeless” Regression Can Generalize", Annals of Statistics, 2020.

Liang et al. "On the Multiple Descent of Minimum-Norm Interpolants and Restricted Lower Isometry of Kernels", COLT 2020.

Mallinar et al. "Benign, Tempered, or Catastrophic: A Taxonomy of Overfitting", NeurIPS 2022.

---

### Decision · Program_Chairs · 2023-09-21

**Decision:**

Accept (poster)

**Comment:**

The paper provides a refined perspective on the phenomenon of benign overfitting for kernels, highlighting the key role played by the smoothness of the estimator. While prior work emphasizes the role of the dimension, the paper shows that the dimension is not the single decisive quantity (one can also achieve benign overfitting in fixed dimension as highlighted by the authors). This is to my knowledge a novel perspective, which is recognized by all the reviewers.

Most reviewers were positive prior to the discussion period, some even increased their scores after discussing with the authors. Overall, the paper makes a novel contribution which is appreciated by the reviewers, I therefore recommend acceptance.